# DAVE 🗿: Distribution-Aware Attribution via ViT Gradient Decomposition

**Adam Wróbel** [* 1 2]   **Siddhartha Gairola** [* 3]   **Jacek Tabor** [1 4]   **Bernt Schiele** [3]   **Bartosz Zieliński** [1 5]
**Dawid Rymarczyk** [1 6]

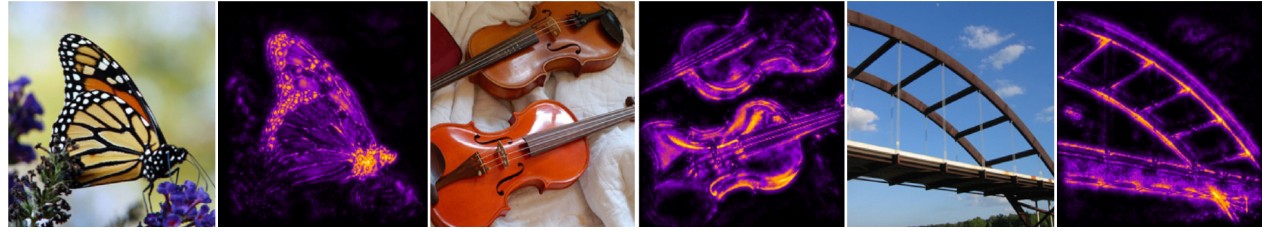

*Figure 1.* **DAVE provides fine-grained, pixel-level attributions that capture class-specific structural patterns present in each object.**
ImageNet-1k (Russakovsky et al., 2015) samples and corresponding DAVE attributions for DeiT III-B-16/224 (Touvron et al., 2022).

## Abstract

Vision Transformers (ViTs) have become a dominant architecture in computer vision, yet producing stable and high-resolution attribution maps remains challenging. Architectural components such as patch embeddings and attention routing often introduce structured artifacts in pixel-level explanations, leading many existing methods to rely on coarse patch-level attributions. We introduce DAVE *(Distribution-Aware Attribution via ViT Gradient DEcomposition)*, a mathematically grounded attribution method for ViTs based on a structured decomposition of the input gradient. By exploiting architectural properties of ViTs, DAVE isolates locally equivariant and stable components of the effective input-output mapping while suppressing architecture-induced artifacts and instability. Consequently, DAVE produces robust, precise, and class-consistent attribution maps that highlight model-relevant visual features. Experimental results show that across supervised, self-supervised, and inherently interpretable ViTs, DAVE outperforms prior methods on localization, faithfulness, and user studies. Code: `https://github.com/a-vrobell/DAVE`.

## 1. Introduction

Deep neural networks have redefined the state of the art in computer vision (Khan et al., 2022), yet their deployment in high-stakes applications such as medical diagnostics and autonomous driving remains limited by challenges in providing reliable and actionable decision explanations (Rudin, 2019). To address this, attribution methods in Explainable AI (XAI) have emerged (Bach et al., 2015; Selvaraju et al., 2017) which aim to identify input features that most influence model predictions. While they are widely adopted for their model-agnostic nature, yet producing *stable* and *high-resolution* attributions has become increasingly challenging as modern architectures grow more complex (Gairola et al., 2025).

Vision Transformers (ViTs) (Dosovitskiy et al., 2021) have become a dominant architecture in computer vision, achieving state-of-the-art performance across a wide range of tasks. Unlike CNNs, ViTs process images as sequences of patch tokens and rely on attention-based token mixing and learned projections. While effective for prediction, these architectural characteristics pose significant challenges for attribution methods (Komorowski et al., 2023). In particular, gradient- and attention-based explanations often exhibit structured, architecture-induced artifacts, leading either to unstable pixel-level attributions or to coarse patch-level explanations that lack fine-grained visual evidence (see Figure 2). As a result, obtaining stable and high-resolution attributions for Vision Transformers remains an open challenge.

To address these challenges, we introduce **DAVE**—(**D**istribution-aware **A**ttribution via **Vi**T Gradient **DE**composition), an attribution method for Vision Transformers that produces stable, high-resolution explanations. DAVE is based on a structured decomposition of the

---
[*]Equal contribution  [1]Jagiellonian University, Faculty of Mathematics and Computer Science [2]Jagiellonian University, Doctoral School of Exact and Natural Sciences [3]Max Planck Institute for Informatics, Saarland Informatics Campus, Saarbruecken, Germany [4]Credible AI, Warsaw University of Technology [5]Jagiellonian Center for Artificial Intelligence [6]Ardigen SA. Correspondence to: Dawid Rymarczyk <dawid.rymarczyk@uj.edu.pl>.

*Proceedings of the 43rd International Conference on Machine Learning*, Seoul, South Korea. PMLR 306, 2026. Copyright 2026 by the author(s).

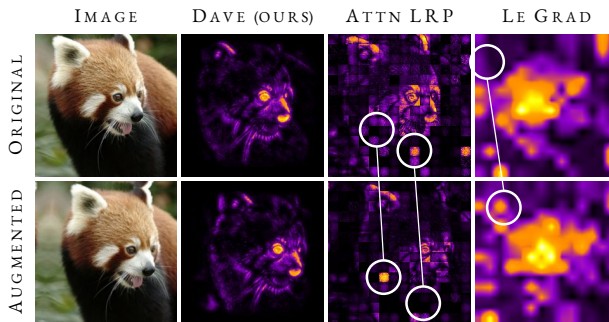

Figure 2. **Attribution consistency.** Under small augmentations (5° rotation, 20px horizontal and 8px vertical shift), DAVE highlights consistent features across the original and augmented images, while AttnLRP and LeGrad show inconsistent attributions (white markers), on a DeiT-III-B-16/224 model.

input gradient that exploits ViT architectural properties to identify locally equivariant and stable components of the effective input-output transformation. By suppressing architecture-induced artifacts and separating transformation effects from unstable input-dependent variations, DAVE enables robust pixel-level attribution without relying on coarse patch-based explanations (see Figure 1). As a result, the method highlights visual features consistently associated with model predictions.

Our **contributions** can be summarized as follows:

- We propose a mathematically grounded attribution framework based on a structured decomposition of the input gradient that identifies stable and locally equivariant components of the model's effective input-output transformation.

- We apply this framework to Vision Transformers as **DAVE**, leveraging ViT properties to suppress patch- and attention-induced artifacts and produce stable, high-resolution attributions.

- We demonstrate DAVE's effectiveness on multiple XAI benchmarks across supervised and self-supervised models (e.g., DeiT, DINO, and DINOv2 with register tokens (Touvron et al., 2021; Caron et al., 2021; Darcet et al., 2024)), and its versatility on other architectures, including inherently interpretable B-cos networks (Böhle et al., 2022).

## 2. Related Work

Model explainability in vision tasks has long focused on post-hoc analysis of convolutional architectures. However, the prominence of Vision Transformers has required a shift towards addressing the unique transformer-specific artifacts and long-range dependencies. In parallel, approaches that modify the model's architecture have been recently introduced with B-cos networks (Böhle et al., 2022) as an exam-

ple. In this section, we briefly review these directions and position DAVE among them.

**Architecture-agnostic attribution methods.** Early feature attribution methods mostly rely on gradient information or input sensitivity to identify salient regions. Saliency Maps (Simonyan et al., 2014) visualize the raw gradient of the output with respect to the input pixels to highlight influential regions. SmoothGrad (Smilkov et al., 2017) reduces visual noise by averaging gradients over multiple noisy copies of the input. Deconvolution (Mahendran & Vedaldi, 2016) computes attributions by backpropagating through a network while overriding ReLU gradients to propagate only non-negative signals. Guided Backpropagation (Rebuffi et al., 2020) further refines this by combining deconvolution with standard backpropagation to visualize features that specifically activate high-level neurons. Integrated Gradients (IG) (Sundararajan et al., 2017) provides an axiomatic foundation by integrating local gradients along a straight-line path from a baseline to the input, satisfying properties such as completeness and sensitivity. Grad-CAM (Selvaraju et al., 2017) uses the gradients of a target class flowing into the feature maps in a chosen layer to produce a localization map highlighting important regions. FullGrad (Srinivas & Fleuret, 2019) captures a more comprehensive signal by aggregating gradients with respect to both inputs and model biases. OMENN (Wrobel et al., 2024) proposes a decomposition of neural network operations to input-dependet operators, and combines them to identify a complete model attribution.

Most of these methods are not specific to ViTs and can be prone to artifacts induced by patch tokenization and attention routing. In contrast, DAVE is designed for ViTs.

**ViT-specific methods.** The shift toward self-attention mechanisms has motivated specialized techniques to handle patch-based interactions and attention artifacts (Lu et al., 2025). Raw attention visualizations use the model's inherent attention weights to show token interactions, though they often include significant background noise and are not necessarily class-specific. Rollout (Abnar & Zuidema, 2020) traces the flow of importance through the entire network by combining attention maps across all layers. Attention Flow (Abnar & Zuidema, 2020) casts the attention mechanism as a max-flow problem to trace information flow. CheferCAM (Chefer et al., 2021) weights self-attention maps by their gradients and aggregates them through the network to provide class-specific explanations. Iterated Integrated Attributions (IIA) (Barkan et al., 2023) generates explanation maps by iteratively integrating gradients and attention maps across all layers. AttnLRP (Achtibat et al., 2024) introduces redistribution rules to handle the non-linear softmax operations within the self-attention. LeGrad (Bousselham et al., 2025) sums layer-wise gradients with respect

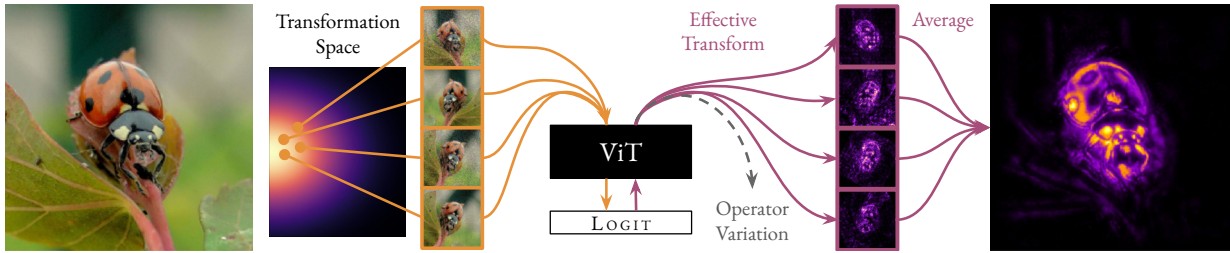

*Figure 3.* **Overview of the DAVE attribution pipeline for Vision Transformers.** Given an input image, DAVE samples small spatial transformations and Gaussian perturbations, computes the effective input–output transformation of the ViT for each sample, and filters out operator-variation term. The resulting attribution operators are inverse-transformed, averaged, and applied element-wise to the input to produce the final DAVE attribution map.

to attention maps, effectively filtering out "register token" artifacts and capturing hierarchical feature formation.

In contrast, DAVE decomposes the input gradient to isolate a locally equivariant and comparatively stable signal of the underlying effective transformation, reducing grid-like artifacts in ViT attributions. This is complementary to model-level artifact mitigation such as register tokens (Darcet et al., 2024), which add learnable tokens to absorb outlier activations during pretraining; we show in Appendix C.1 that DAVE provides further gains on DINOv2 with registers, indicating that the two directions are orthogonal.

**Inherently interpretable architectures.** Instead of post-hoc explanations, some approaches modify the model's architecture to be inherently interpretable (Chen et al., 2019; Brendel & Bethge, 2019). B-cos networks (Böhle et al., 2022) enforce stronger alignment between inputs and weights by replacing standard linear layers with B-cos transformations, making the weights themselves a faithful explanation. The B-cos Vision Transformer (Böhle et al., 2024) applies this principle to ViTs, replacing linear transformations within the transformer to promote human-interpretable weight-input alignment. B-cosification (Arya et al., 2024) is a novel technique that fine-tunes existing pre-trained models to adopt these transparent B-cos transformations at a fraction of the cost of training from scratch.

Unlike these approaches, DAVE improves explanations for standard pretrained ViTs, and we demonstrate that it can also be applied to inherently interpretable models such as B-cos networks.

## 3. DAVE

DAVE interprets attribution as the stable and locally equivariant *effective transformation* that a Vision Transformer applies to its input (Figure 3). Under architectural assumptions of ViTs, it extracts this transformation from the input gradient by decomposing it into a direct input–output transformation and a local variation term (Section 3.1). The method discards the local variation term, which captures

input-dependent sensitivity of internal model mechanisms, and aggregates the remaining transformation over a distribution of inputs. This distribution is constructed to suppress locally non-equivariant components of the transformation (Section 3.2), as well as high-frequency operator fluctuations (Section 3.3), while preserving consistent input-dependent attribution structure.

### 3.1. Extracting the Effective Transformation

The input gradient is commonly used as a measure of model sensitivity. However, the architectural structure of ViT layers reveals that their gradient entangles two distinct components: the *effective transformation* applied to the input and the variation of this transformation with respect to the input. In this section, we show how these contributions can be decomposed, enabling us to recover the effective transformation used as a baseline attribution signal.

#### 3.1.1. LAYER STRUCTURE

We begin by unifying the structure of ViT layers under a single definition, which allows for a principled analysis of their derivatives.

**Layer definition.** Let $V_{\text{in}}$ and $V_{\text{out}}$ be finite-dimensional real vector spaces. We model each ViT layer $F : V_{\text{in}} \to V_{\text{out}}$ as a differentiable operator-valued map $L : V_{\text{in}} \to \mathbb{L}(V_{\text{in}}, V_{\text{out}})$ followed by a constant bias addition[1]

$$F(\boldsymbol{X}) := L(\boldsymbol{X})(\boldsymbol{X}) + \boldsymbol{B} \tag{1}$$

where $\boldsymbol{B} \in V_{out}$ represents a constant bias parameter.

**Layer realisation.** Motivated by the architectural structure of ViT layers, we identify layer inputs and outputs with matrices in $\mathbb{R}^{t \times d_{\text{in}}}$ and $\mathbb{R}^{t \times d_{\text{out}}}$, where $t$ denotes the number of tokens and $d_{\text{in}}, d_{\text{out}}$ token dimensionalities. We consider

---
[1] $\mathbb{L}(V_{\text{in}}, V_{\text{out}})$ denotes the space of linear maps from $V_{\text{in}}$ to $V_{\text{out}}$. The map $L$ assigns to each input $\boldsymbol{X}$ a linear operator $L(\boldsymbol{X})$, yielding an input-dependent linear transformation, also referred as *dynamic linear* (Böhle et al., 2022; 2024).

two classes of maps $L$:

$$(I) \quad L(\boldsymbol{X})(\boldsymbol{X}) := \boldsymbol{W}_t(\boldsymbol{X}) \, \boldsymbol{X} \, \boldsymbol{W}_d$$
$$(II) \quad L(\boldsymbol{X})(\boldsymbol{X}) := \boldsymbol{\Phi}(\boldsymbol{X}) \odot \boldsymbol{X} \tag{2}$$

where $\boldsymbol{W}_t(\boldsymbol{X}) \in \mathbb{R}^{t \times t}$ is an input-dependent token-mixing matrix, $\boldsymbol{W}_d \in \mathbb{R}^{d_{in} \times d_{out}}$ is a static, dimensional projection, and $\boldsymbol{\Phi}(\boldsymbol{X}) \in \mathbb{R}^{t \times d_{in}}$ is an input-dependent gating. Although generated nonlinearly, these matrices act linearly on $\boldsymbol{X}$ for a fixed input, via matrix multiplication or elementwise multiplication.

These operators cover all ViT components, including attention (I), normalization (I), and pointwise activations (II); details are provided in the Appendix A.1.

**Layer derivative.** Under the considered layer definition (Equation 2) the derivative operator of $F$ decomposes into the effective transformation $L(\boldsymbol{X})$ and its variation with respect to the input:

$$\underbrace{D_{\boldsymbol{X}}F}_{\substack{\text{layer} \\ \text{derivative}}} = \underbrace{L(\boldsymbol{X})}_{\substack{\text{effective} \\ \text{transformation}}} + \underbrace{\left((D_{\boldsymbol{X}}L(\boldsymbol{X})(\cdot)) \, \boldsymbol{X}\right)}_{\substack{\text{operator} \\ \text{variation}}} \tag{3}$$

The effective transformation $L(\boldsymbol{X})$ captures the direct, input-conditioned action of the layer on its input. In contrast, the variation term corresponds to the derivative of an input-dependent operator and measures its changes under small input perturbations.

### 3.1.2. EFFECTIVE TRANSFORMATION

Operator variation (Equation 3) can amplify high-frequency and locally unstable components, leading to gradient-based attribution signals that are sensitive to small input perturbations (see Figure 4). Since our goal is to explain the effective transformation underlying a specific prediction rather than infinitesimal sensitivity, we omit the operator-variation term and retain only the effective transformation $L(\boldsymbol{X})$ (see Figure 5, columns 2 and 3).

This yields a pointwise effective operator. Subsequent sections reintroduce stable input-dependent structure by averaging the effective transformation over small neighborhoods of the input, while suppressing high-frequency artifacts.

**Effective transformation representation.** For fixed bases $\mathcal{B}_{\text{in}}$ and $\mathcal{B}_{\text{out}}$ of the input and output vector spaces of the layer, the effective transformation admits an input-dependent matrix representation $\boldsymbol{W}_L(\boldsymbol{X}) \in \mathbb{R}^{td_{\text{out}} \times td_{\text{in}}}$. Applied to the input vector $\boldsymbol{x} = [\boldsymbol{X}]_{\mathcal{B}_{\text{in}}} \in \mathbb{R}^{td_{\text{in}}}$, this matrix summarizes the direct action of the layer on its input:

$$[L(\boldsymbol{X})(\boldsymbol{X})]_{\mathcal{B}_{out}} = \boldsymbol{W}_L(\boldsymbol{X})\boldsymbol{x} \tag{4}$$

Importantly, $\boldsymbol{W}_L(\boldsymbol{X})$ is not a local sensitivity descriptor (as captured by the full Jacobian), but a matrix representation

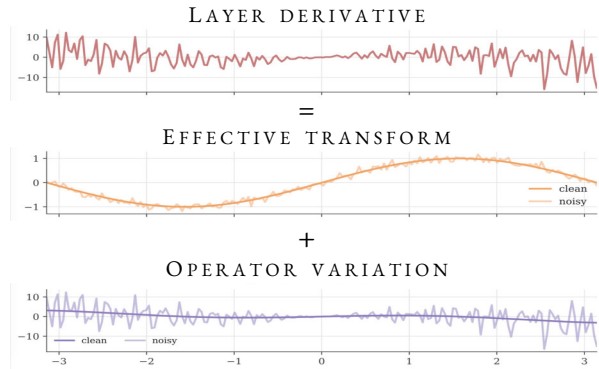

*Figure 4.* **Toy example:** illustrating operator variation dominating the layer derivative despite a stable effective transformation (Eq. 3). Top: full layer derivative (sum), dominated by operator variation. Middle: effective transformation with a small perturbation. Bottom: operator-variation term, where the perturbation is amplified.

of the input-conditioned transformation applied to $\boldsymbol{X}$ (see Appendix A.2 for details).

For a network composed of $n$ layers, the effective transformation of the entire model is obtained by composing the layerwise effective transformations:

$$\boldsymbol{W}_L(\boldsymbol{X}) = \prod_{i=n}^{1} \boldsymbol{W}_{L_i}(\boldsymbol{X}_{i-1}) \tag{5}$$

This input-dependent effective weight matrix combines attention weights, learned projections, normalization statistics, and gating across all layers, and serves as the baseline attribution operator used by DAVE.

The effect of isolating the effective transformation is further analyzed in Section 5.2, where we provide quantitative and qualitative comparisons between DAVE and its variant that retains the operator-variation term.

### 3.2. Extracting the Equivariant Transformation

The effective transformation often exhibits architecture-induced patterns that are stable across inputs, such as grid-like artifacts from patch embedding and attention routing. While these components contribute to the model's overall behavior, their relative input-invariance limits their usefulness for explaining individual predictions and can produce visually dominant but uninformative attribution patterns (Figure 5, column 3). We therefore introduce a Reynolds-inspired (Serre, 1977) operator that suppresses such artifacts by averaging over a local neighborhood of transformed inputs, isolating components that vary consistently under small spatial transformations. This allows us to isolate transformation-consistent attribution signal.

**Equivariance criterion.** Vision models typically recognize the same semantic content under small spatial transfor-

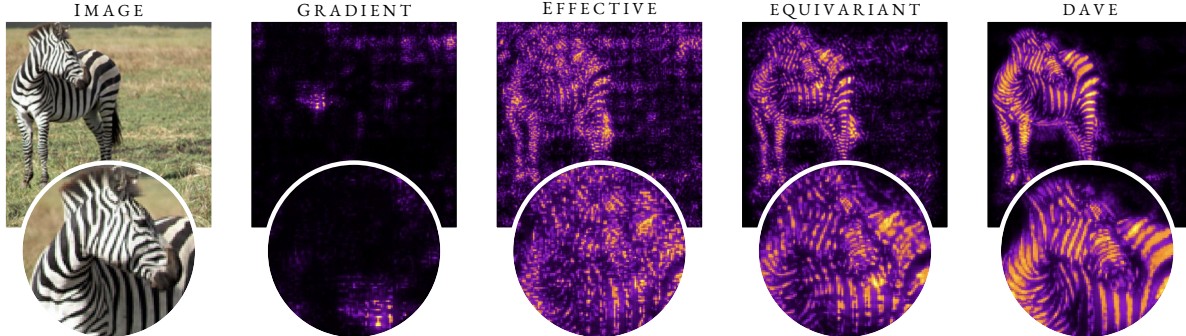

*Figure 5.* **Progressive construction of DAVE attribution.** The effective transformation captures the direct input–output action of the network while discarding operator-variation terms. Equivariant aggregation suppresses architecture-induced artifacts by enforcing local transformation consistency. DAVE further applies low-pass stabilization, yielding a stable and interpretable attribution map. Columns show input, input×gradient, input×effective transformation, input×equivariant transformation, and final DAVE attribution, respectively.

mations of the input, resulting in approximately invariant predictions. Since attribution aims to localize the visual evidence supporting a prediction, attribution maps should transform equivariantly under such transformations.

Formally, given an attribution map $\boldsymbol{W}_L(\boldsymbol{X})$, and a group of spatial transformations $\tau : V_{\text{in}} \to V_{\text{in}}$ we aim to extract a locally equivariant component satisfying[2]:

$$\boldsymbol{W}_L^{\text{eq}}(\boldsymbol{X}) \approx \left[\tau^{-1} \circ \boldsymbol{W}_L \circ \tau\right](\boldsymbol{X}) \qquad (6)$$

for transformations $\tau$ near the identity. This condition enforces that the attribution map transforms consistently under small spatial transformations of the input, providing a criterion for isolating locally equivariant attribution components.

**Equivariant transformation.** To suppress components of $\boldsymbol{W}_L(\boldsymbol{X})$ that violate this local consistency, we apply a Reynolds-inspired filtering operator defined over a group of spatial transformations.

Let $G$ be a compact group of spatial transformations (e.g., rotations), and let $\nu$ be a probability measure supported in a neighborhood of the identity, enforcing locality of the equivariance constraint. We define:

$$\boldsymbol{W}_L^{eq}(\boldsymbol{X}) = \int_G [\tau^{-1} \circ \boldsymbol{W}_L \circ \tau](\boldsymbol{X}) \, d\nu(\tau) \qquad (7)$$

This operator suppresses components of $\boldsymbol{W}_L(\boldsymbol{X})$ that violate local equivariance, while retaining signals consistent under transformations in $G$ (see Appendix A.3).

**Controlled equivariant variation.** While discarding the operator-variation term removes infinitesimal sensitivity, the subsequent Reynolds-style averaging captures

transformation-consistent changes across a local neighborhood of inputs, yielding controlled equivariant variation while avoiding amplification of high-frequency instabilities (see Figure 5, columns 3 and 4).

### 3.3. Low-Pass Filtering

To suppress high-frequency artifacts in the attribution signal, we apply a low-pass filtering operation. Specifically, we perform local averaging of the equivariant effective transformation by evaluating it under small input perturbations drawn from a local probability distribution and taking the expectation. For Gaussian perturbations, this operation is equivalent to convolution with the corresponding smoothing kernel $\mathcal{K}$, as shown in Appendix A.4:

$$\mathbb{E}_{\boldsymbol{\epsilon} \sim \mathcal{N}(0,\Sigma)} \left[\boldsymbol{W}_L^{\text{eq}}(\boldsymbol{X} + \boldsymbol{\epsilon})\right] = (\boldsymbol{W}_L^{\text{eq}} * \mathcal{K})(\boldsymbol{X}) \qquad (8)$$

This convolution attenuates attribution components that are unstable under small input variations (see Figure 5 column 5). Although this operation is analogous in form to SmoothGrad-style (Smilkov et al., 2017) averaging, it is applied to the equivariant effective transformation rather than to raw input gradients, thereby stabilizing the attribution without reintroducing operator-variation artifacts.

### 3.4. DAVE Framework

We summarize the DAVE attribution framework by combining equivariant filtering and low-pass stabilization into a single expression. The final distribution-aware effective transformation (i.e., the attribution-relevant effective input–output transformation) is obtained by taking the expectation of the effective transformation under local spatial transformations and small input perturbations. For a Gaussian smoothing kernel, this yields:

$$\tilde{\boldsymbol{W}}_L^{\text{eq}}(\boldsymbol{X}) = \mathbb{E}_{\tau \sim \nu, \, \boldsymbol{\epsilon} \sim \mathcal{N}(0,\Sigma)} \left[\tau^{-1}(\boldsymbol{W}_L(\tau(\boldsymbol{X} + \boldsymbol{\epsilon})))\right]$$
$$(9)$$

---

[2]We assume that the network outputs a scalar logit ($\dim(V_{\text{out}}) = 1$), so that the effective transformation $\boldsymbol{W}_L(\boldsymbol{X})$ has the same dimensionality as the input, as in gradient-based attribution methods.

where $\tau \sim \nu$ denotes a spatial transformation drawn from a probability measure supported in a neighborhood of the identity, and $\epsilon \sim \mathcal{N}(0, \Sigma)$ denotes Gaussian noise.

**Attribution map.** The DAVE attribution map is obtained by applying this distribution-aware effective transformation to the input via element-wise multiplication:

$$\text{DAVE}(\boldsymbol{X}) = \tilde{\boldsymbol{W}}_L^{\text{eq}}(\boldsymbol{X}) \odot \boldsymbol{X} \qquad (10)$$

This element-wise product follows from the matrix form of the effective transformation (Equation 4): the filtered operator defines an input-dependent linear map, so each input dimension contributes proportionally to its value weighted by the corresponding entry of the effective transformation.

**Practical realisation.** We compute the effective transformation by formulating network layers according to Equation 2 and detaching the input dependence of the operators (attention weights, GELU multipliers, and LayerNorm statistics) during a modified forward pass. The layerwise effective transformations are then aggregated during a single backward pass, yielding the effective transformation of the full model. This procedure is closely related to implementation techniques used in (Böhle et al., 2022; 2024; Wrobel et al., 2024) and allows the effective transformation to be computed with one modified forward and backward pass per Monte Carlo sample.

The expectation in Equation 9 is approximated via Monte Carlo sampling. We draw a small number of spatial transformations from the distribution $\nu$ and evaluate the effective transformation at the corresponding transformed inputs, additionally applying a small Gaussian perturbation. The final distribution-aware effective transformation is obtained by averaging the inverse-transformed attribution operators across samples (Figure 3). Full pseudocode is provided in Appendix B.

## 4. Experimental Setup

In this section, we describe the evaluation setup for **DAVE**. We evaluate explanations along two quantitative metrics, **localization** and **faithfulness**, complemented by **human user studies**. Additional details are in the Appendix B.

**Models.** We evaluate DAVE on standard ViT-based classifiers, including ViT-B/16 (Dosovitskiy et al., 2021), DeiT-B/16, and DeiT-III-B/16 (Touvron et al., 2021; 2022), as well as a self-supervised DINO ViT-B/16 model (Caron et al., 2021), all at $224 \times 224$ input resolution. For all models, we use the authors' publicly released pretrained checkpoints from their official repositories. We additionally evaluate on inherently interpretable B-cos models (B-cos-ViT and B-cos-ViT-C) (Böhle et al., 2024), which provide class-specific explanations and serve as strong transparent baselines.

*Table 1.* **Localization on conventional models.** GridPG (left) and EnergyPG (right) on ImageNet-1k. $\Delta$ denotes the improvement of DAVE over the best competing method (higher is better).

| Method | GridPG (%) | | | | EnergyPG (%) | | | |
|---|---|---|---|---|---|---|---|---|
| | ViT-B | DeiT-B | D-III-B | DINO-B | ViT-B | DeiT-B | D-III-B | DINO-B |
| I×G | 32.67 | 30.25 | 30.01 | 33.28 | 55.33 | 62.72 | 67.32 | 69.98 |
| IntGrad | 39.86 | 36.11 | 31.68 | 36.98 | 58.51 | 64.22 | 68.12 | 72.15 |
| S-Grad | 34.27 | 30.18 | 31.48 | 33.13 | 56.02 | 60.78 | 68.10 | 70.93 |
| LeGrad | 47.71 | 42.58 | 34.62 | 28.96 | 80.06 | 77.83 | 77.54 | 82.26 |
| A-LRP | 58.40 | 54.63 | 53.84 | 37.49 | 60.75 | 68.16 | 77.65 | 75.98 |
| C-LRP | 54.98 | 55.47 | 52.27 | 49.99 | **80.82** | 79.62 | 81.94 | 81.56 |
| DAVE (ours) | **60.19** | **63.52** | **65.76** | **51.33** | 78.60 | **82.23** | **82.43** | **83.38** |
| $\Delta$ | +1.79 | +8.05 | +11.92 | +1.35 | -2.22 | +2.61 | +0.49 | +1.12 |

*Table 2.* **Localization on B-cos models.** GridPG (left) and EnergyPG (right) on ImageNet-1k. $\Delta$ denotes the improvement of DAVE over the inherent B-cos explanation (higher is better).

| Method | Grid PG (%) | | Energy PG (%) | |
|---|---|---|---|---|
| | B-cos-ViT | B-cos-ViT-C | B-cos-ViT | B-cos-ViT-C |
| I×G | 55.71 | 55.38 | 65.29 | 66.72 |
| IntGrad | 57.98 | 57.68 | 65.33 | 67.75 |
| S-Grad | 60.27 | 61.30 | 68.16 | 69.59 |
| B-cos | 79.67 | 87.66 | 69.41 | 75.61 |
| DAVE (ours) | **84.00** | **88.43** | **78.55** | **79.63** |
| $\Delta$ | +4.33 | +0.77 | +9.14 | +4.02 |

**Datasets.** We use the publicly available ImageNet-1k (Russakovsky et al., 2015) dataset for all attribution evaluations and qualitative comparisons. For localization experiments, we evaluate on the ImageNet validation set using the official bounding-box annotations (metrics defined below).

**Baselines.** We compare against representative gradient-based and transformer-specific attribution methods: Input×Gradient (I×G) (Shrikumar et al., 2017), Integrated Gradients (IntGrad) (Sundararajan et al., 2017), Smooth-Grad (S-Grad) (Smilkov et al., 2017), LeGrad (Bousselham et al., 2025), AttnLRP (A-LRP) (Achtibat et al., 2024), and Chefer-LRP (C-LRP) (Chefer et al., 2021). For B-cos networks, we additionally include the model inherent B-cos explanations (Böhle et al., 2024).

**Attribution map generation.** All methods produce class-conditional attribution maps for the predicted class at the model input resolution. For methods that operate at patch resolution (*e.g.,* C-LRP, LeGrad), we upsample attribution maps to pixel space using bilinear interpolation for evaluation. Unless stated otherwise, we follow the authors' official implementations and default hyperparameters.

**Localization metrics.** We quantify attribution localization using two pointing-game metrics: *Grid Pointing Game (GridPG)* and *Energy Pointing Game (EnergyPG)*. GridPG measures whether the maximally attributed location falls inside the ground-truth object region, while EnergyPG measures the fraction of attribution mass contained inside the ground-truth bounding box (higher is better for both).

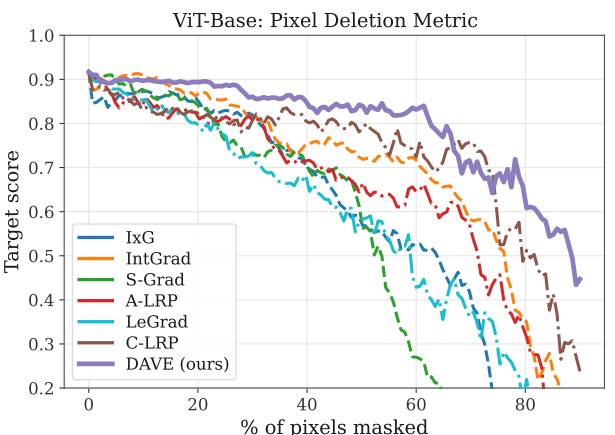
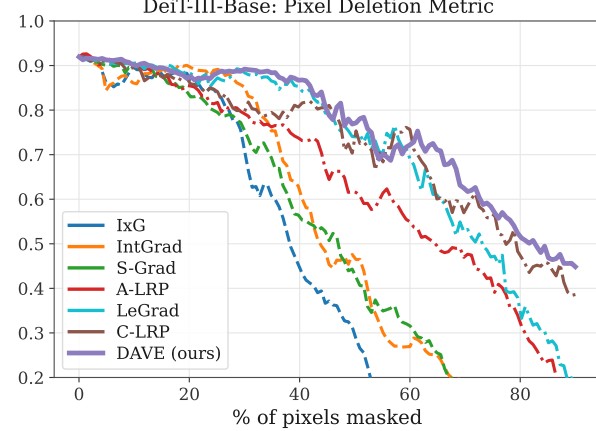

*Figure 6.* **Pixel deletion faithfulness.** Target-class probability versus fraction of pixels removed on ViT-B/16 (left) and DeiT-III-B/16 (right). DAVE (purple) yields the flattest curves (higher AUC), indicating the strongest stability under deletion.

**Faithfulness metric.** We evaluate faithfulness using *pixel deletion* curves for all models. We progressively remove pixels in order of increasing attribution (least to most important) and track the target-class probability. We plot the target-class probability versus the fraction of pixels removed; flatter curves highlight more stable attributions that are consistent with the model's decision. We follow standard deletion protocols from prior work (Chefer et al., 2021).

**Human user studies.** To complement the quantitative metrics, we conduct two user studies with 120 participants. The first is a *pairwise preference* task in which participants are shown an input image and two attribution maps from different methods, and asked which explanation better helps verify that the model made its prediction for the right reasons. The second follows the HIVE protocol (Kim et al., 2022): participants are shown attribution maps for the model's top-4 predicted classes and asked to identify the predicted class from the attributions alone. Both tasks measure whether attribution maps convey class-discriminative evidence in a form that is actionable to a human evaluator; full study design, interface, and per-method results are in Appendix C.4.

## 5. Results

In this section, we present quantitative and qualitative results for DAVE compared to prior attribution methods.

### 5.1. Main Results

**Localization.** Table 1 reports localization (%) on conventional ViTs. DAVE improves **GridPG** across all evaluated models. On ViT-B/16, DAVE achieves **60.19**% GridPG, improving over the strongest competing method by **+1.79** p.p. On DeiT-B/16 and DeiT-III-B/16, DAVE yields **63.52**% and **65.76**% GridPG, with gains of **+8.05** and **+11.92** p.p., respectively. On the self-supervised DINO-B/16 back-

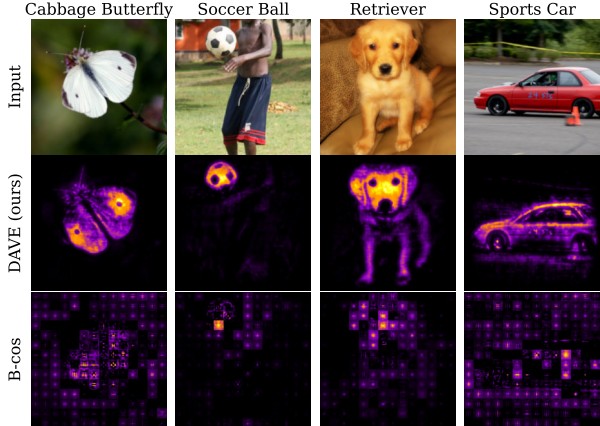

*Figure 7.* **Improved attributions for B-cos models.** We show ImageNet inputs (top), DAVE (middle), and inherent B-cos explanations (bottom). DAVE produces sharper, more object-aligned attributions with reduced background responses.

bone, DAVE reaches **51.33**% GridPG, improving over the strongest competing method by **+1.35** p.p. For **EnergyPG**, DAVE achieves the best results on DeiT-B/16 (**82.23**%) and DeiT-III-B/16 (**82.43**%), and is competitive on ViT-B/16. On DINO-B/16, DAVE achieves **83.38**% EnergyPG, improving over the strongest competing method by **+1.12** p.p.

Table 2 reports localization (%) on B-cos models. DAVE improves over the inherent B-cos explanations on both GridPG and EnergyPG. For B-cos-ViT, DAVE achieves **84.00**% GridPG and **78.55**% EnergyPG, improving over B-cos by **+4.33** and **+9.14** p.p., respectively. For B-cos-ViT-C (with a convolutional stem), DAVE achieves **88.43**% GridPG and **79.63**% EnergyPG, corresponding to gains of **+0.77** and **+4.02** p.p. over B-cos.

**Faithfulness via pixel deletion.** Figure 6 shows pixel deletion curves on ViT-B/16 (left) and DeiT-III-B/16 (right), tracking target-class probability versus the fraction of pixels

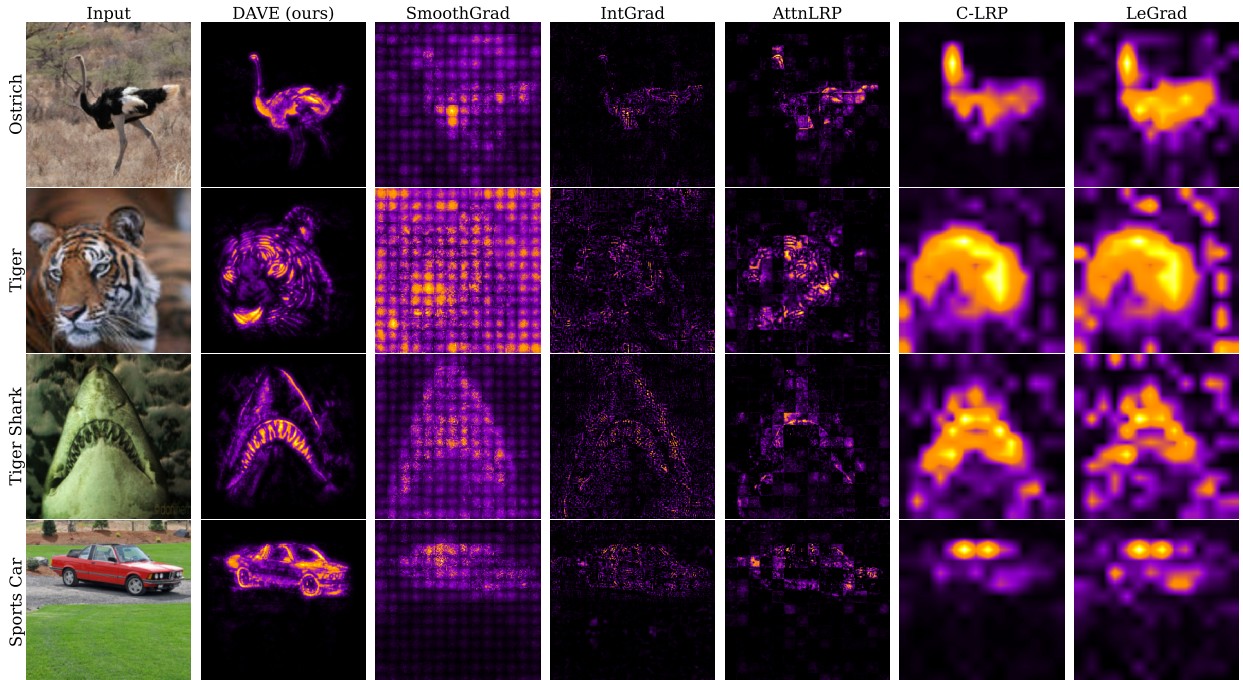

*Figure 8.* **Qualitative comparison of attribution maps.** Dave yields sharper, more object-aligned explanations with reduced background noise compared to prior methods. Note that for a set of ImageNet-1k examples (rows), we show the input image (left) and attribution maps produced by **DAVE (ours)**, SmoothGrad, IntGrad, AttnLRP, C-LRP, and LeGrad (columns).

removed. DAVE remains the most stable under deletion, exhibiting the flattest curves compared to prior methods.

**Human user studies.** We assess the practical usefulness of attributions through two studies with 120 participants on DeiT-III-B/16 (protocols and full results in Appendix C.4). In the *pairwise preference* task, DAVE is preferred over every baseline in **70–77**% of comparisons ($p < 0.001$). In the HIVE-style (Kim et al., 2022) *class-identification* task, where participants infer the predicted class from attribution maps alone, DAVE enables the highest accuracy at **67.5**%, well above AttnLRP (52.8%), IntGrad (52.4%), SmoothGrad (52.0%), LeGrad (47.3%), and Chefer-LRP (46.3%). This indicates that DAVE's attributions are not only quantitatively faithful but also more interpretable and class-discriminative to human evaluators.

**Qualitative comparisons.** Figures 7 and 8 provide qualitative comparisons on ImageNet-1k validation examples. In Figure 8, several prior methods exhibit patch-grid artifacts and more diffuse explanations, particularly SmoothGrad (col. 3), IntGrad (col. 4), and transformer-specific baselines (cols. 5-6). In contrast, DAVE (col. 2) produces sharper, more object-aligned attribution maps with reduced noise, consistent with the localization gains in Tables 1, and 2.

For B-cos ViTs (Figure 7) trained without a convolutional stem, the inherent B-cos explanations are comparatively noisy, whereas DAVE yields more fine-grained and object-specific attributions.

For additional qualitative results on additional models, class-consistency and varied classes see Appendix D.

### 5.2. Ablation Studies and Analysis

In this section, we analyze the contribution of key components of DAVE and study their effect on attribution quality, stability, and computational behavior. Extended analyses and additional results are provided in Appendix C.2.

*Table 3.* **Effect of removing operator variation.** GridPG (%) and EnergyPG (EPG, %) on ImageNet-1k for DeiT-III-B/16 and ViT-B/16. "DAVE (with operator variation)" applies DAVE's averaging to the raw gradient (augmented SmoothGrad).

| | DeiT-III-B/16 | | ViT-B/16 | |
|---|---|---|---|---|
| **Method** | **GridPG ↑** | **EPG ↑** | **GridPG ↑** | **EPG ↑** |
| DAVE (with operator variation) | 43.67 | 79.80 | 46.64 | 77.31 |
| **DAVE (ours)** | **65.29** | **83.01** | **58.80** | **78.66** |
| Δ | +21.63 | +3.20 | +12.15 | +1.36 |

**Isolating the effective transformation.** Table 3 isolates the effect of the operator-variation term from Eq. (3). Reintroducing this term inside the DAVE pipeline (DAVE with operator variation) is equivalent to applying DAVE's neighborhood averaging directly to the raw input gradient, i.e., an *augmented SmoothGrad* baseline. This consistently reduces localization, with GridPG drops of 21.63 pp on DeiT-III-B/16 and 12.15 pp on ViT-B/16. Qualitatively (Appendix D.6), operator variation introduces noisier and less consistent responses, whereas DAVE produces clearer, more

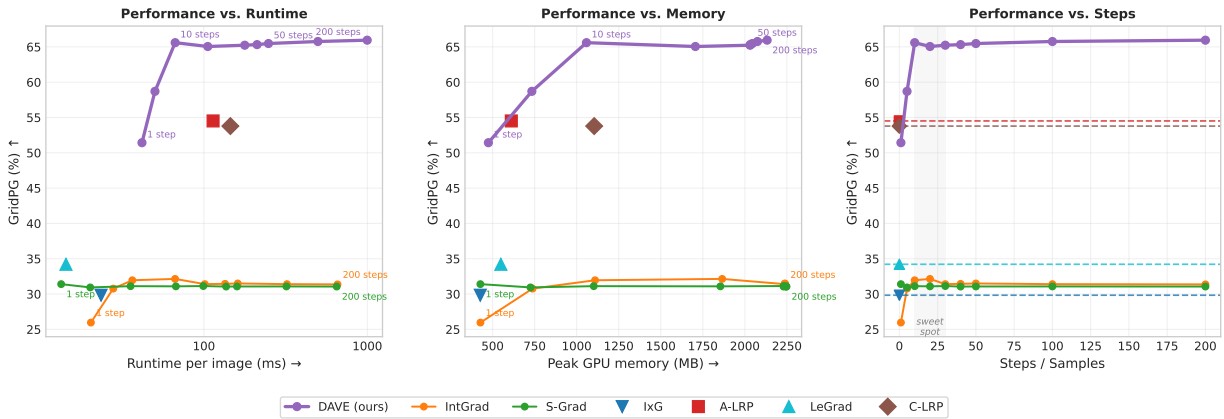

*Figure 9.* **Performance–efficiency tradeoff for attribution methods on DeiT-III-B/16** (NVIDIA L40S, 46 GB VRAM). GridPG (%) vs. runtime (*left*), peak GPU memory (*centre*), and number of steps (*right*). Single-pass methods are shown as markers, multi-step methods as curves (steps $\in \{1, 5, 10, 20, 30, 40, 50, 100, 200\}$, fixed internal batch size 25).

class-consistent patterns. This supports that DAVE's gains stem from isolating the effective transformation rather than from averaging alone.

**Transformation sensitivity.** We define a stable attribution neighborhood using rotations within $(-20°, 20°)$, horizontal flips with probability $p = 0.5$, horizontal and vertical shifts up to $0.1$ of the image size, and model-specific noise schedules. These choices are motivated by the observed sensitivity of the models to such transformations, measured as the absolute change in softmax probabilities (see Appendix C.2). Additional analysis of transformation ranges, input perturbations, and setup choices is provided in Appendix B.1 and C.2.

*Table 4.* **DAVE transformations ablation (GridPG %).** Each transformation is evaluated in isolation (top) and cumulatively (bottom) on DeiT-B/16 and DeiT-III-B/16.

| Config. | HFlip | Rotate | Translate | Noise | GridPG (%) DeiT | GridPG (%) DeiT-III |
|---|---|---|---|---|---|---|
| No aug. | | | | | 53.0 | 52.4 |
| HFlip | ✓ | | | | 52.0 | 52.1 |
| Rotate | | ✓ | | | 62.7 | 65.2 |
| Translate | | | ✓ | | 63.8 | 66.4 |
| Noise | | | | ✓ | 62.1 | 64.7 |
| HFlip | ✓ | | | | 51.8 | 52.8 |
| + Rot. | ✓ | ✓ | | | 63.4 | 65.9 |
| + Trans. | ✓ | ✓ | ✓ | | 64.0 | 67.1 |
| + Noise (DAVE) | ✓ | ✓ | ✓ | ✓ | **65.8** | **68.4** |

**Transformations ablation.** Table 4 analyzes the contribution of individual augmentation components used in DAVE. Geometric transformations, particularly rotation and translation, provide the largest gains, improving GridPG by 10-14 pp over the non-augmented baseline. In contrast, horizontal flipping alone has minimal impact. Adding progressive noise on top of geometric transformations further improves

performance by approximately 1-2 pp, indicating that geometric and stochastic perturbations provide complementary sources of stabilization.

**Computational efficiency.** We benchmark runtime and memory against all baselines on DeiT-III-B/16 (NVIDIA L40S). As shown in Figure 9, DAVE reaches $65.6\%$ GridPG at $10$ samples in only $66.7$ ms with $\sim 1.0$ GB peak memory, and $65.7\%$ at $50$ samples ($248.3$ ms, $\sim 2.0$ GB), surpassing all single-pass baselines (e.g., A-LRP $54.5\%$ at $113.8$ ms, C-LRP $53.8\%$ at $145.1$ ms) by a large margin while remaining near the practical Pareto frontier. The asymptotic complexity analysis and full tables are provided in Appendix C.2.

**Further analyses.** We provide additional analyses in the appendix: behavior under misclassification and across confusable classes (Appendix D.5), robustness to distribution shift (Appendix D.8), evaluation on DINOv2 with register tokens against a PCA baseline (Appendix C.1), and an extension to convolutional backbones (Appendix D.10). We discuss limitations in Appendix F.

## 6. Conclusions

In this work, we introduced DAVE, a mathematically grounded framework designed to overcome the challenges of stability and resolution in ViT explainability. By performing a structured decomposition of the input gradient, DAVE isolates the effective transformation of the model from architecture-induced artifacts and input-dependent noise. Across localization, faithfulness, and human user studies, DAVE consistently outperforms existing state-of-the-art methods, producing spatially precise and class-consistent attributions. DAVE is also architecture-versatile, providing robust explanations for standard ViTs, self-supervised models, inherently interpretable B-cos models, and convolutional backbones such as ConvNeXt, while remaining robust across pretraining paradigms.

## Acknowledgments

The work was funded by the "Interpretable and Interactive Multimodal Retrieval in Drug Discovery" project. The "Interpretable and Interactive Multimodal Retrieval in Drug Discovery" project (FENG.02.02-IP.05-0040/23) is carried out within the First Team programme of the Foundation for Polish Science co-financed by the European Union under the European Funds for Smart Economy 2021-2027 (FENG).

Experiments were performed on servers that were funded by National Centre of Science (Poland) grant no. 2023/49/B/ST6/01137 and the flagship project entitled "Artificial Intelligence Computing Center Core Facility" from the Digi-World Priority Research Area within the Excellence Initiative – Research University program at Jagiellonian University in Krakow.

We gratefully acknowledge Polish high-performance computing infrastructure PLGrid (HPC Center: ACK Cyfronet AGH) for providing computer facilities and support within computational grant no. PLG/2025/018158

We would also like to thank Bartłomiej Pogodziński, Raphael Maser, Anurag Das and Amin Parchami for proofreading the manuscript and providing constructive feedback on the writing.

## Impact Statement

This research contributes to the broader field of Explainable AI (XAI) by providing a more reliable mechanism for auditing high-capacity vision models. As ViTs are increasingly integrated into high-stakes applications such as medical diagnostics and autonomous driving, the ability to verify that a model is attending to relevant visual features rather than architectural artifacts is critical for safety and trust.

By producing stable, high-resolution attribution maps, DAVE can assist practitioners in (1) Identifying Model Bias (2) Model Debugging by providing visual evidence to understand why a model might misclassify specific samples. (3) Enhancing Human-AI Collaboration by offering explainable signals that align more closely with human visual perception, thereby facilitating better decision-making in collaborative environments.

Lastly, this work supports the development of more transparent and accountable machine learning systems, reducing the opaque nature of state-of-the-art computer vision models.

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

# Appendix

In this appendix, we provide additional material supporting the main paper. We begin with the mathematical derivations underlying DAVE, including the unified layer-wise effective-transformation formulation, the equivariant averaging operator and its non-expansiveness properties, and the interpretation of stochastic averaging as Gaussian convolution (Appendix A.1–A.4). We then detail our implementation and evaluation setup, covering the transformation neighborhood and noise schedule, pretrained checkpoints, linear-probing protocols for self-supervised models, the GridPG and EnergyPG localization metrics, the pixel-deletion faithfulness protocol, and pseudocode for the conditioned forward pass and DAVE attribution (Appendix B). We next report extended quantitative results, including localization on DINOv2 with register tokens against PCA-based patch-feature baselines (Appendix C.1), an ablation and analysis of DAVE's components covering its performance–efficiency tradeoff, sample convergence, asymptotic complexity, and sensitivity to rotations, additive noise, and horizontal translations (Appendix C.2), extended pixel-deletion and pixel-insertion faithfulness curves, and two complementary user studies (Appendix C.4). We then provide qualitative analyses across architectures and settings (Appendix D), including comparisons across ViT backbones, B-cos models, behavior under misclassification and across confusable classes (Appendix D.5), the role of the operator-variation term (Appendix D.6), robustness on ImageNet-Rendition (Appendix D.8), DINOv2 with register tokens (Appendix D.9), and an extension to convolutional backbones via ConvNeXt-S (Appendix D.10). We conclude with a completeness discussion (Appendix E) and an outline of limitations and future directions (Appendix F).

# A. Mathematical derivations

## A.1. Layers Representations

In this section, we represent Vision Transformer layers according to the unified layer formulation introduced in Section 3.1. We adopt the layer definition (Equation 1) together with the two operator forms (Equation 2), and show how we represent principal components of Vision Transformer (ViT) architectures within this framework. Specifically, we consider fully-connected layers, self-attention and multi-head self-attention, layer normalization, pointwise nonlinearities (GELU and Swish), and residual connections.

In all derivations, the input to a layer is denoted by $\boldsymbol{X} \in \mathbb{R}^{t \times d_{\mathrm{in}}}$, representing a sequence of length $t$ with $d_{\mathrm{in}}$-dimensional token embeddings.

### A.1.1. FULLY-CONNECTED

A fully-connected layer acting independently on each token can be represented within the operator form (I) of Equation 2 by defining the operator-valued map

$$L(\boldsymbol{X})(\boldsymbol{X}) = \boldsymbol{I}_t \, \boldsymbol{X} \, \boldsymbol{W}_d,$$

where $\boldsymbol{I}_t \in \mathbb{R}^{t \times t}$ is the identity matrix over tokens and $\boldsymbol{W}_d \in \mathbb{R}^{d_{\mathrm{in}} \times d_{\mathrm{out}}}$ is the learned feature projection. Substituting this expression into the layer definition (Equation 1) yields the standard fully-connected transformation

$$F(\boldsymbol{X}) = \boldsymbol{X}\boldsymbol{W}_d + \boldsymbol{B}.$$

In this case, the operator $L(\boldsymbol{X})$ is input-independent, and the corresponding operator-variation term in the layer derivative is identically zero.

### A.1.2. SELF-ATTENTION

We consider a single-head self-attention layer. Let $\boldsymbol{W}_Q, \boldsymbol{W}_K, \boldsymbol{W}_V \in \mathbb{R}^{d_{\mathrm{in}} \times d_h}$ denote the query, key, and value projections, and let $\boldsymbol{W}_O \in \mathbb{R}^{d_h \times d_{\mathrm{out}}}$ denote the output projection. Given an input $\boldsymbol{X} \in \mathbb{R}^{t \times d_{\mathrm{in}}}$, the attention weights are computed as

$$\boldsymbol{A}(\boldsymbol{X}) = \mathrm{softmax}\left( \frac{\boldsymbol{X}\boldsymbol{W}_Q(\boldsymbol{X}\boldsymbol{W}_K)^\top}{\sqrt{d_h}} \right) \in \mathbb{R}^{t \times t},$$

with the softmax applied row-wise.

This operation admits a representation of the form (I) in Equation 2 by defining the operator-valued map

$$L(\boldsymbol{X})(\boldsymbol{X}) = \boldsymbol{W}_t(\boldsymbol{X}) \, \boldsymbol{X} \, \boldsymbol{W}_d,$$

where $\boldsymbol{W}_t(\boldsymbol{X}) = \boldsymbol{A}(\boldsymbol{X})$ is the input-dependent token-mixing matrix and $\boldsymbol{W}_d = \boldsymbol{W}_V\boldsymbol{W}_O$ is the composed feature projection. Substituting this operator into the layer definition (Equation 1) yields the self-attention transformation

$$F(\boldsymbol{X}) = \boldsymbol{A}(\boldsymbol{X}) \, \boldsymbol{X}\boldsymbol{W}_V\boldsymbol{W}_O + \boldsymbol{B},$$

where $\boldsymbol{B} \in \mathbb{R}^{t \times d_{\mathrm{out}}}$ is the bias associated with the output projection. If the value projection includes an additive bias, it can be interpreted as a preceding fully-connected layer and absorbed into the formulation accordingly.

### A.1.3. MULTI-HEAD SELF-ATTENTION

Multi-head self-attention extends the single-head formulation by computing $H$ attention heads in parallel, each operating on a distinct feature subspace. For head $h \in \{1, \ldots, H\}$, let $\boldsymbol{A}^{(h)}(\boldsymbol{X}) \in \mathbb{R}^{t \times t}$ denote the corresponding attention matrix. Within the operator form (I) of Equation 2, the resulting token-mixing operator can be expressed as

$$\boldsymbol{W}_t(\boldsymbol{X}) = \sum_{h=1}^{H} \boldsymbol{A}^{(h)}(\boldsymbol{X}) \otimes \boldsymbol{P}_h,$$

where $\boldsymbol{P}_h$ denotes the projection onto the feature subspace associated with head $h$, and $\otimes$ denotes the Kronecker product. The feature projection $\boldsymbol{W}_d$ accounts for the head-wise value projections, concatenation of head outputs, and the subsequent output projection. Substituting these definitions into the layer formulation (Equation 1) recovers the standard multi-head self-attention operation.

### A.1.4. LAYER NORMALIZATION

We consider Layer Normalization applied independently to each token. Given an input $\boldsymbol{X} \in \mathbb{R}^{t \times d_{\mathrm{in}}}$, let $\boldsymbol{\mu}(\boldsymbol{X}) \in \mathbb{R}^{t \times 1}$ and $\boldsymbol{\sigma}(\boldsymbol{X}) \in \mathbb{R}^{t \times 1}$ denote the per-token mean and standard deviation across the feature dimension. Layer normalization acts as

$$\mathrm{LN}(\boldsymbol{X}) = \frac{\boldsymbol{X} - \boldsymbol{\mu}(\boldsymbol{X})\mathbf{1}^\top}{\boldsymbol{\sigma}(\boldsymbol{X})} \odot \boldsymbol{\gamma} + \boldsymbol{\beta},$$

where $\boldsymbol{\gamma}, \boldsymbol{\beta} \in \mathbb{R}^{d_{\mathrm{in}}}$ are learned affine parameters.

We express this operation within the operator form (I) of Equation 2. Mean subtraction corresponds to multiplication by a fixed centering matrix

$$\boldsymbol{W}_d = \boldsymbol{I}_{d_{\mathrm{in}}} - \frac{1}{d_{\mathrm{in}}}\mathbf{1}\mathbf{1}^\top \in \mathbb{R}^{d_{\mathrm{in}} \times d_{\mathrm{in}}},$$

which defines the feature-space component of the operator. The variance normalization induces an input-dependent, token-wise scaling, captured by a diagonal matrix

$$\boldsymbol{W}_t(\boldsymbol{X}) = \mathrm{diag}\big(\boldsymbol{\sigma}(\boldsymbol{X})^{-1}\big) \in \mathbb{R}^{t \times t}.$$

Accordingly, we formulate operator-valued map for normalization as:

$$L(\boldsymbol{X})(\boldsymbol{X}) = \boldsymbol{W}_t(\boldsymbol{X})\,\boldsymbol{X}\,\boldsymbol{W}_d$$

The learned affine parameters $(\boldsymbol{\gamma}, \boldsymbol{\beta})$ are treated as a subsequent fully-connected layer. Substituting this operator into the layer definition (Equation 1), and composing with the affine map, recovers the standard Layer Normalization transformation.

### A.1.5. GELU / SWISH

We express pointwise nonlinearities used in Vision Transformers, such as GELU and Swish, within the gating form (II) of Equation 2. Consider a scalar nonlinearity $f : \mathbb{R} \to \mathbb{R}$ of the form

$$f(x) = x\,\phi(x),$$

where $\phi : \mathbb{R} \to \mathbb{R}$ is a smooth gating function. For GELU, $\phi(x)$ is the cumulative distribution function of the standard normal distribution, while for Swish, $\phi(x)$ is the sigmoid function.

Applied elementwise to an input $\boldsymbol{X} \in \mathbb{R}^{t \times d}$, this nonlinearity can be written as

$$F(\boldsymbol{X}) = \boldsymbol{X} \odot \boldsymbol{\Phi}(\boldsymbol{X}),$$

where $\boldsymbol{\Phi}(\boldsymbol{X})$ denotes the elementwise application of $\phi$. Accordingly, the operator-valued map is given by

$$L(\boldsymbol{X})(\boldsymbol{X}) = \boldsymbol{\Phi}(\boldsymbol{X}) \odot \boldsymbol{X},$$

which corresponds directly to form (II) in Equation 2. Substituting this expression into the layer definition (Equation 1) recovers the standard GELU and Swish activations.

### A.1.6. RESIDUAL CONNECTIONS

Residual connections combine the input with the output of a transformation branch. Let $H : V_{\mathrm{in}} \to V_{\mathrm{in}}$ denote a residual branch that admits a representation consistent with our framework, i.e.,

$$H(\boldsymbol{X}) = L_H(\boldsymbol{X})(\boldsymbol{X}) + \boldsymbol{B}_H,$$

or a composition of such transformations. The corresponding residual layer is given by

$$F(\boldsymbol{X}) = \boldsymbol{X} + H(\boldsymbol{X}).$$

Substituting the representation of $H$ yields

$$F(\boldsymbol{X}) = \big(\boldsymbol{I} + L_H(\boldsymbol{X})\big)(\boldsymbol{X}) + \boldsymbol{B}_H,$$

which shows that residual connections extend the effective transformation by adding the identity operator. Accordingly, the residual layer admits an effective operator

$$L_F(\boldsymbol{X}) = \boldsymbol{I} + L_H(\boldsymbol{X})$$

## A.2. Effective Transformation Matrix

In this section, we show that for both operator forms (I) and (II) introduced in Equation 2, the effective transformation $L(\boldsymbol{X})$ admits an input-dependent matrix representation acting on the vectorized layer input.

**Matrix representation.** Let $V_{\text{in}}$ and $V_{\text{out}}$ denote the input and output vector spaces of a layer, and fix bases $\mathcal{B}_{\text{in}}$ and $\mathcal{B}_{\text{out}}$ for these spaces. For any operator-valued map $L(\boldsymbol{X}) \in \mathcal{L}(V_{\text{in}}, V_{\text{out}})$, the linear operator $L(\boldsymbol{X})$ admits a (generally input-dependent) matrix representation $\boldsymbol{W}_L(\boldsymbol{X})$ with respect to these bases, such that

$$[L(\boldsymbol{X})(\boldsymbol{X})]_{\mathcal{B}_{\text{out}}} = \boldsymbol{W}_L(\boldsymbol{X}) \, [\boldsymbol{X}]_{\mathcal{B}_{\text{in}}}.$$

**ViT vector spaces.** In the case of Vision Transformer layers, we identify $V_{\text{in}} \cong \mathbb{R}^{t \times d_{\text{in}}}$ and $V_{\text{out}} \cong \mathbb{R}^{t \times d_{\text{out}}}$, and represent these spaces via vectorization. We take $\mathcal{B}_{\text{in}}$ and $\mathcal{B}_{\text{out}}$ to be the standard bases of $\mathbb{R}^{td_{\text{in}}}$ and $\mathbb{R}^{td_{\text{out}}}$, respectively, so that:

$$[L(\boldsymbol{X})(\boldsymbol{X})]_{\mathcal{B}_{\text{out}}} = \text{vec}(L(\boldsymbol{X})(\boldsymbol{X})).$$

**Form (I).** For operator form (I) (see Equation 2), by applying the vectorization operator, we obtain:

$$\text{vec}(L(\boldsymbol{X})(\boldsymbol{X})) = \left( \boldsymbol{W}_d^\top \otimes \boldsymbol{W}_t(\boldsymbol{X}) \right) \text{vec}(\boldsymbol{X}).$$

where $\otimes$ denotes a Kronecker product. Thus, under the bases induced by vectorization, the effective transformation admits the matrix representation:

$$\boldsymbol{W}_L(\boldsymbol{X}) = \boldsymbol{W}_d^\top \otimes \boldsymbol{W}_t(\boldsymbol{X}) \in \mathbb{R}^{(td_{\text{out}}) \times (td_{\text{in}})}$$

In particular, for self-attention layers, $\boldsymbol{W}_t(\boldsymbol{X})$ is given by the attention matrix and $\boldsymbol{W}_d$ encodes the value and output projections, yielding a Kronecker-structured effective weight matrix.

**Form (II)** For operator form (II) (see Equation 2), vectorizing the elementwise product yields

$$\text{vec}(\boldsymbol{\Phi}(\boldsymbol{X}) \odot \boldsymbol{X}) = \text{diag}(\text{vec}(\boldsymbol{\Phi}(\boldsymbol{X}))) \, \text{vec}(\boldsymbol{X}),$$

where $\text{diag}(\cdot)$ denotes the diagonal matrix formed from its vector argument. Hence the effective transformation admits the matrix representation

$$\boldsymbol{W}_L(\boldsymbol{X}) = \text{diag}(\text{vec}(\boldsymbol{\Phi}(\boldsymbol{X}))) \in \mathbb{R}^{(td_{\text{in}}) \times (td_{\text{in}})},$$

## A.3. Equivariant Transformation

In this section, we show that the proposed Reynolds-inspired operator preserves equivariant attribution components and suppresses (is non-expansive) on locally non-equivariant components.

### A.3.1. REYNOLDS OPERATOR

Given a group action on a vector space, the Reynolds operator provides a principled way to decompose each vector into components that are invariant and non-invariant under the action.

**Definition.** Let $G$ be a compact group acting linearly on a real vector space $V$ via a representation $\rho : G \to \mathcal{L}(V, V)$. We assume that $G$ is equipped with a normalized Haar (uniform) measure $\mu$. The Reynolds operator associated with this group action is defined as the linear map $\mathcal{R} : V \to V$ given by:

$$\mathcal{R}(\boldsymbol{x}) := \int_G \rho(g)(\boldsymbol{x}) \, d\mu(g) \tag{11}$$

**Projection onto invariant subspace.** Defining the subspace of $G$-invariant elements as:

$$V^G := \{\boldsymbol{x} \in V \mid \rho(g)(\boldsymbol{x}) = \boldsymbol{x} \ \forall g \in G\} \tag{12}$$

the Reynolds operator $\mathcal{R} : V \to V$ satisfies $\text{Im}(\mathcal{R}) = V^G$ and $\mathcal{R} \circ \mathcal{R} = \mathcal{R}$, and therefore is a projection onto $V^G$.

**Vector space decomposition.** As a consequence of projection from the previous paragraph, the vector space $V$ decomposes as a direct sum:

$$V = V^G \oplus \ker(\mathcal{R}) \tag{13}$$

Which decomposes each $\boldsymbol{x} \in V$ to $\boldsymbol{x} = \mathcal{R}(\boldsymbol{x}) + (\boldsymbol{x} - \mathcal{R}(\boldsymbol{x}))$. The component $\mathcal{R}(\boldsymbol{x}) \in V^G$ captures the $G$-invariant part of $\boldsymbol{x}$, and $(\boldsymbol{x} - \mathcal{R}(\boldsymbol{x})) \in \ker(\mathcal{R})$ contains all components that are not invariant under the group action.

### A.3.2. SUPPRESSION OF NON-EQUIVARIANT COMPONENTS.

In this section we show that the Reynolds-inspired operator defined in Eq. (6) suppresses (i.e., is non-expansive on) non-equivariant attribution components.

Let $V$ denote the Hilbert space of attribution maps equipped with the inner product induced by the spatial domain, and let $V^G \subset V$ be the subspace of $G$-equivariant elements under the action

$$(\tau \cdot \boldsymbol{W}_L)(\boldsymbol{X}) := [\tau^{-1} \circ \boldsymbol{W}_L \circ \tau](\boldsymbol{X}).$$

According to Equation 13, the ideal Reynolds operator associated with this action induces an orthogonal decomposition

$$\boldsymbol{W}_L = \boldsymbol{W}_L^G + \boldsymbol{W}_L^\perp,$$

where $\boldsymbol{W}_L^G \in V^G$ and $\boldsymbol{W}_L^\perp \in (V^G)^\perp$.

Assuming that each spatial transformation $\tau$ acts unitarily on $V$ (i.e., preserves the inner product on the spatial domain), the averaging operator from Eq. (7) is non-expansive:

$$\|\boldsymbol{W}_L^{\text{eq}}\| \leq \|\boldsymbol{W}_L\|.$$

Moreover, equivariant components are preserved exactly: if $\boldsymbol{W}_L \in V^G$, then $\boldsymbol{W}_L^{\text{eq}} = \boldsymbol{W}_L$.

Thus, while the proposed operator does not define an exact Reynolds projection (unless $\nu$ is the Haar measure), it preserves the equivariant component of the attribution map and is non-expansive on the non-equivariant component.

### A.4. Stochastic Averaging as Convolution

In this section, we show that the local averaging operation used in Section 3.3 is equivalent to a convolution with the noise distribution. Let $\boldsymbol{X} \in \mathbb{R}^d$ denote the (vectorized) layer input and let $\boldsymbol{W}_L^{\text{eq}} : \mathbb{R}^d \to \mathbb{R}^m$ denote the equivariant effective transformation (the argument below applies componentwise when $m > 1$).

Let $\boldsymbol{\epsilon}$ be a random variable in $\mathbb{R}^d$ with probability density $\mathcal{K} : \mathbb{R}^d \to \mathbb{R}_{\geq 0}$, $\int_{\mathbb{R}^d} \mathcal{K}(\boldsymbol{z}) \, \mathrm{d}\boldsymbol{z} = 1$. The low-pass filtered equivariant effective transformation is defined by stochastic averaging,

$$\tilde{\boldsymbol{W}}_L^{\text{eq}}(\boldsymbol{X}) = \mathbb{E}_{\boldsymbol{\epsilon}}\left[\boldsymbol{W}_L^{\text{eq}}(\boldsymbol{X} + \boldsymbol{\epsilon})\right] = \int_{\mathbb{R}^d} \boldsymbol{W}_L^{\text{eq}}(\boldsymbol{X} + \boldsymbol{\epsilon}) \, \mathcal{K}(\boldsymbol{\epsilon}) \, \mathrm{d}\boldsymbol{\epsilon}.$$

We now rewrite this as a convolution via a change of variables. Setting $\boldsymbol{u} = \boldsymbol{X} + \boldsymbol{\epsilon}$, $\boldsymbol{\epsilon} = \boldsymbol{u} - \boldsymbol{X}$, $\mathrm{d}\boldsymbol{\epsilon} = \mathrm{d}\boldsymbol{u}$, to obtain:

$$\tilde{\boldsymbol{W}}_L^{\text{eq}}(\boldsymbol{X}) = \int_{\mathbb{R}^d} \boldsymbol{W}_L^{\text{eq}}(\boldsymbol{u}) \, \mathcal{K}(\boldsymbol{u} - \boldsymbol{X}) \, \mathrm{d}\boldsymbol{u}.$$

Which is the convolution of $\boldsymbol{W}_L^{\text{eq}}$ with the kernel $\mathcal{K}$:

$$\tilde{\boldsymbol{W}}_L^{\text{eq}}(\boldsymbol{X}) = (\boldsymbol{W}_L^{\text{eq}} * \mathcal{K})(\boldsymbol{X}).$$

In the setting of Section 3.3, $\mathcal{K}$ is chosen to be the density of a zero-mean Gaussian distribution $\mathcal{N}(0, \Sigma)$, which corresponds to convolution with a Gaussian kernel and yields the low-pass filtering effect described in the main text.

# B. Additional Implementation Details

In this appendix, we provide additional implementation details that supplement the experimental setup described in the main paper. We first describe DAVE's transformation neighborhood and noise-injection scheme used throughout our experiments (Appendix B.1). We then detail the evaluation pipeline (Appendix B.2), including the pretrained checkpoints used for each backbone, the linear-probing protocol for self-supervised models (DINO and DINOv2), the GridPG and EnergyPG localization metrics, the pixel-deletion faithfulness protocol, attribution post-processing, and pseudocode for the conditioned forward pass and DAVE attribution computation (Algorithms 1–2).

## B.1. DAVE Setup

**Transformations.**  For spatial transformations, we apply random horizontal flips with probability 0.5, small in-plane rotations uniformly sampled from $(-20°, 20°)$, and small wrapped translations with offsets up to 0.1 of the image extent along both spatial axes.

**Noise injection.**  In Section 3.3, we introduce low-pass filtering via expectation under Gaussian perturbations, corresponding to single-scale Gaussian smoothing. While this is sufficient for B-cos models (using additive noise with $\sigma = 0.2$), we empirically find that conventional architectures benefit from a more robust variant.

Specifically, we employ a variance-preserving noise interpolation scheme: for each sample we draw $\epsilon \sim \mathcal{N}(0, I)$ and sample $t \in [0, 0.5]$, constructing

$$\boldsymbol{x}_t = (1 - t)\boldsymbol{x} + \sqrt{1 - (1 - t)^2}\,\boldsymbol{\epsilon}.$$

Sampling $t$ effectively averages responses across noise scales, yielding a multi-scale generalization of the low-pass filter, which we find more robust for conventional models.

## B.2. Evaluation Details

**Models and checkpoints.**  All backbones are evaluated using the authors' publicly released *pretrained* checkpoints from their official repositories. Unless stated otherwise, we do not fine-tune model weights for attribution evaluation; DAVE and all baselines are computed on frozen models under the same preprocessing and input resolution ($224 \times 224$).

**Baseline implementations.**  We follow the authors' official implementations and default hyperparameters for all attribution baselines. SmoothGrad (Smilkov et al., 2017) uses 25 noise samples and Integrated Gradients (Sundararajan et al., 2017) uses 200 interpolation steps; in both cases we empirically verified that increasing the number of steps does not improve the resulting localization or faithfulness metrics (see Table 6). AttnLRP (Achtibat et al., 2024), Chefer-LRP (Chefer et al., 2021), and LeGrad (Bousselham et al., 2025) are single-step methods with no comparable hyperparameter to tune.

**DINO and DINOv2 linear probing.**  For both the self-supervised DINO ViT-B/16 (Caron et al., 2021) and the DINOv2 ViT-B/14 backbone with register tokens (reg4) (Darcet et al., 2024), we follow the standard linear-probing protocol on ImageNet-1k: we freeze the pretrained backbone and train a linear classifier on top of the [CLS] representation using ImageNet-1k train labels, and evaluate on the ImageNet-1k validation set. We use a single linear layer trained with cross-entropy, SGD with momentum (0.9), and cosine learning-rate schedule (0.001) for 100 epochs with batch size 128. We use standard data augmentation for linear probing (random resized crop and horizontal flip) and otherwise default training settings from the respective authors' reference implementations (Caron et al., 2021).

All attribution methods are then computed with respect to the resulting linear-probe classifier while keeping the backbone frozen.

**Localization metrics.**  For GridPG, we follow the popular protocol (Böhle et al., 2021; Zhang et al., 2018): we construct $500\ 2 \times 2$ grids from ImageNet-1k validation images of *distinct* classes that are correctly classified with high confidence.[3] For each target class $i$, we compute the fraction of *positive* attribution mass assigned to its corresponding grid cell. Let $A(p)$ be the attribution value at pixel $p$ and $A^+(p) = \max(A(p), 0)$ its positive part. The per-cell score is

$$L_i = \frac{\sum_{p \in \text{cell}_i} A^+(p)}{\sum_{j=1}^{4} \sum_{p \in \text{cell}_j} A^+(p)}, \tag{14}$$

---

[3]We use the same grid construction procedure across all attribution methods and models.

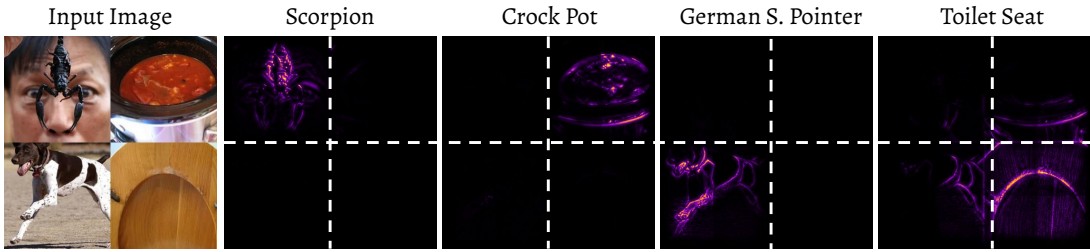

| Input Image | Scorpion | Crock Pot | German S. Pointer | Toilet Seat |

*Figure 10.* **GridPG Illustration.** 2x2 cells.

and the GridPG score is the average of $L_i$ over all constructed grids (see Figure 10 for illustration).

Note: Samples with no-positive attributions in the entire grid are dropped from evaluation as is commonly done (Böhle et al., 2021).

For the EnergyPG, we follow the common practice (Wang et al., 2020) and evaluate on the ImageNet-1k validation set using the official ILSVRC (Russakovsky et al., 2015) bounding-box annotations. Let $\Omega$ denote the ground-truth bounding box region for the image label (union if multiple boxes are provided) and $\bar{\Omega}$ its complement (see Figure 11 for illustration). We report

$$\text{EnergyPG} = \frac{\sum_{p \in \Omega} A^+(p)}{\sum_{p \in \Omega \cup \bar{\Omega}} A^+(p)}. \tag{15}$$

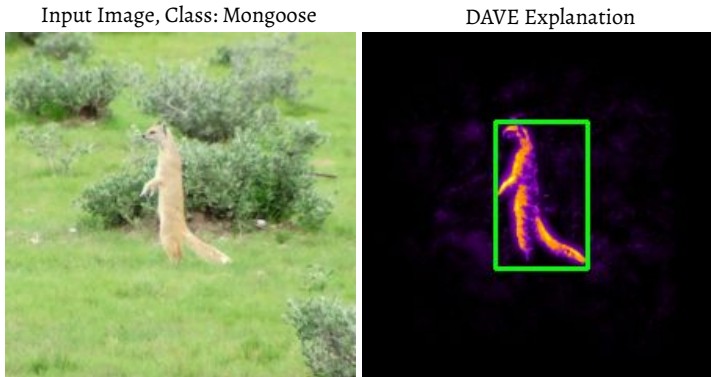

| Input Image, Class: Mongoose | DAVE Explanation |

*Figure 11.* **EnergyPG Illustration.**

**Faithfulness via pixel deletion.** We evaluate faithfulness with *pixel deletion* curves following prior work (Chefer et al., 2021; Böhle et al., 2021). Given an attribution map, we rank pixels in *increasing* order of attribution (least to most important) and progressively set the lowest-ranked pixels to zero. After each deletion step, we forward the modified image through the model and record the target-class probability. We plot the target probability versus the fraction of pixels removed.

The *pixel addition* setup is complementary to this where we start a heavily blurred out version of the original image, and replace most important pixels to least important pixels (in that order) from the original image to our initial blurred image.

**Protocol details.** GridPG is evaluated on 500 constructed grids (4000 ImageNet-1k validation images), EnergyPG uses ILSVRC (Russakovsky et al., 2015) bounding boxes (50k validation images), and pixel deletion uses a random 5k subset, following standard protocols ((Chefer et al., 2021; Böhle et al., 2021)).

**Attribution post-processing.** Unless stated otherwise, we evaluate attributions at the model input resolution. For methods that produce patch-level maps, we upsample to pixel space using bilinear interpolation. For all metrics, we use the positive attribution mass $A^+(p)$ as defined above.

**Algorithm.** We preserve the standard forward pass while enabling efficient evaluation of the effective transformation via a modified gradient computation. Specifically, we reimplement each layer $F_i$ of the ViT according to Equation 1. For

each operator-valued map $L$ within a layer, we introduce a separate conditioning variable $c_i$ that parameterizes the operator as $L(c_i)$, while the input variable $x_i$ is used to evaluate $L(c_i)(x_i)$. Although $c_i$ and $x_i$ take identical values during the forward pass (see Algorithm 1), gradients through $c_i$ are blocked during backpropagation. This construction leaves the forward computation unchanged while allowing gradients with respect to the input $x$ to ignore operator variation, enabling efficient computation of the effective transformation.

Using the conditioned forward pass, we compute DAVE attributions as described in Algorithm 2. At each step, we sample a spatial transformation and input noise to generate a perturbed input. We then evaluate the class score using the conditioned forward pass and compute the effective transformation via a modified backward pass with respect to the input, which ignores the operator variation. The resulting effective transformation matrix is mapped back to the original input space by applying the inverse spatial transformation and accumulated across samples. Finally, the aggregated effective transformation is multiplied elementwise with the input, yielding the final DAVE attribution map.

---

**Algorithm 1** Conditioned forward pass

---

**Input:** input $x$; layers $\{F_i\}_{i=1}^n$
**Output:** logits $z$
$x_i \leftarrow x$
**for** $i = 1$ **to** $n$ **do**
   $c_i \leftarrow \mathrm{detach}(x_i)$
   $x_i \leftarrow F_i(\text{input} = x_i, \text{conditioning} = c_i)$
**end for**
$z \leftarrow x_i$
**return** $z$

---

**Algorithm 2** DAVE attribution

---

**Input:** data sample $x$, class label $y$, number of samples $T$
**Hyperparameters:** transformation distribution $\nu$, low-pass noise distribution $\mathcal{K}$
**Output:** attribution map $A$
$\tilde{W}_L^{\mathrm{eq}} \leftarrow 0$
**for** $t = 1$ **to** $T$ **do**
   $\tau \sim \nu, \quad \epsilon \sim \mathcal{K}$
   $x_t \leftarrow \tau(x + \epsilon)$
   $s_t \leftarrow \textsc{ConditionedForward}(x_t)[y]$
   $W_t \leftarrow \nabla_{x_t} s_t$ {ignores operator variation due to detach in conditioned forward pass}
   $\tilde{W}_L^{\mathrm{eq}} \leftarrow \tilde{W}_L^{\mathrm{eq}} + \tau^{-1}(W_t)$
**end for**
$A \leftarrow \left(\frac{1}{T}\tilde{W}_L^{\mathrm{eq}}\right) \odot x$
**return** $A$

---

# C. Quantitative Results

In this appendix, we report additional quantitative results complementing the main paper. These include localization on DINOv2 with register tokens against PCA-based baselines on the patch-token representations (Appendix C.1); an ablation and analysis of DAVE's components covering its performance–efficiency tradeoff, sample convergence, asymptotic complexity, and sensitivity to input rotations, additive noise, and horizontal translations (Appendix C.2); extended faithfulness curves via pixel deletion and pixel insertion; and two complementary user studies (Appendix C.4).

### C.1. DINOv2 with Register Tokens and PCA Baseline

We evaluate DAVE on DINOv2 ViT-B/14 with register tokens (Darcet et al., 2024) (reg4), which add learnable tokens that absorb high-norm outlier activations to reduce patch-grid artifacts in features. We compare DAVE against standard attribution baselines and against PCA-based baselines on the patch-token representations: *PCA (raw)* uses the top principal component of the raw patch tokens, while *PCA (aug)* aggregates tokens across the same 50-sample augmentation neighborhood used by DAVE before selecting the top component.

Table 5 reports localization on ImageNet-1k. Both PCA variants score at chance level (25% GridPG), confirming they lack class-discriminative localization; PCA (aug) improves only marginally over PCA (raw) on EnergyPG, indicating that augmentation alone does not yield class-specific attribution. DAVE clearly outperforms all baselines. Register tokens reduce artifacts but do not close the gap for standard attribution methods, confirming that DAVE's advantage stems from its attribution mechanism rather than from model-level artifact mitigation alone. The corresponding qualitative comparison appears in Appendix D.9.

*Table 5.* **Localization on DINOv2 ViT-B/14 with register tokens.** GridPG and EnergyPG (%) on ImageNet-1k using DINOv2 with register tokens. PCA (raw) uses the top-1 principal component of raw patch tokens; PCA (aug) aggregates tokens across 50 DAVE-style augmentation steps before selecting the top-1 component. Both PCA variants score at chance level (25%) on GridPG, confirming they lack class-discriminative localization. On EnergyPG, PCA (aug) improves only marginally over PCA (raw) (65.38% vs. 61.54%), indicating that DAVE-style augmentations alone do not yield class-specific attribution. DAVE outperforms all baselines on both metrics.

| Method | GridPG (%) DINO-B (reg4) | EnergyPG (%) DINO-B (reg4) |
|---|---|---|
| PCA (raw) | 25.00 | 61.54 |
| PCA (aug) | 25.00 | 65.38 |
| IntGrad | 33.79 | 59.78 |
| S-Grad | 33.08 | 61.46 |
| LeGrad | 29.92 | 73.72 |
| A-LRP | 37.97 | 66.94 |
| DAVE (ours) | **62.94** | **75.73** |
| $\Delta$ | +24.97 | +2.01 |

### C.2. Ablation and Analysis

We provide additional analysis supporting the design choices used in DAVE attribution. We examine: (i) the performance–efficiency tradeoff of DAVE relative to other attribution methods, (ii) the asymptotic complexity of DAVE relative to other methods, (iii) convergence of the iterative attribution estimates with respect to the number of neighborhood samples, and model sensitivity to (iv) input rotations, (v) additive input noise, and (vi) horizontal translations. Unless stated otherwise, analyses are conducted on ViT-B/16-224, DeiT-B/16-224, and DeiT-III-B/16-224, using 6,000 images randomly sampled from the ImageNet-1k validation set.

**Performance-efficiency tradeoff.** Table 6 compares attribution quality against computational cost. Single-pass methods provide low runtime and memory usage but achieve comparatively weaker localization performance, while multi-step methods exhibit diminishing returns beyond approximately 10 steps. DAVE lies near the practical Pareto frontier, consistently providing stronger localization than competing methods while maintaining modest computational requirements. In particular, DAVE requires only ∼1 GB peak memory at 10 steps and ∼2 GB at the 50-step configuration used throughout this work, corresponding to less than 5% of the available 46 GB GPU memory. *This analysis was performed on a random subset of 200 grid images (2 × 2 grids as constructed for GridPG) and then averaged.*

*Table 6.* **Attribution method benchmark on DeiT-III ViT-B/16** (NVIDIA L40S, 46 GB VRAM). Multi-step methods are reported at 5, 10, 20, and 50 steps; all DAVE experiments in this paper use 50 steps, which provides the best quality–efficiency balance (see Fig. 9). **Bold** indicates the best value per column. **DAVE (ours)** outperforms all baselines by a wide margin while using only ∼**2 GB of GPU memory** (< **5**% of available VRAM) for 50 steps which makes it practically useful even on smaller GPUs with less compute. *Note: For multi-step methods SmoothGrad, IntGrad and DAVE we use a fixed internal batch-size of 25 for fairness.*

| Method | Steps | GridPG (%)↑ | Time (ms)↓ | Mem (MB)↓ |
|---|---|---|---|---|
| *Single-pass* | | | | |
| IxG | – | 29.8 | 23.5 | **427** |
| LeGrad | – | 34.2 | **14.3** | 549 |
| A-LRP | – | 54.5 | 113.8 | 611 |
| C-LRP | – | 53.8 | 145.1 | 1104 |
| *Multi-step (5 steps)* | | | | |
| IntGrad | 5 | 30.8 | 27.9 | 738 |
| S-Grad | 5 | 30.9 | 20.2 | 725 |
| DAVE (ours) | 5 | 58.7 | 50.2 | 733 |
| *Multi-step (10 steps)* | | | | |
| IntGrad | 10 | 32.0 | 36.5 | 1109 |
| S-Grad | 10 | 31.1 | 35.6 | 1101 |
| DAVE (ours) | 10 | 65.6 | 66.7 | 1058 |
| *Multi-step (20 steps)* | | | | |
| IntGrad | 20 | 32.1 | 66.6 | 1865 |
| S-Grad | 20 | 31.1 | 67.4 | 1854 |
| DAVE (ours) | 20 | 65.1 | 105.6 | 1706 |
| *Multi-step (50 steps)* | | | | |
| IntGrad | 50 | 31.5 | 161.1 | 2238 |
| S-Grad | 50 | 31.1 | 159.4 | 2246 |
| DAVE (ours) | 50 | **65.7** | 248.3 | 2044 |

*Table 7.* **Computational and memory complexity of attribution methods relative to one forward pass.** For DAVE, Samples refers to a number of neighborhood samples.

| Method | Computation | Memory |
|---|---|---|
| I×G | $\mathcal{O}(1)$ | $\mathcal{O}(\sqrt{\text{Layers}})$ |
| IntGrad | $\mathcal{O}(\text{Steps})$ | $\mathcal{O}(\sqrt{\text{Layers}})$ |
| S-Grad | $\mathcal{O}(\text{Steps})$ | $\mathcal{O}(\sqrt{\text{Layers}})$ |
| LeGrad | $\mathcal{O}(\text{Steps})$ | $\mathcal{O}(\sqrt{\text{Layers}})$ |
| C-LRP | $\mathcal{O}(1)$ | $\mathcal{O}(\sqrt{\text{Layers}})$ |
| A-LRP | $\mathcal{O}(\text{Steps})$ | $\mathcal{O}(\sqrt{\text{Layers}})$ |
| DAVE (ours) | $\mathcal{O}(\text{Samples})$ | $\mathcal{O}(\sqrt{\text{Layers}})$ |

**Asymptotic complexity.** DAVE attribution has linear complexity in the number of neighborhood samples, i.e. $\mathcal{O}(\text{Samples})$, as described in Section 3.4. Table 7 compares the asymptotic cost of DAVE against other attribution methods.

**Convergence analysis.** We analyze convergence of DAVE by measuring the summed $L_1$ distance between cumulative average attributions at successive averaging steps, which quantifies the marginal contribution of additional neighborhood samples (see Figure 12). For each step, results are summarized using the median and interquartile range across images and visualized on a logarithmic scale. Across all considered models, the convergence measure decreases rapidly, reaching values on the order of $10^0$ after approximately 100 steps. In practice, we observe no significant improvement in the evaluation metrics used in the paper beyond 50 steps (see Figure 22 for qualitative results), and therefore adopt this setting in the main experiments.

**Convergence w.r.t. number of samples.** All experiments for DAVE use 50 samples, after which we observe no further improvement in evaluation metrics across models; attribution convergence analysis is provided in Main-Paper Figure 9. Also see Figure 12.

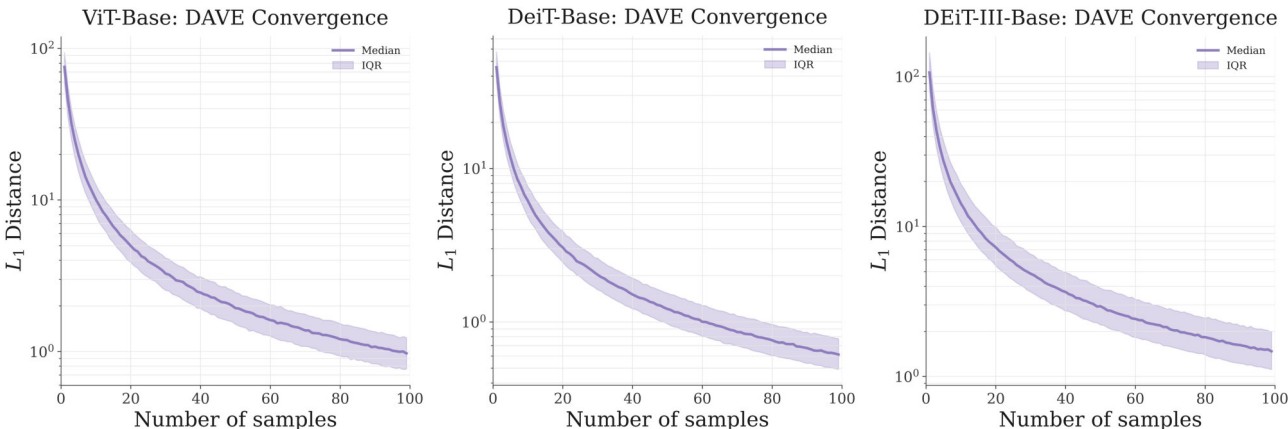

*Figure 12.* **Convergence analysis of DAVE for ViT-B/16-224, DeiT-B/16-224, and DeiT-III-B/16-224.** Convergence is measured as the summed $L_1$ distance between cumulative average attributions at successive averaging steps. Results are reported as median and interquartile range across images and visualized on a logarithmic scale. All models exhibit monotonically decreasing convergence, reaching values on the order of $10^0$ after approximately 100 steps.

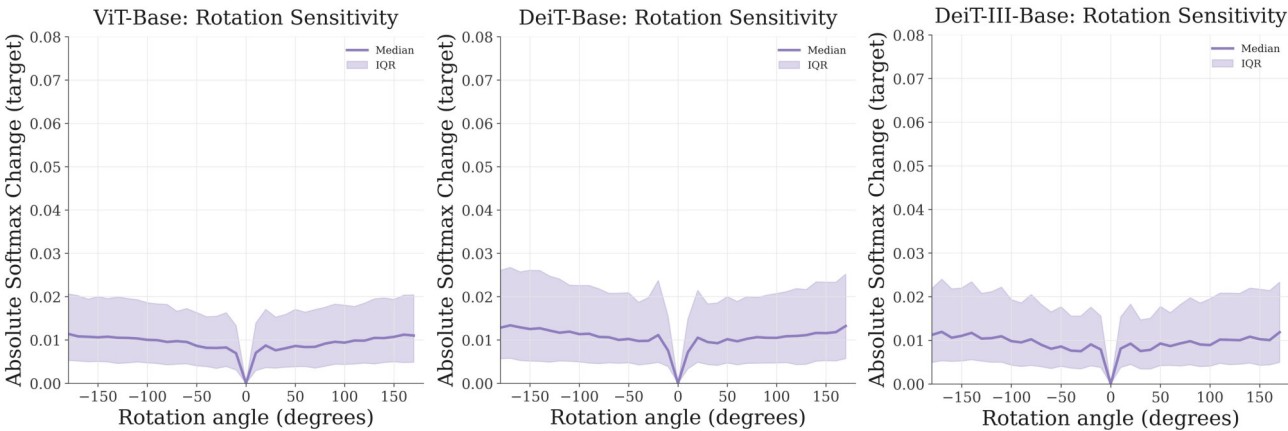

*Figure 13.* **Rotation sensitivity analysis for ViT-B/16-224, DeiT-B/16-224, and DeiT-III-B/16-224.** Sensitivity is measured as the absolute change in softmax probabilities relative to the unrotated input over rotation angles from $-180°$ to $180°$. Results are summarized using the median and interquartile range across images. All models exhibit relatively stable behavior under rotations, with smaller probability changes within the interval $(-20°, 20°)$.

**Rotation sensitivity.** We assess sensitivity to input rotations by measuring the absolute change in softmax probabilities of a target class relative to the unrotated input over a range of rotation angles. For each rotation angle, results are summarized using the median and interquartile range across the evaluated images. All considered models exhibit relatively stable behavior under rotations over a broad range of angles (see Figure 13). Notably, rotations within $(-20°, 20°)$ result in smaller probability changes across models, and we therefore use this interval to define the attribution neighborhood.

**Noise sensitivity.** We assess sensitivity to additive input noise using the same metric as for rotation analysis, measuring the absolute change in softmax probabilities relative to the unperturbed input. Gaussian noise with standard deviation $\sigma \in [0, 2]$ is added to the input, and results are summarized using the median and interquartile range across images for each noise level. We observe model-dependent sensitivity to noise: ViT-B/16-224 exhibits larger probability changes as noise increases, whereas DeiT-III-B/16-224 remains comparatively more stable (see Figure 14). This analysis is used to guide the selection of appropriate noise scales for effective weight smoothing (Section 3.3) in subsequent experiments.

**Shift sensitivity.** We similarly assess sensitivity to horizontal translations, measuring the absolute change in softmax probabilities relative to the unshifted input over a range of pixel shifts (see Figure 15). This analysis motivates the small wrapped translations (up to $0.1$ of the image extent) used to define the attribution neighborhood.

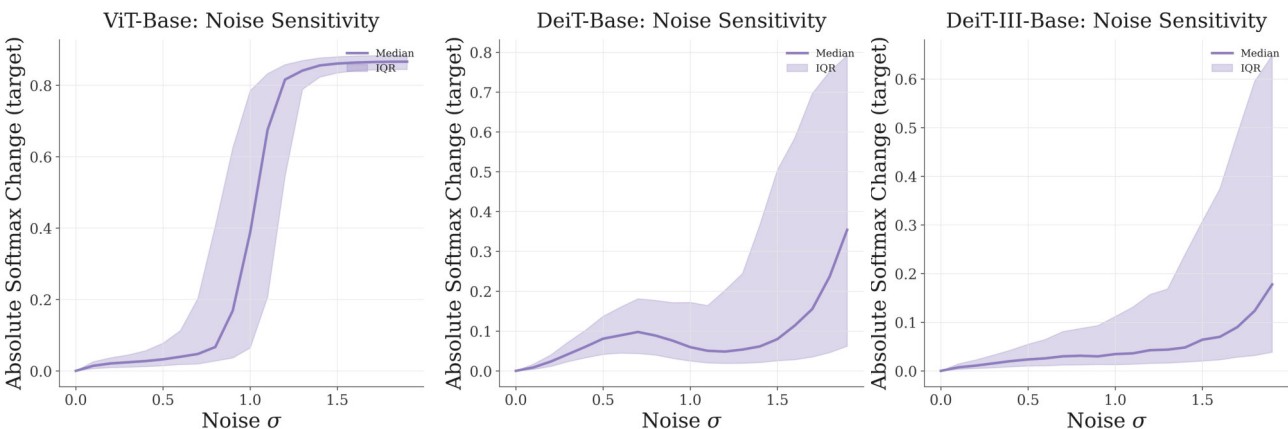

*Figure 14.* **Noise sensitivity analysis for ViT-B/16-224, DeiT-B/16-224, and DeiT-III-B/16-224.** Sensitivity is measured as the absolute change in softmax probabilities relative to the unperturbed input under additive Gaussian noise with standard deviation $\sigma \in [0, 2]$. Results are summarized using the median and interquartile range across images.

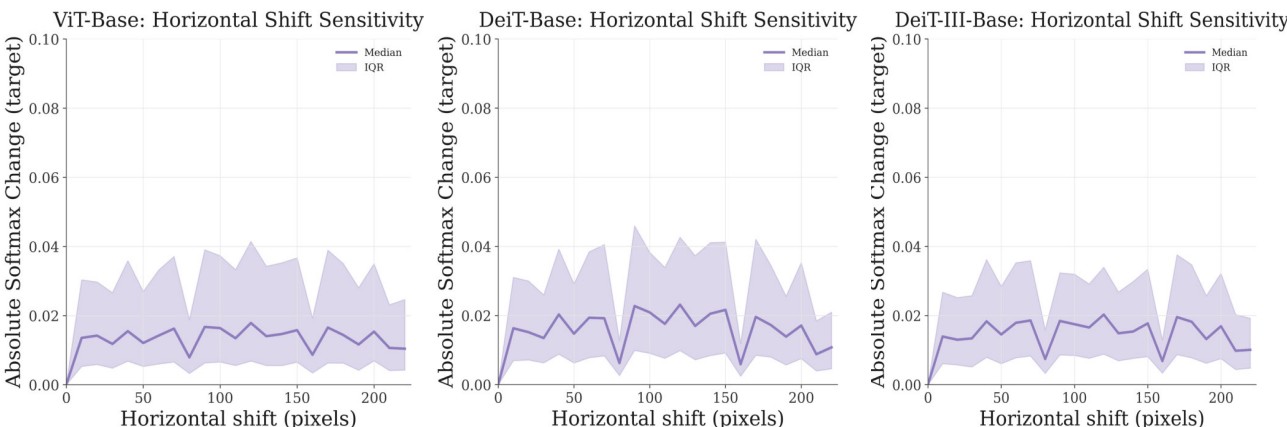

*Figure 15.* **Horizontal shift sensitivity analysis for ViT-B/16-224, DeiT-B/16-224, and DeiT-III-B/16-224.** Sensitivity is measured as the absolute change in softmax probabilities relative to the unshifted input over pixel shifts from 0 to 214. Results are summarized using the median and interquartile range across a subset of randomly sampled 2000 ImageNet-1k evaluation images.

### C.3. Additional Results

**Pixel deletion results.** Figure 16 extends the pixel deletion evaluation to DeiT-B/16 and B-cos-ViT-B/16 (bottom row), in addition to ViT-B/16 and DeiT-III-B/16 (top row). Across all backbones, DAVE exhibits among the flattest deletion curves, indicating strong stability under progressive pixel removal. Notably, on B-cos-ViT-B/16, DAVE remains competitive with the inherent B-cos explanations and improves over standard gradient-based baselines.

**Pixel insertion evaluation.** Figure 17 reports pixel insertion results on DeiT-III-Base, where pixels are progressively revealed according to attribution importance and the target-class probability is tracked. DAVE achieves the highest AUC (66.7), outperforming all compared methods. The faster increase in prediction confidence indicates that DAVE more effectively identifies class-relevant evidence and prioritizes informative image regions.

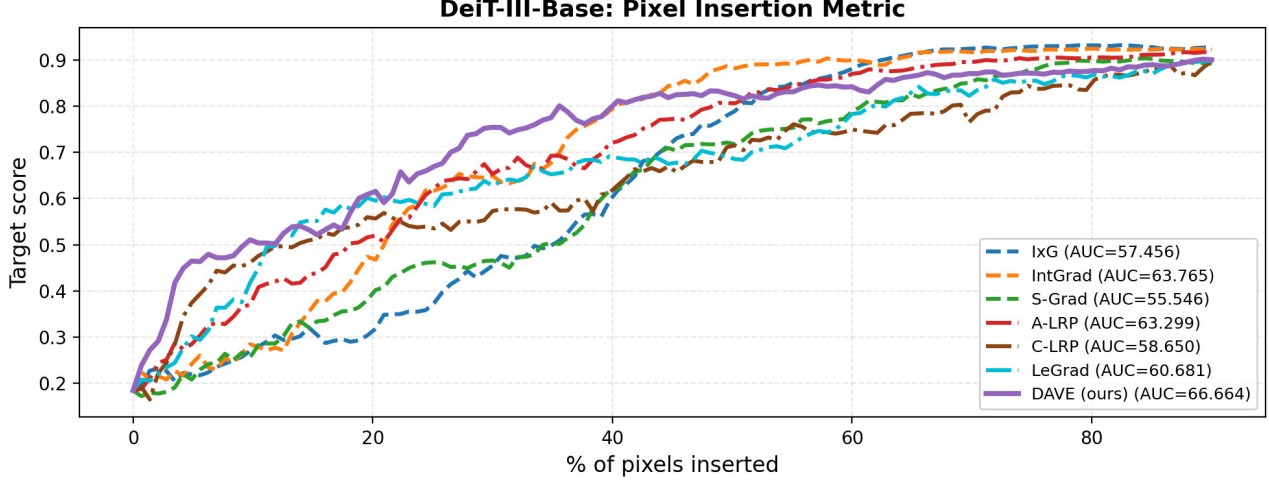

*Figure 16.* **Pixel Deletion (%)** for ViT-B/16 (row 1, col. 1), DeitIII-B/16 (row 1, col. 2), Deit-B/16 (row 2, col. 1) and B-cos-ViT-B/16 (row 2, col. 2) across attribution methods. Each curve corresponds to an attribution method; DAVE (ours) is highlighted in purple.

*Figure 17.* **Pixel insertion evaluation on DeiT-III-Base. DAVE (ours) achieves the highest AUC (66.7),** outperforming all compared methods including Attention LRP (63.3), Integrated Gradients (63.8), LeGrad (60.7), Chefer LRP (58.7), Input×Gradient (57.5), and SmoothGrad (55.5).

## C.4. User Study

To evaluate the practical usefulness of attribution maps beyond standard quantitative metrics, we conduct two user studies assessing human preference and the ability of explanations to convey model decisions.

**Human preference study.**   In the first study, participants were shown an input image together with two attribution maps generated by different methods and asked which explanation better helped them verify that the model made its prediction for the right reasons (Figure 20). The Average Preference Rate (APR) results in Figure 18 show that DAVE is consistently preferred over all compared methods. This suggests that DAVE explanations are perceived as more helpful and trustworthy for understanding model behavior.

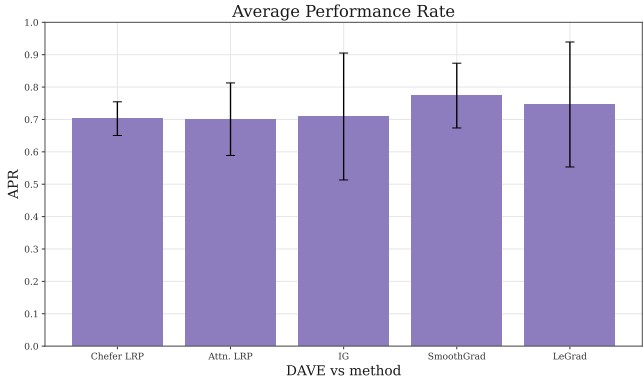

*Figure 18.* **User study 1: Human preference study.** Bar plot showing the Average Preference Rate (APR), defined as the proportion of trials in which DAVE is selected over competing methods (excluding "Can't decide" responses). DAVE is consistently preferred across all baselines, indicating that it provides more helpful and trustworthy explanations of the model's decisions.

**HIVE-style class identification.**   In the second study, following a HIVE-style setup (Kim et al., 2022), participants were presented with attribution maps corresponding to the top-4 predicted classes and asked to infer the correct class using only the highlighted evidence (Figure 21). As shown in Figure 19, DAVE achieves the highest identification accuracy across all methods. These results indicate that DAVE more effectively captures class-discriminative evidence and produces explanations that better communicate the model's reasoning process.

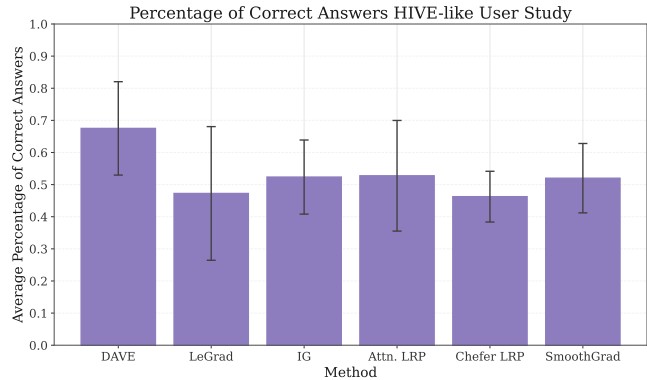

*Figure 19.* **User study 2: Percentage of correct answers in the HIVE-style evaluation.** Participants are asked to identify the correct class based solely on attribution maps for the top-4 predicted classes. DAVE achieves the highest accuracy, significantly outperforming all baseline methods, indicating that its explanations more clearly convey class-discriminative evidence and enable users to better infer the model's prediction.

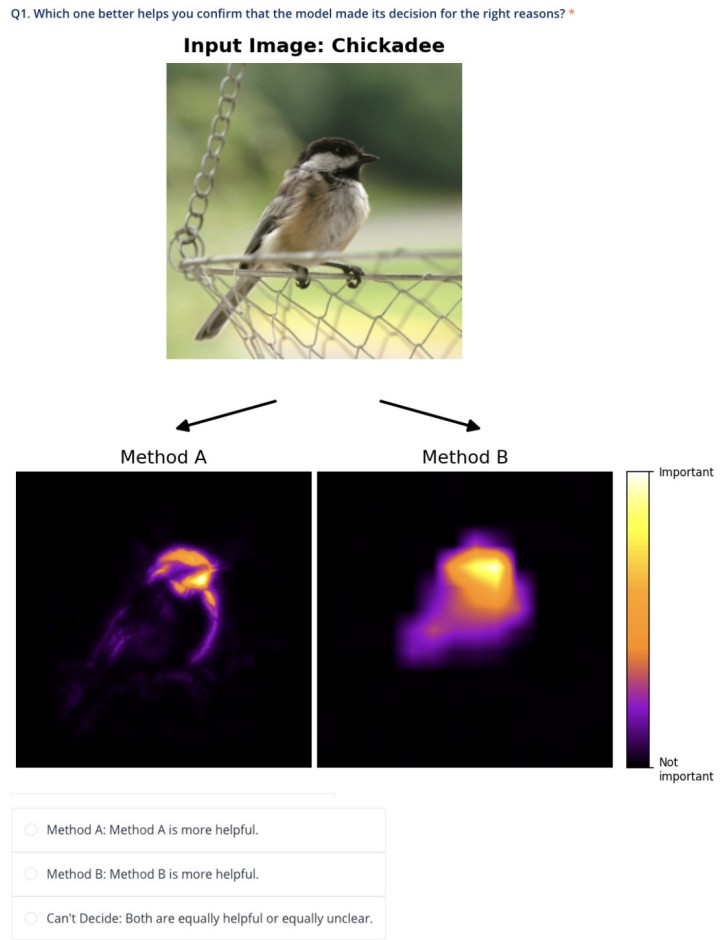

*Figure 20.* **User study 1: Human preference study.** Participants were shown an input image with two attribution maps (Method A vs. Method B) and asked: "Which explanation better helps you confirm that the model made its decision for the right reasons?" Possible responses were: Method A, Method B, or Can't decide.

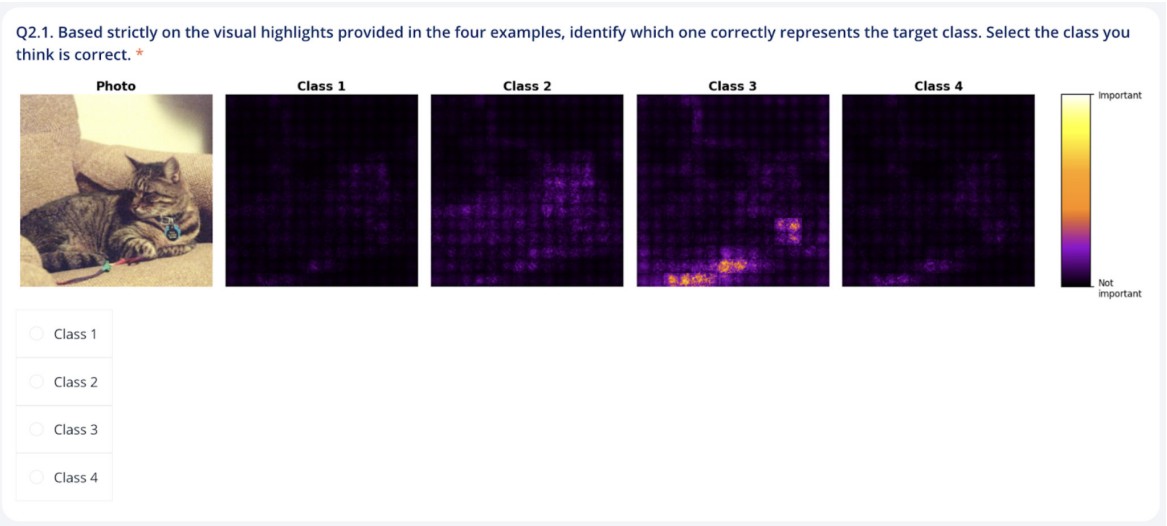

*Figure 21.* **User study 2: HIVE-style class identification**. Participants are shown an input image with attribution maps for the top-4 predicted classes (here: SmoothGrad) and asked to identify the correct class based only on the highlighted regions.

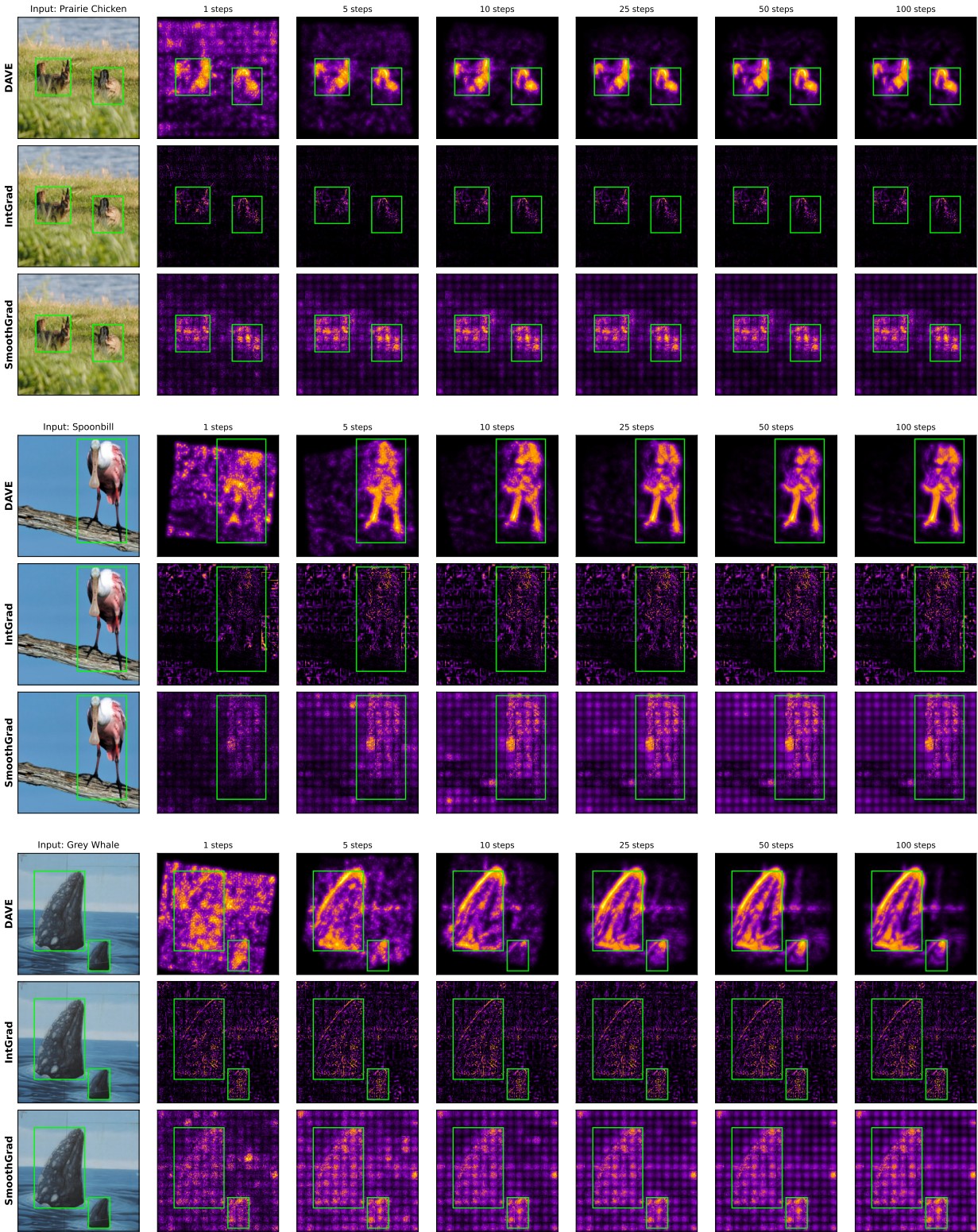

*Figure 22.* **Qualitative comparison of attribution maps across step counts on DeiT-III-B/16.** Each row group shows attribution maps for DAVE (ours), IntGrad, and SmoothGrad at 1, 5, 10, 25, 50, and 100 steps. The green bounding box indicates the target object region. **DAVE produces sharper, semantically focused attribution maps that converge rapidly with steps**, while IntGrad and SmoothGrad remain noisy and diffuse regardless of step count. All our main results use 50 steps in practice.

# D. Qualitative Results

In this appendix, we provide qualitative analyses of DAVE attributions on ImageNet-1k validation samples. We compare DAVE explanations across ViT-B/16-224, DeiT-B/16-224, and DeiT-III-B/16-224 (Figure 23); contrast DAVE with prior post-hoc attribution methods across these same ViT backbones (Appendix D.2); illustrate class-consistent attributions in which recurring class-relevant features are highlighted across different images (Figure 25); and compare DAVE with the model inherent explanations of B-cos ViTs (Appendix D.4). We then analyze class-discriminative behavior under misclassification and across confusable classes (Appendix D.5); the role of the operator-variation term, contrasting DAVE with an augmented SmoothGrad variant (Appendix D.6); qualitative comparisons against post-hoc methods on a self-supervised DINO ViT (Appendix D.7); and robustness under distribution shift on ImageNet-Rendition (Hendrycks et al., 2021a) (Appendix D.8). Finally, we extend the analysis to DINOv2 with register tokens (Appendix D.9) and to convolutional backbones via ConvNeXt-S (Appendix D.10).

## D.1. Model Comparison

We qualitatively compare attributions produced by DAVE on ViT-B/16-224, DeiT-B/16-224, and DeiT-III-B/16-224 for ImageNet-1k evaluation samples (see Figure 23). While for some samples, the models attend to similar regions and features (indicating shared representations of class-relevant cues), for others, the models emphasize different regions. However, in both cases these regions remain semantically associated with the target object, suggesting alternative but valid feature representations across architectures.

## D.2. Comparison with Post-Hoc Attributions across Backbones

Figure 24 compares DAVE to common post-hoc attribution methods across three ViT backbones (DeiT-III-B/16-224, DeiT-B/16-224, and ViT-B/16-224). Across all backbones, several baselines exhibit structured patch-grid artifacts (notably SmoothGrad), fragmented responses, or diffuse saliency that extends beyond the target object. In contrast, DAVE consistently produces sharper and more spatially coherent maps that concentrate on object-centric and class-relevant structures (e.g., animal contours and distinctive parts), while suppressing background responses. The examples suggest that DAVE's qualitative advantages are not tied to a specific ViT variant, but persist across different transformer backbones and training recipes.

## D.3. Class Consistency

In the qualitative analysis, we observe class-consistent feature attributions produced by DAVE, where highlighted regions correspond to semantically meaningful and recurring features associated with the predicted class across different images (see Fig. 25).

## D.4. Comparison with Inherently Interpretable Models

Figure 26 compares DAVE to the model inherent explanations of B-cos ViTs across ImageNet-1k validation examples. While B-cos explanations can be noisy—especially for B-cos ViTs trained without a convolutional stem (B-ViT)—DAVE yields sharper, more object-aligned attributions that better suppress background noisy responses. Notably, DAVE preserves class-relevant structure (e.g., eyes, contours, and distinctive textures) and produces more spatially coherent maps across both B-ViT and the convolutional-stem B-ViT-C variant, consistent with the localization improvements reported in Table 2. Figure 27 provides an extended comparison of DAVE against gradient-based baselines and the native B-cos explanations on a B-cos ViT-B/16-224 model.

## D.5. Class-Discriminative Attribution Analysis

We further evaluate the ability of DAVE to produce class-discriminative explanations under both distribution shift and class ambiguity. Figure 28 compares attribution maps on misclassified ImageNet-A (Hendrycks et al., 2021b) samples by visualizing explanations for both the predicted and ground-truth classes. While competing methods often produce nearly identical or noisy maps for different targets, DAVE generates spatially distinct explanations that align with class-specific visual evidence. This enables more interpretable analysis of model failures by both the evidence supporting the incorrect prediction and the features associated with the correct class.

Figure 29 further analyzes class specificity by comparing attribution maps for the top-4 predicted classes of the same image. DAVE consistently shifts attention between semantically meaningful regions depending on the target class, whereas baseline methods frequently highlight similar regions across classes or produce weak and noisy responses. These results show that DAVE more effectively captures class-discriminative visual features.

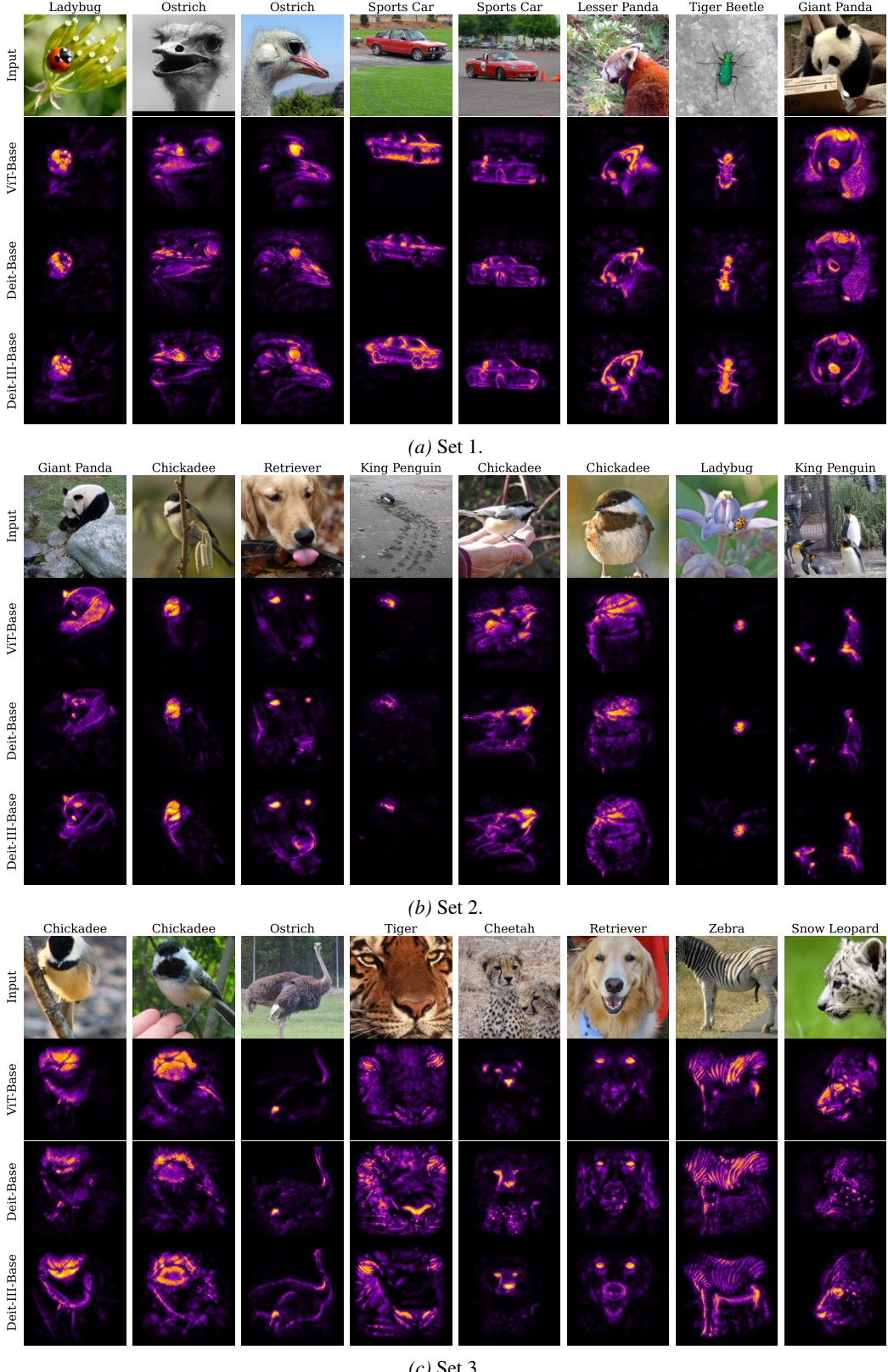

*Figure 23.* **DAVE explanations across ViT backbones.** For ImageNet-1K validation examples (columns), we show the input image (top) and DAVE attribution maps produced by ViT-B/16-224, DeiT-B/16-224, and DeiT-III-B/16-224 (rows). Across samples, the three models often highlight similar object-centric, class-relevant regions, while in some cases emphasizing different but semantically meaningful cues, reflecting differences in learned feature representations.

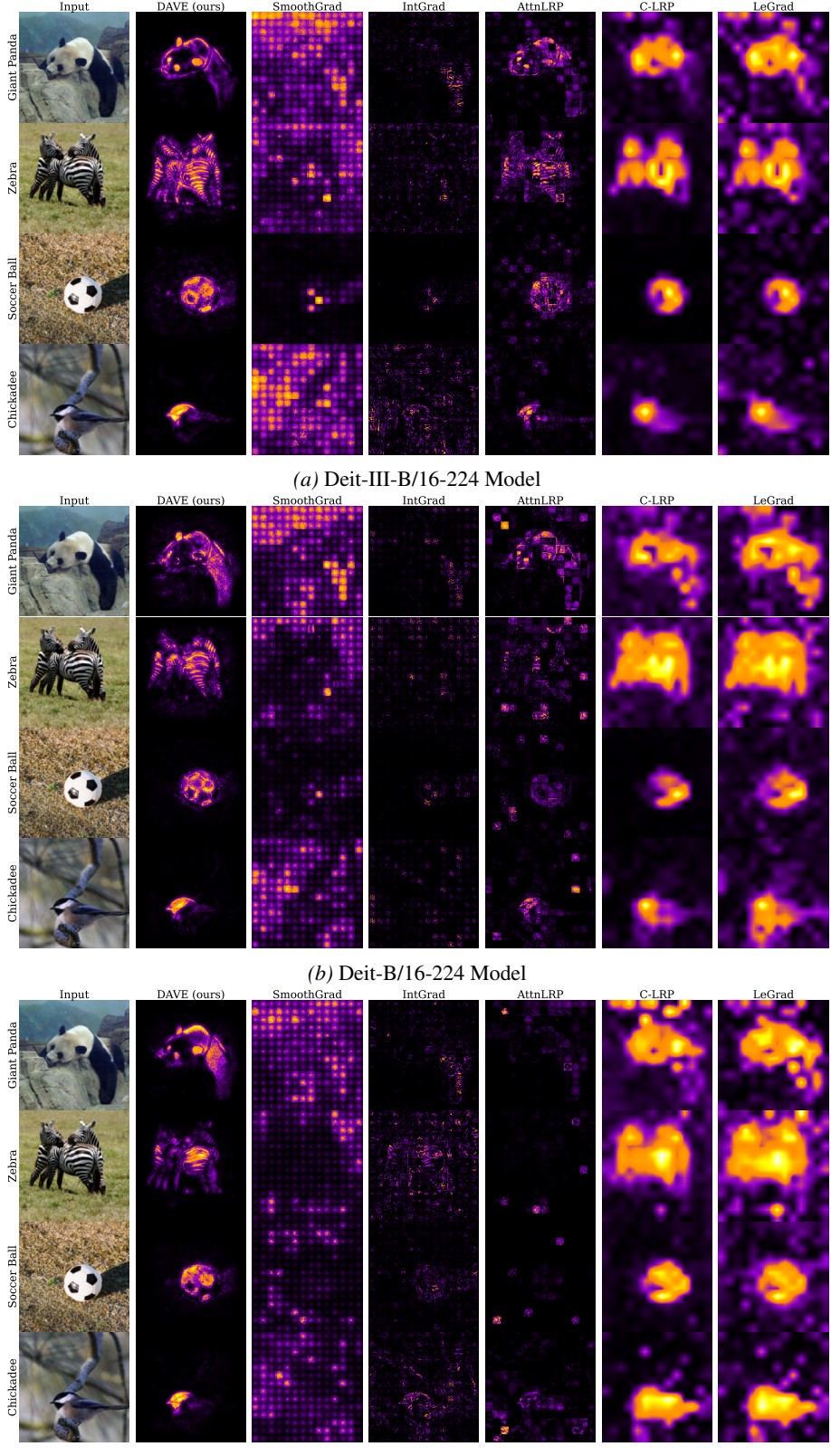

*(a)* Deit-III-B/16-224 Model

*(b)* Deit-B/16-224 Model

*(c)* ViT-B/16-224 Model

*Figure 24.* **DAVE vs. post-hoc attribution methods across ViT backbones.** For ImageNet-1K validation examples (rows), we show the input image (left) and attribution maps produced by DAVE, SmoothGrad, Integrated Gradients, AttnLRP, C-LRP, and LeGrad (columns). Results are shown for (a) DeiT-III-B/16-224, (b) DeiT-B/16-224, and (c) ViT-B/16-224. Across backbones, DAVE yields sharper, more object-aligned and spatially coherent explanations with reduced patch-grid artifacts compared to prior methods.

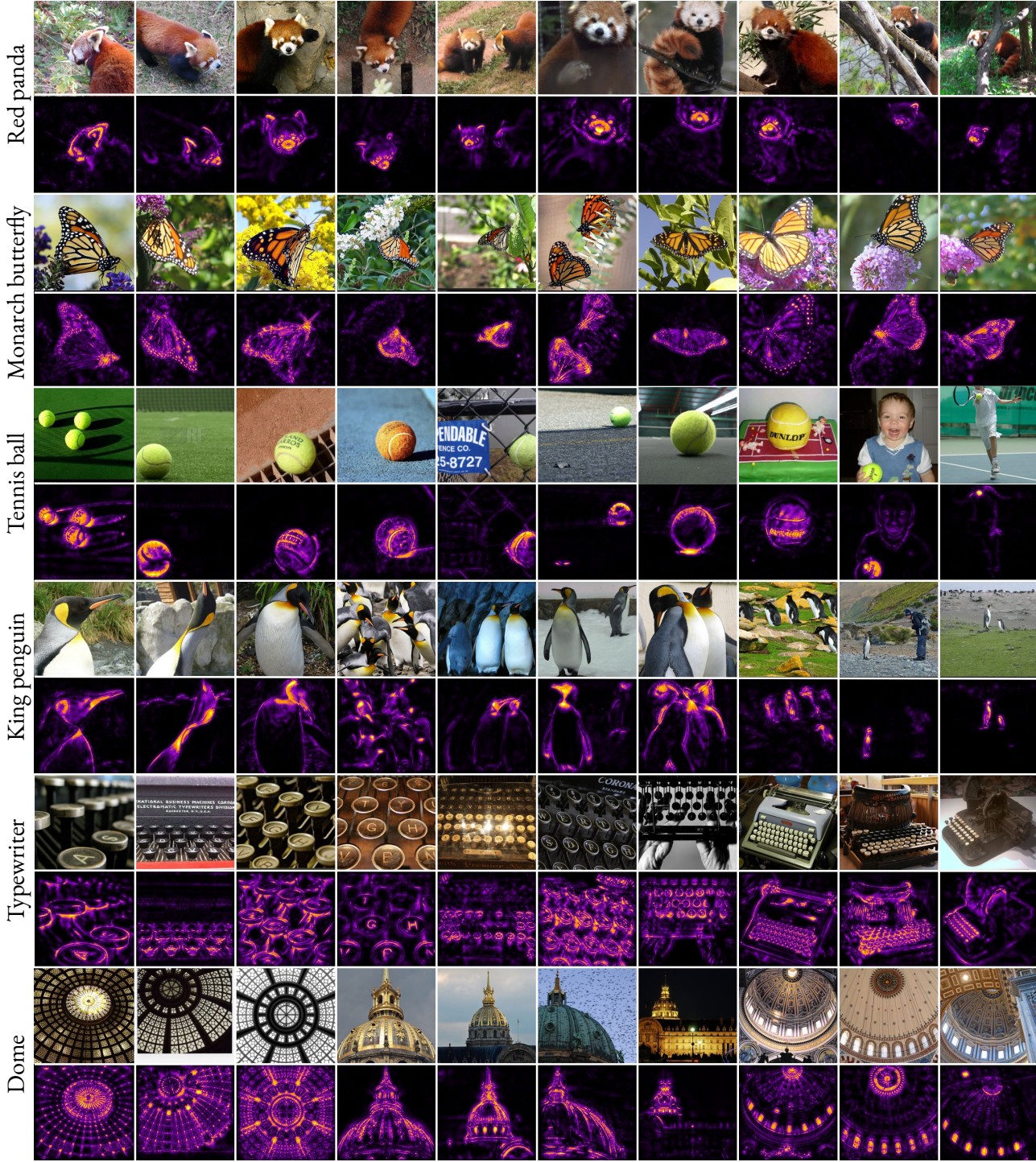

*Figure 25.* **Examples of class-consistent features detected by DAVE on DeiT-III-B/16-224.** For each ImageNet-1k class (rows), we show multiple validation images (top) and the corresponding DAVE attribution maps (bottom). Across diverse instances, DAVE consistently highlights recurring, semantically meaningful class cues (e.g., characteristic parts, textures, and contours), while adapting to changes in pose, scale, and background.

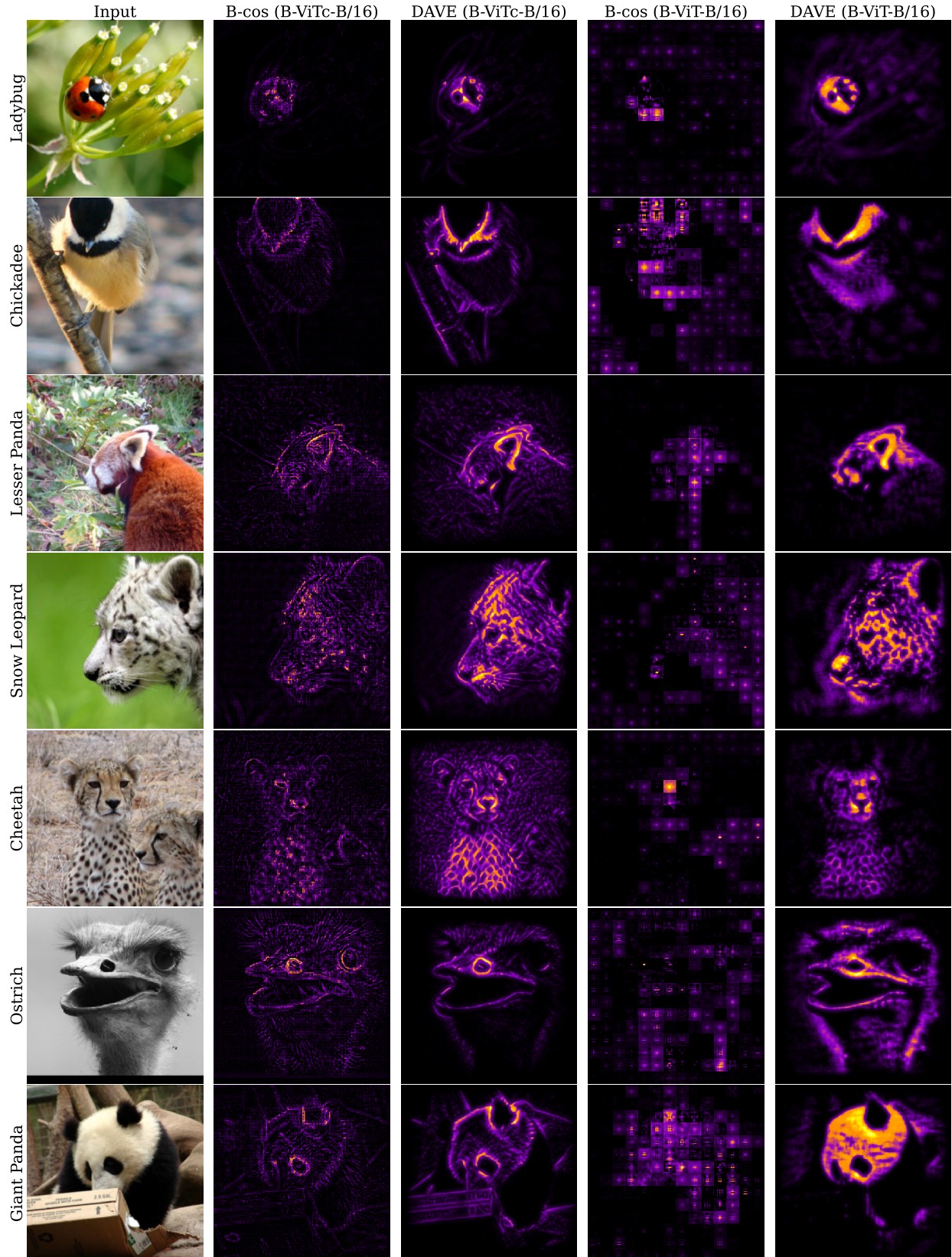

*Figure 26.* **DAVE explanations on inherently interpretable B-cos ViTs**: Our (DAVE) explanations visually highlight object centric features precisely especially for B-cos ViTs trained without a convolutional stem, showing robustness of the proposed method to architectural variations (see col. 4 vs col. 5) Even for B-cos ViTs with a convolutional stem, the DAVE attributions seem to highlight similar regions as the B-cos ones (col. 2 vs col 3), and also improving upon the B-cos localization performance.

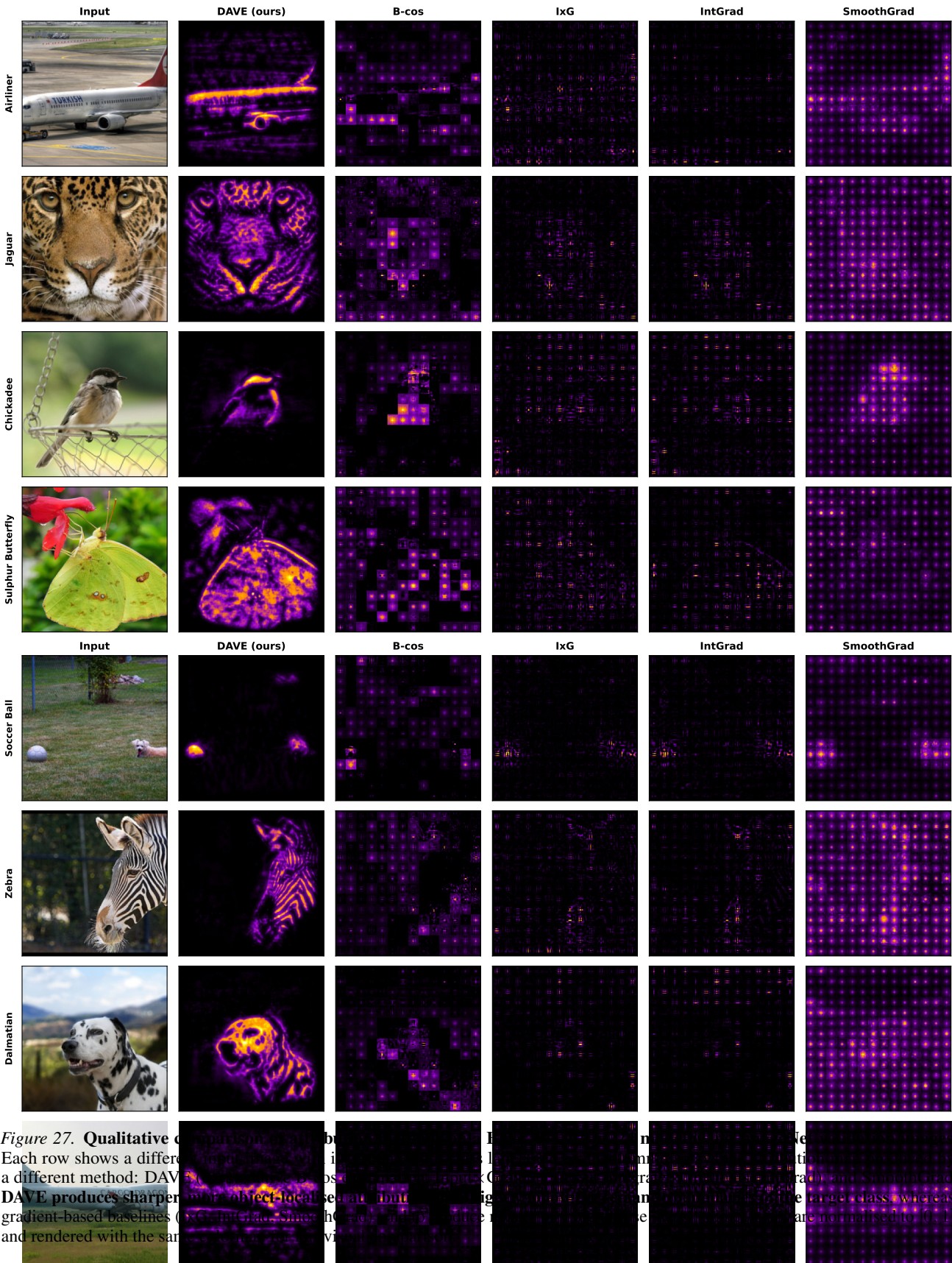

*Figure 27.* **Qualitative comparison of attribution methods on ImageNet classes.** Each row shows a different input image, with each column showing attributions from a different method: DAVE (ours), B-cos, IxG, IntGrad, and SmoothGrad. **DAVE produces sharper, more object-localised attributions that highlight semantically meaningful regions of the target class, whereas gradient-based baselines produce noisier maps. All attributions are normalised and rendered with the same colour map for visualisation.

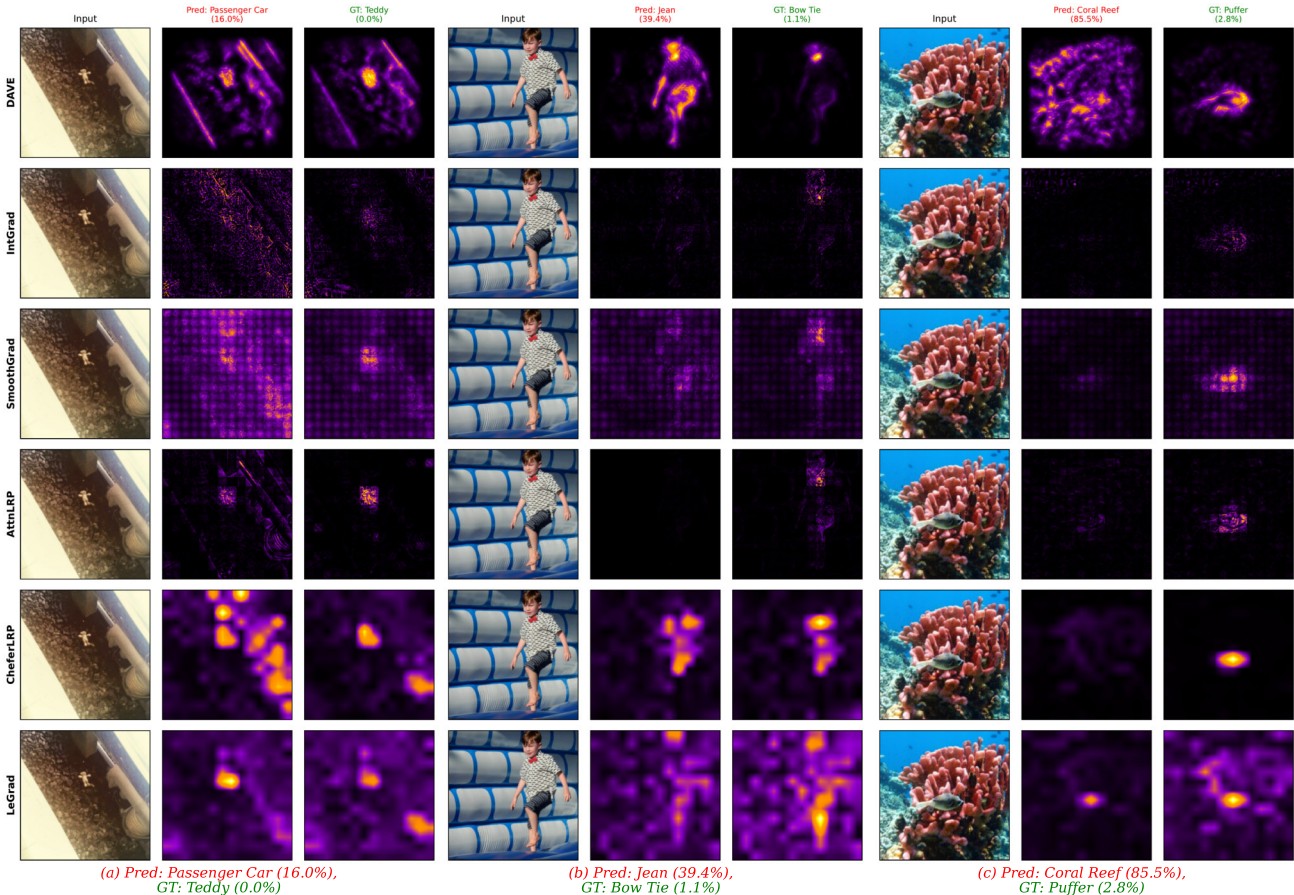

*(a) Pred: Passenger Car (16.0%),
GT: Teddy (0.0%)*

*(b) Pred: Jean (39.4%),
GT: Bow Tie (1.1%)*

*(c) Pred: Coral Reef (85.5%),
GT: Puffer (2.8%)*

*Figure 28.* **DAVE vs baselines, class-discriminative attributions on misclassified ImageNet-A samples.** Each column triplet shows the input image, the attribution map for the predicted class, and the attribution map for the ground-truth class; rows correspond to different attribution methods. Despite the incorrect predictions, DAVE produces spatially distinct maps for the two classes: in (a), the predicted-class map highlights the elongated conveyor surface (consistent with *Passenger Car*), while the GT map precisely localises the teddy-bear figurine; in (b), the predicted-class map focuses on the child's shorts (*Jean*), whereas the GT map shifts to the bow tie at the neckline; in (c), the predicted-class map covers the coral background (*Coral Reef*), and the GT map isolates the puffer fish. Baseline methods (IntGrad, SmoothGrad, AttnLRP, CheferLRP, LeGrad) either produce similar maps for both classes or yield weak, noisy attributions, failing to distinguish the visual evidence for the predicted versus ground-truth class. This demonstrates that DAVE can reliably surface class-discriminative features even under distribution shift, providing interpretable explanations for *why* the model errs and *what* it would need to attend to for the correct class.

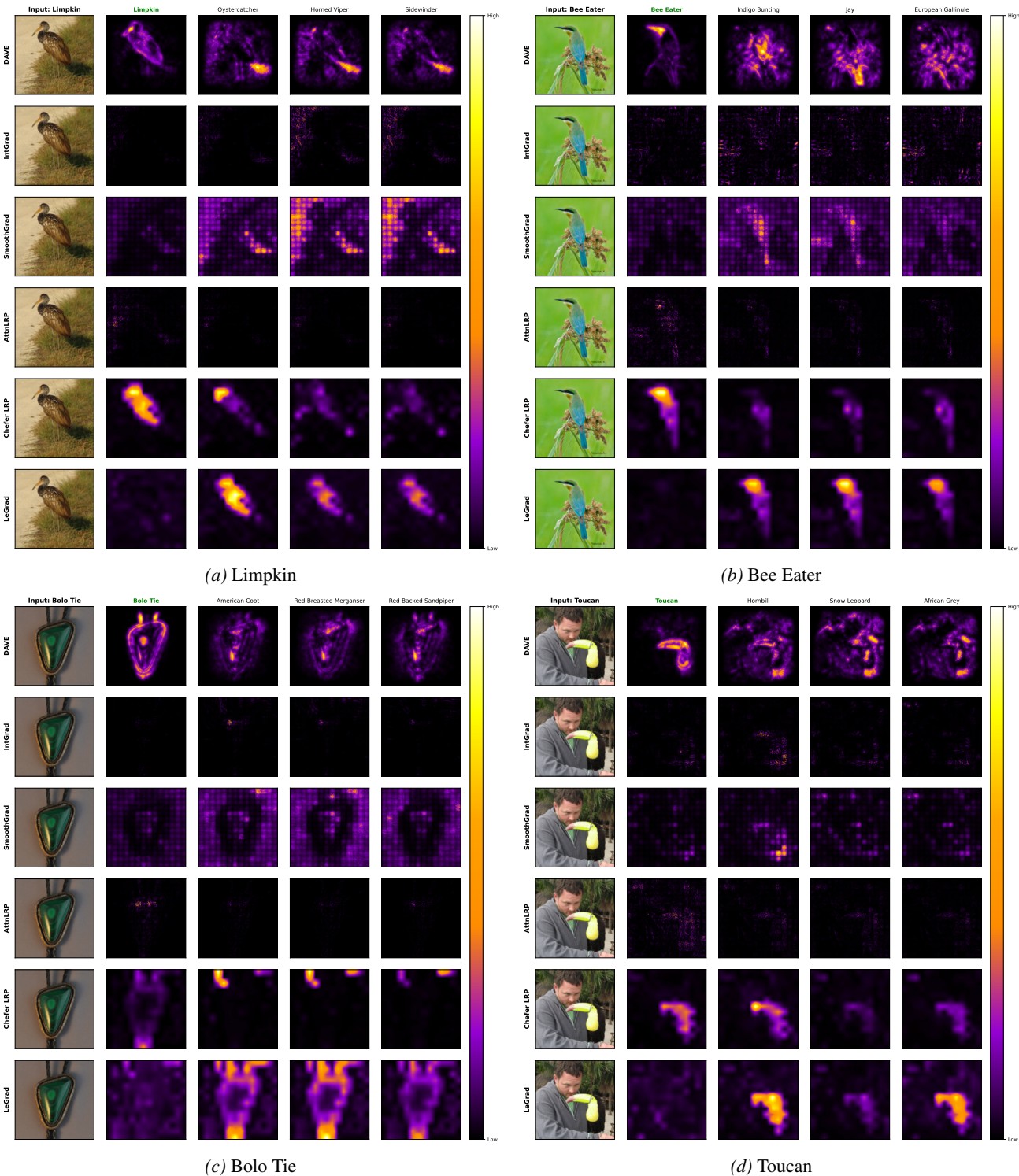

*Figure 29.* **Class-discriminative attribution comparison.** For each sample, we show attribution maps computed for the model's top-4 predicted classes across six methods: DAVE, IntGrad, SmoothGrad, AttnLRP, Chefer LRP, and LeGrad (rows top to bottom). The ground-truth class is highlighted in **green**; remaining columns show confusable classes. **DAVE (top row)** consistently produces spatially distinct attribution maps across target classes (col. 2 for each panel), highlighting different semantically relevant regions. For instance, in (d) DAVE localizes the toucan's beak for "Toucan" but shifts attention to the background for "Snow Leopard." In contrast, gradient-based methods (IntGrad, SmoothGrad) yield near-zero or noisy patch-level maps, while attention-based methods (AttnLRP, Chefer LRP, LeGrad) tend to highlight similar regions regardless of the target class, indicating limited class specificity.

## D.6. Isolating Operator Variation

We visualize the effect of retaining operator variation in the DAVE pipeline, effectively turning the method into an augmented variant of SmoothGrad. Figures 30 and 31 compare DAVE with this ablated formulation on both single-object and multi-object examples. Reintroducing the full gradient produces noisier and less spatially coherent attribution maps, often highlighting background regions or multiple objects simultaneously. In contrast, DAVE yields sharper and more semantically aligned explanations that better isolate class-specific visual evidence. These qualitative results support the quantitative findings from Section 5.2 Table 3), indicating that the operator-variation term degrades attribution quality.

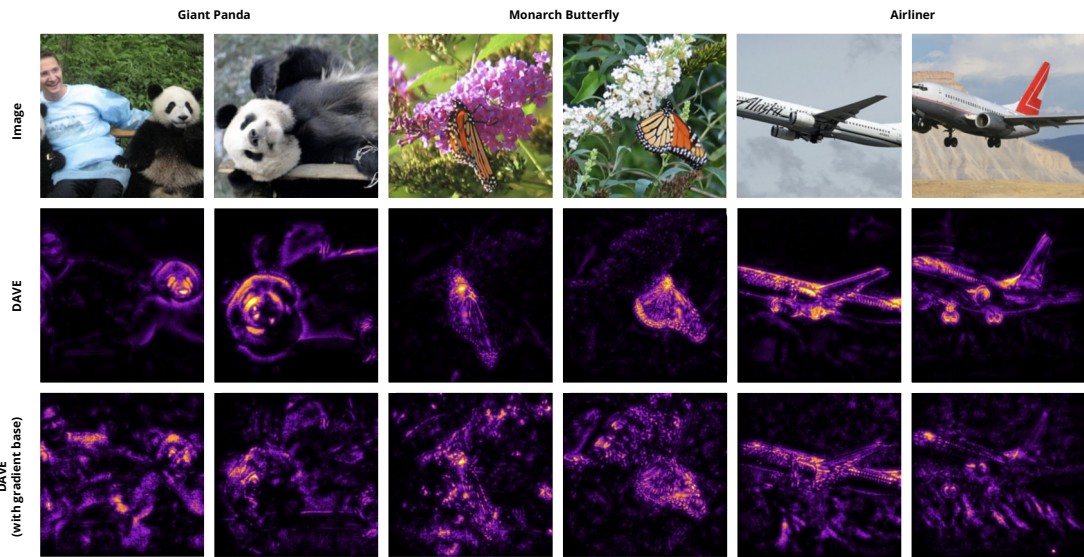

*Figure 30.* **Comparison between DAVE and augmented SmoothGrad.** Top: input images from three ImageNet-1k classes (Giant Panda, Monarch Butterfly, Airliner). Middle: DAVE. Bottom: augmented SmoothGrad (DAVE with entire gradient instead of effective transformation). **Using full gradients within the DAVE pipeline leads to significantly noisier and less structured attributions, whereas DAVE produces clearer and more semantically coherent explanations.** *Results on DeiT-III-B/16 model.*

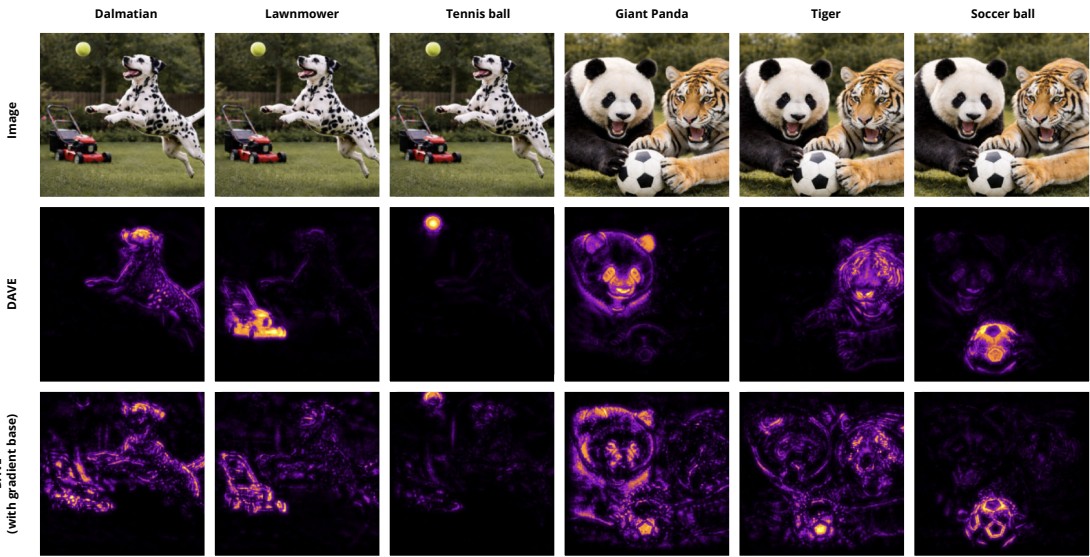

*Figure 31.* **Comparison between DAVE and augmented SmoothGrad on multi-object images.** Top: input images containing multiple objects from different classes. Middle: DAVE. Bottom: augmented SmoothGrad (DAVE with entire gradient instead of effective transformation). **DAVE is able to separate class-specific objects in a semantically aligned manner, whereas using full gradients leads to noisier attributions that often highlight multiple objects simultaneously.** *Results on DeiT-III-B/16 model.*

## D.7. Comparison with Post-Hoc Attributions on a Self-Supervised Model

Figure 32 compares DAVE to common post-hoc attribution methods on a self-supervised DINO ViT-B/16-224 model (Caron et al., 2021), including SmoothGrad, Integrated Gradients, AttnLRP, Chefer-LRP, and LeGrad. Across examples, several baselines exhibit pronounced patch-grid artifacts, diffuse responses, or fragmented saliency that extends into the background. In contrast, DAVE consistently yields sharper, more spatially coherent attributions that concentrate on object-centric and class-relevant structures (e.g., contours and distinctive parts), while reducing structured artifacts introduced by tokenization and attention routing. These observations align with our quantitative localization results in the main paper and support DAVE's ability to produce high-resolution, visually interpretable attributions on ViT based models.

## D.8. Qualitative Out-of-Distribution Analysis

We evaluate the robustness of DAVE under domain shift using ImageNet-Rendition (Hendrycks et al., 2021a) samples, which depict ImageNet classes in different styles such as paintings, sketches, and cartoons. Figure 33 compares DAVE with existing attribution methods across paired in-distribution and out-of-distribution examples of the same class. While baseline methods often become noisy or lose spatial focus under distribution shift, DAVE consistently highlights semantically meaningful and class-discriminative object regions across domains.

Figure 34 further demonstrates that DAVE preserves consistent attribution structure across a wide range of ImageNet-Rendition classes despite appearance changes. In particular, DAVE continues to localize characteristic object features rather than domain-specific textures or stylistic patterns. These qualitative results suggest that DAVE captures more stable semantic evidence underlying the model's prediction and generalizes more reliably under distribution shift.

## D.9. DINOv2 with Register Tokens and PCA Baseline

Figure 35 compares DAVE, PCA (raw), and PCA (aug) on DINOv2 ViT-B/14 with register tokens (Darcet et al., 2024) and on DeiT-III. PCA (aug) produces visibly cleaner, more object-centric maps on DINOv2 with registers than on DeiT-III, but remains class-agnostic in both cases, whereas DAVE consistently localizes the target class. The corresponding quantitative results are reported in Appendix C.1.

## D.10. Extension to Convolutional Networks (ConvNeXt-S)

DAVE is not strictly limited to ViTs: the framework applies to any architecture whose layers admit the decomposition in Equation 2. We provide an initial evaluation on ConvNeXt-S (Liu et al., 2022), a modern CNN backbone. Although DAVE's augmentation design is motivated by ViT-specific artifacts and may not be optimal for CNNs, DAVE produces qualitatively sharper and more semantically focused maps than gradient-based baselines (SmoothGrad, IntGrad, Input×Gradient, GradCAM) without any CNN-specific tuning (Figure 36). Designing CNN-tailored transformations and noise schedules is a promising direction for future work.

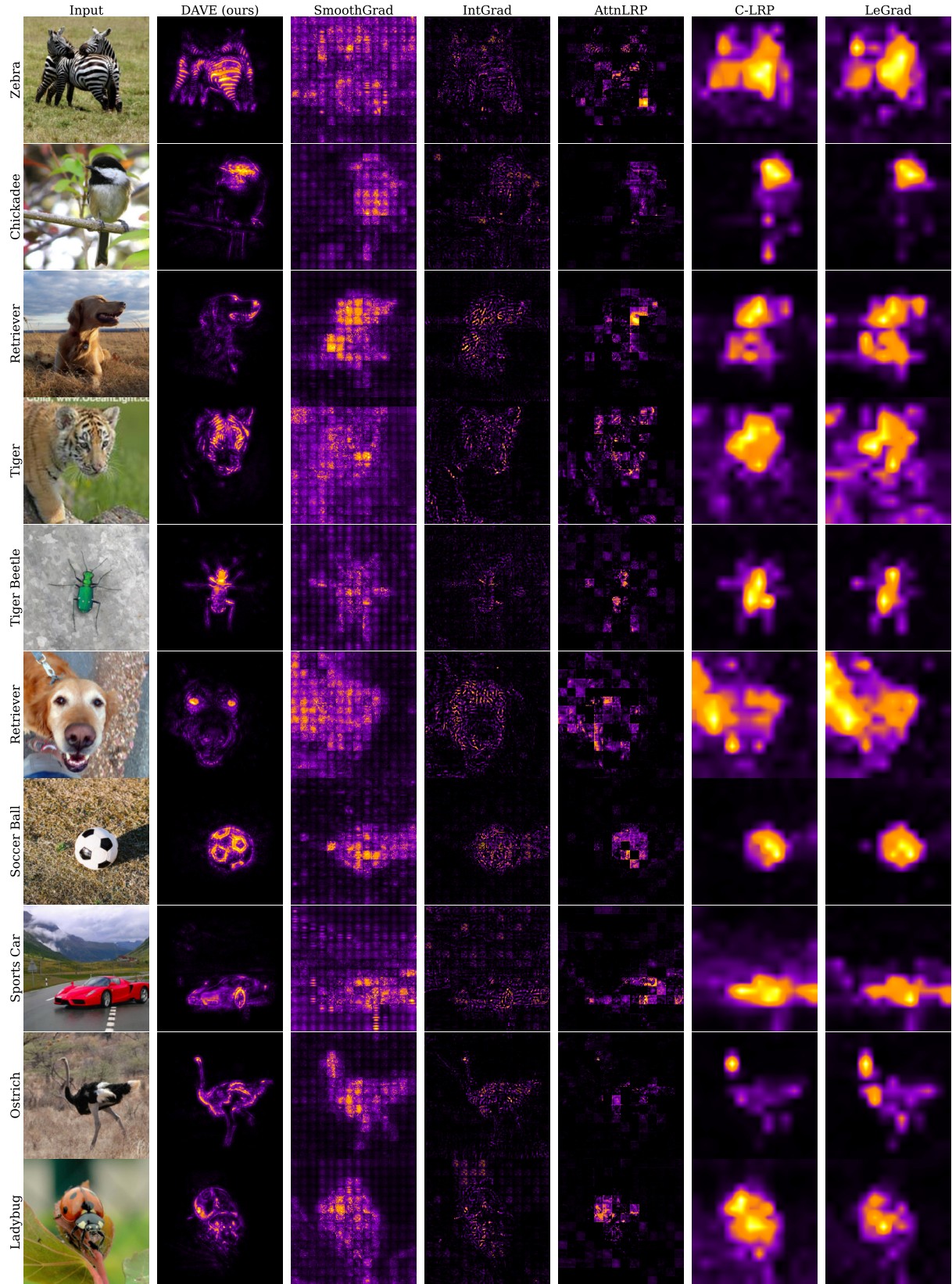

*Figure 32.* **DAVE vs. post-hoc attribution methods on DINO ViT-B/16-224.** For ImageNet-1k validation examples (rows), we show the input image (left) and attribution maps produced by DAVE, SmoothGrad, Integrated Gradients, AttnLRP, C-LRP (Chefer-LRP), and LeGrad (columns). DAVE yields sharper, more object-aligned and spatially coherent explanations with reduced patch-grid artifacts compared to prior methods.

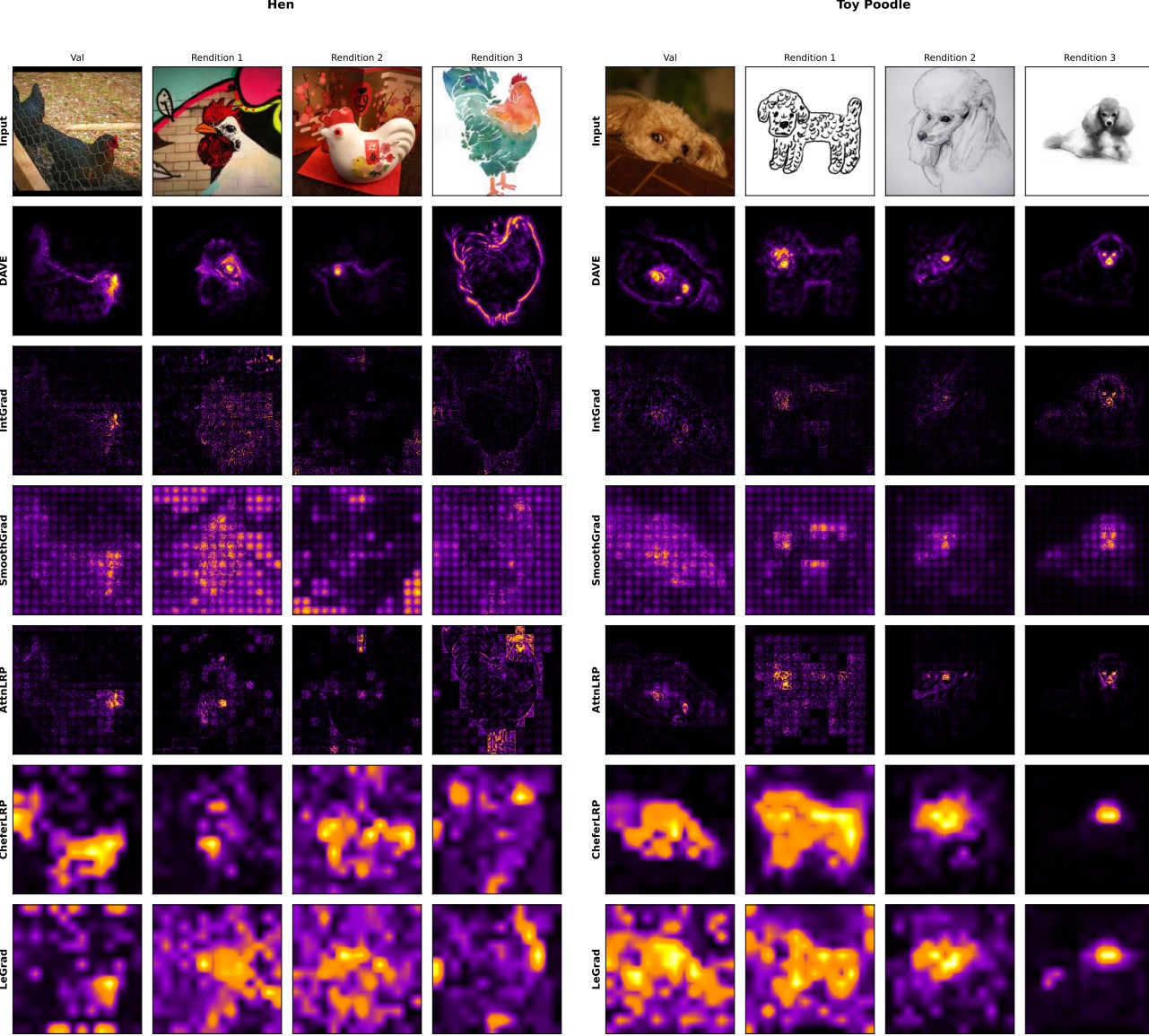

*Figure 33.* **DAVE produces consistent, discriminative attributions across in-distribution and out-of-distribution samples.** Each panel shows attributions for a single class across an ImageNet validation image (left column) and ImageNet-Rendition samples (remaining columns), which depict the same class in varied artistic styles (paintings, cartoons, sketches, etc.). DAVE consistently highlights class-discriminative regions-such as the object's defining features, regardless of domain shift, whereas baseline methods (IntGrad, SmoothGrad, AttnLRP, CheferLRP, LeGrad) produce noisier or less focused attributions that degrade under distribution shift.

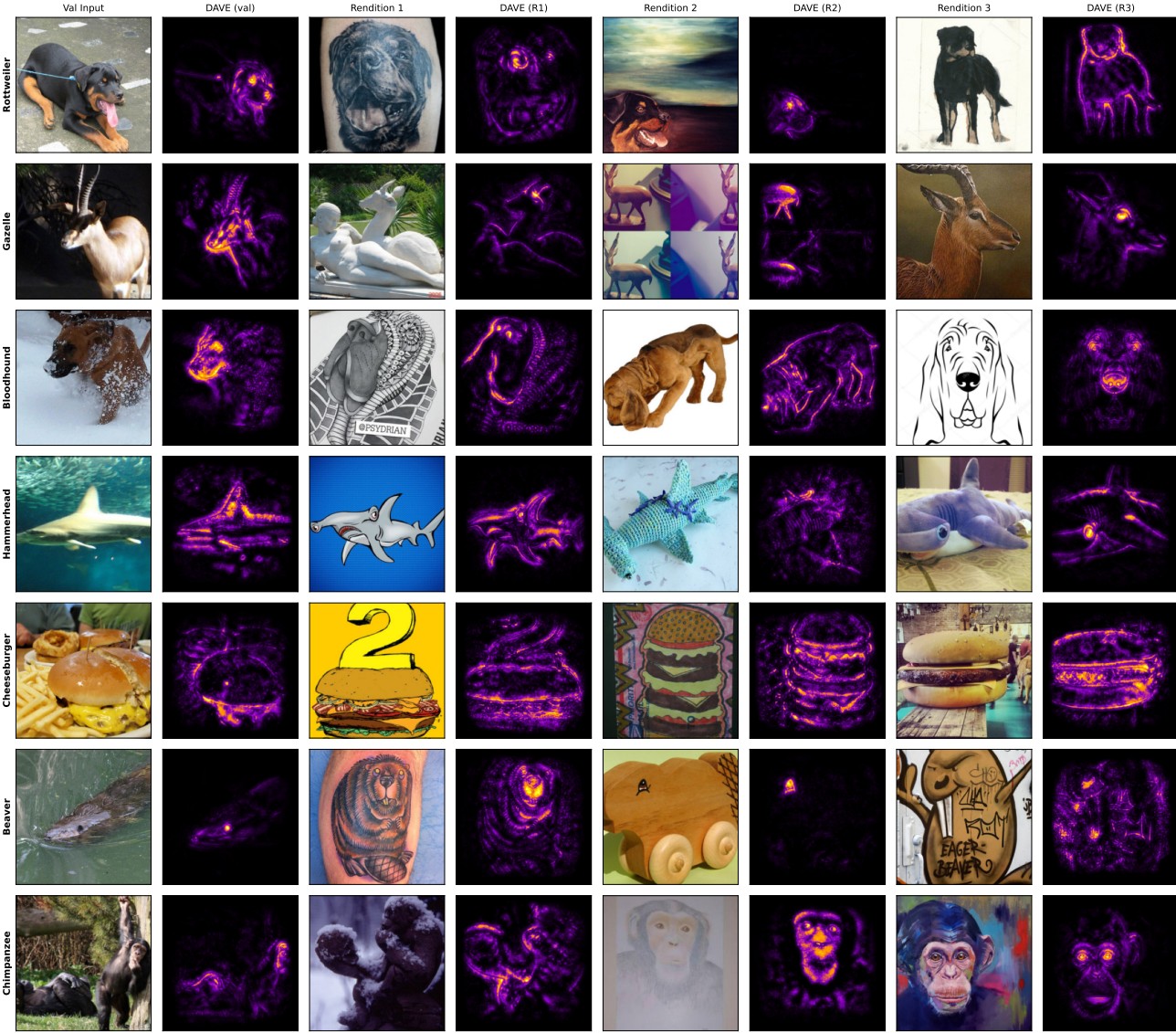

*Figure 34.* **DAVE attributions generalize across domain shift for diverse ImageNet-Rendition classes.** Each row corresponds to a different class. The first two columns show an ImageNet validation image and its DAVE attribution map, while the subsequent columns show ImageNet-Rendition samples of the same class, depicting the object in varied artistic styles such as paintings, cartoons, and sketches, alongside their corresponding DAVE attributions. Despite significant appearance changes across domains, DAVE consistently localizes the same class-discriminative features (e.g., distinctive body parts, object silhouettes), demonstrating that the learned attribution mechanism captures semantic class identity rather than domain-specific texture or style cues.

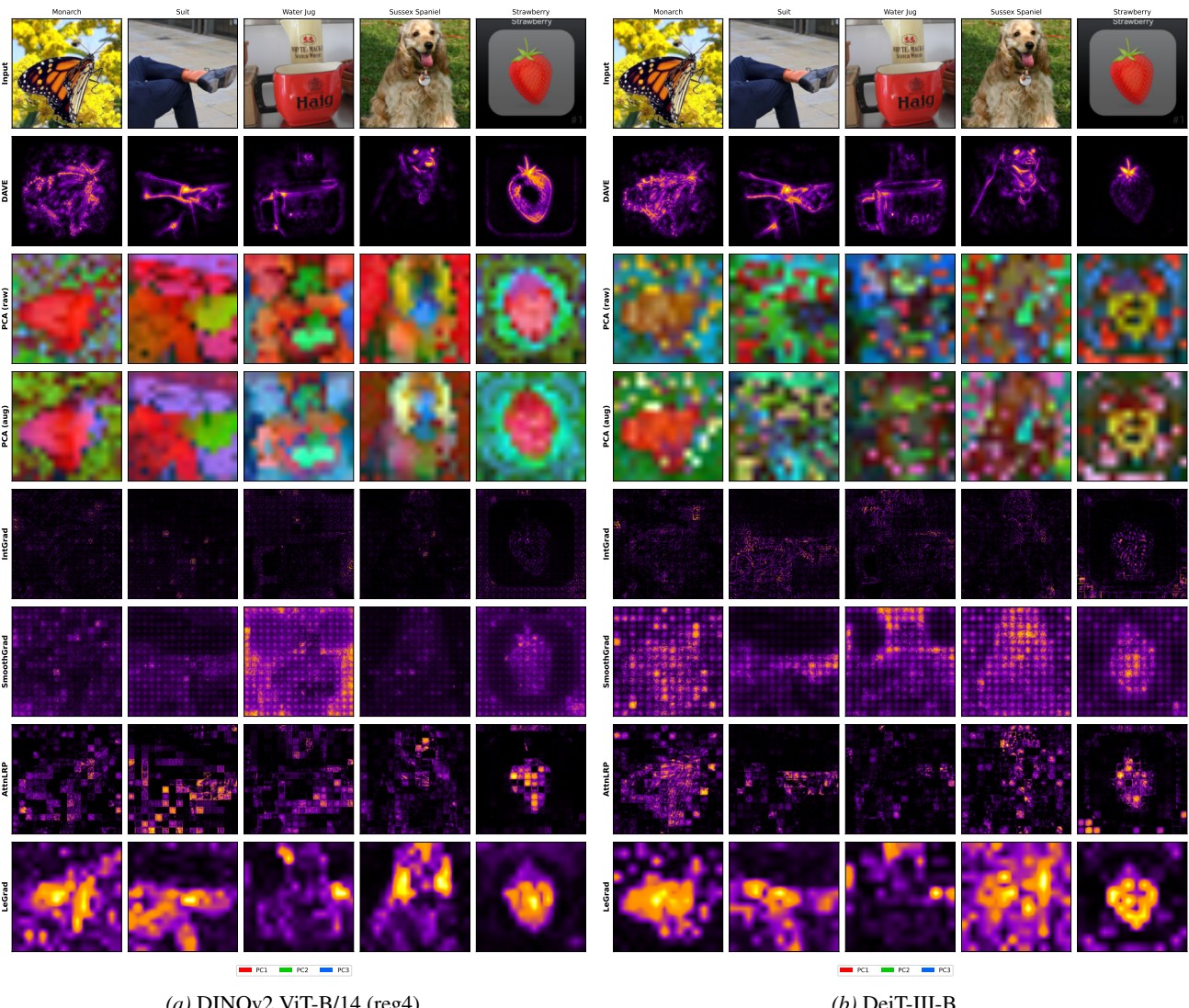

*(a)* DINOv2 ViT-B/14 (reg4)  *(b)* DeiT-III-B

*Figure 35.* **Qualitative comparison of attribution methods on DINOv2 with registers (left) vs. DeiT-III (right).** Rows show: input, DAVE, PCA (raw), and PCA (aug). For DINOv2 with register tokens (left), PCA produces visibly cleaner and more object-centric maps compared to DeiT-III (right), where PCA maps remain noisy and lack spatial coherence. However, while PCA benefits from the improved feature quality of register tokens, it still does not provide class-specific attribution — it highlights spatially coherent structure rather than discriminating between object categories. In contrast, **DAVE** consistently localizes the target class across both architectures. This is consistent with our quantitative results (Table 5), where PCA scores at chance level on GridPG despite reasonable EnergyPG scores. *Note: For PCA the top 3 components are displayed in* red, green, *and* blue *colors respectively.*

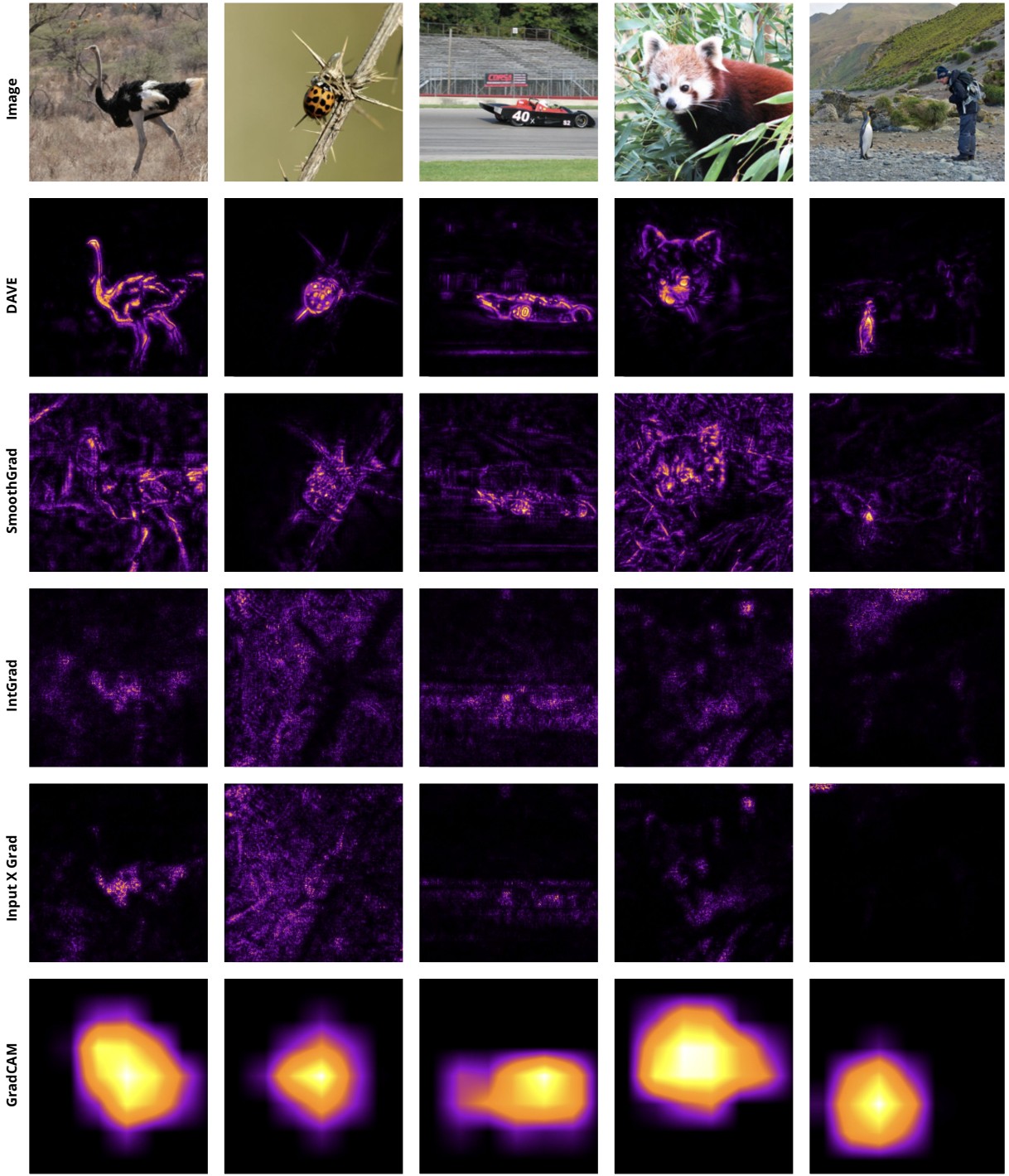

*Figure 36.* **Qualitative comparison of attribution maps for Convolutional ConvNeXt-S model** on ImageNet-1K images. Each row shows DAVE (ours), SmoothGrad, IntGrad, Input × Gradient and GradCAM. While not being designed for convolutional models, **DAVE achieves sharper and more semantically focused maps compared to other methods.**

# E. Completeness Discussion

Following FullGrad (Srinivas & Fleuret, 2019), we discuss the completeness axiom, i.e., whether the attribution sums to the model output, $\sum_p A(\boldsymbol{X})_p = F(\boldsymbol{X})$, where $A$ denotes the attribution and $F$ the network output, in the context of DAVE's effective-transformation formulation.

**Piecewise-linear models.** For piecewise-linear networks, the effective transformation $L(\boldsymbol{X})$ is locally constant and the operator-variation term in Eq. 3 vanishes. Input $\times$ Effective Transformation therefore reduces to the input gradient and satisfies completeness.

**Dynamic linear models without bias (e.g., B-cos ViTs).** For dynamic linear models without bias, $F(\boldsymbol{X}) = L(\boldsymbol{X})(\boldsymbol{X})$ by construction, so Input $\times$ Effective Transformation satisfies completeness exactly. In contrast, Input $\times$ Gradient does *not* satisfy completeness for these models, precisely because of the operator-variation term that DAVE discards.

**Dynamic linear models with bias (standard ViTs).** For dynamic linear models with bias, $F(\boldsymbol{X}) = L(\boldsymbol{X})(\boldsymbol{X}) + \boldsymbol{B}$, completeness is generally not satisfied, since the constant bias terms $\boldsymbol{B}$ are not attributed back to input features.

**Effect of neighborhood averaging.** DAVE's Reynolds-style equivariant averaging and low-pass filtering (Eq. 9) suppress high-frequency, locally non-equivariant components while preserving equivariant ones (Appendix A.3). Empirically, this trade-off yields the strongest GridPG, EnergyPG, and pixel-deletion performance, indicating that prioritizing stable, semantically meaningful signal improves attribution quality even at the cost of strict completeness.

# F. Limitations and Future Work

While DAVE achieves strong and stable attributions across architectures, a few limitations remain.

**Computational overhead.** DAVE estimates the distribution-aware effective transformation via Monte Carlo sampling over a neighborhood of transformed inputs, requiring one modified forward and backward pass per sample. This is more expensive than single-pass attribution methods. Although DAVE remains near the practical Pareto frontier (Main-Paper Figure 9, Appendix C.2) and 10 samples already surpass single-pass baselines, reducing this cost through more efficient or amortized sampling is a natural direction for future work.

**Transformation-group selection.** The neighborhood of transformations (rotation, translation, flip, noise) and their ranges are currently selected from per-model sensitivity analyses rather than learned (Appendix C.2). Automatically adapting the transformation group across architectures, for example to convolutional or hybrid backbones, would remove this manual step and may further improve transfer.

**Completeness.** For Vision Transformers with bias, DAVE prioritizes stable, locally equivariant attribution over strict completeness, and constant bias terms are not attributed back to input features (Appendix E). Restoring completeness would require additional, non-unique heuristics in the spirit of FullGrad (Srinivas & Fleuret, 2019), which we leave to future work.

**Downstream applications.** Our evaluation focuses on attribution quality (localization, faithfulness, and human interpretability) rather than downstream transfer. The stable, class-discriminative maps produced by DAVE may be useful for downstream tasks that rely on reliable region selection, such as prototype-based reasoning, attribution-guided distillation, and weakly supervised localization. We view these as promising directions rather than claims established in this work.

# G. Author Contributions

Adam Wróbel and Siddhartha Gairola contributed equally to this work.

**Adam Wróbel:** Conceptualization of DAVE, development of the mathematical framework, and implementation of the attribution method core.

**Siddhartha Gairola:** Experimental setup design, benchmark and ablation analysis, experimental validation.

**Adam Wróbel and Siddhartha Gairola:** Paper writing, result interpretation, manuscript revision, and final paper preparation.

