# OpenReview forum: "DAVE: Distribution-Aware Attribution via ViT Gradient Decomposition"
_ICML.cc/2026/Conference — ICML 2026 spotlight_

### Official Review · Reviewer_fMCC · 2026-03-01

**Soundness:** 2
**Presentation:** 3
**Significance:** 3
**Originality:** 2
**Overall Recommendation:** 4
**Confidence:** 4

**Summary:**

This paper introduces a visual attribution method called DAVE, which aims to generate stable, high-resolution pixel-level attribution maps. The key idea is to structurally decompose the input gradient into two parts: (1) an effective transformation L(X)  that captures the direct input-conditional linear action of the model; (2) an operator-variation term that captures the sensitivity of this transformation to input perturbations, which DAVE discards. To further suppress architecture-induced artifacts, a Reynolds-inspired equivariant filtering operator averages the effective transformations over small spatial transformations. A final Gaussian low-pass filtering step attenuates the remaining high-frequency fluctuations. The method is evaluated on ImageNet-1k across multiple ViT variants using localization metrics and pixel deletion faithfulness curves, showing improvements over multiple baselines.

**Compliance With Llm Reviewing Policy:**

Affirmed.

**Final Justification:**

The rebuttal thoroughly addresses all my concerns. The ablation study confirms that the effective-transformation decomposition is the key contributor rather than averaging alone, and the DINOv2 with register tokens experiments clearly demonstrate that DAVE maintains a substantial advantage over PCA. I am maintaining my score of Weak Accept.

**Key Questions For Authors:**

My main concerns have been elaborated in detail in the weaknesses section above. I'm looking forward to the author's responses to these questions.

**Limitations:**

yes

**Strengths And Weaknesses:**

Strengths

1. The mathematical derivation is clear and detailed, with a progressive construction and logical coherence, where each step targets a clearly identified problem.
2. The attribution visualizations are compelling, clear and explicit, providing visual evidence for understanding the model's decision-making process.
3. The framework is versatile and experiments have been conducted on various models, demonstrating its applicability.

Weaknesses

1. (Figure 3) The method proposed in the paper can effectively extract the target object from the image, but the core idea seems to be averaging over multiple transformations. Could you apply the same equivariant averaging to raw gradients or other baselines? This would clarify whether the improvement comes from the effective transformation decomposition or from the averaging procedure itself. Furthermore, even simpler approaches may achieve similar localization. For example, when using PCA, one can also directly extract the main objects in the image from the feature map, and averaging PCA results across multiple transformations may yield similar outcomes.

2. The paper demonstrates clear attribution maps. Could these attribution maps benefit downstream tasks such as object retrieval?

3. The method proposed in the paper seems to work well, but the overall process involves many steps and is computationally involved. Recent work (Darcet et al., 2024, "Vision Transformers Need Registers") addresses the artifact problem of ViTs by introducing register tokens to absorb outlier noise. It remains unclear whether a model like DINOv2, after using registers to remove outliers, can achieve similar results to your method through any baseline approach (such as AttnLRP or IntGrad).

---

> ### Author Rebuttal · Authors · 2026-03-31
>
> We thank the reviewer for the clear summary and for recognizing **the mathematical formulation, the visual quality of the attributions, and the applicability of the framework across models**. We also appreciate the questions regarding the role of equivariant averaging, downstream uses of the attributions, and comparisons to recent artifact-mitigation approaches.
>
> Additional figures/tables are provided in the **anonymous reference file [RF]** https://anonymous.4open.science/api/repo/anrepo-4E3C/file/5431_rebuttal.pdf?v=774bc52f.
>
> ### W1: Could the same equivariant averaging be applied to raw gradients or simpler baselines?
>
> Yes! This is an important question, and we have now included exactly this ablation.
>
> Replacing the effective transformation in DAVE with the **raw gradient** (i.e., using the same augmentation/averaging pipeline but without the decomposition) yields substantially noisier and less structured attributions (Figs. 9-10 [RF]) and fails to separate class-specific objects in multi-object scenes (Fig. 10 [RF]). Quantitatively, performance drops markedly (Table 1 [RF]): on DeiT-III / ViT-B, GridPG decreases by **21.63 / 12.15 pp** and EnergyPG by **3.20 / 1.36 pp**. This shows that the gain does **not** come from averaging alone; the effective-transformation decomposition is a key component.
>
>
> ### W2: DINOv2 with register tokens and PCA.
>
> We agree this comparison is important and have added experiments on **DINOv2 ViT-B/14 with register tokens (reg4)** ([Darcet et al., ICLR 2024](https://arxiv.org/abs/2309.16588)), together with PCA baselines and standard attribution methods (Table 4, Fig. 17 [RF]).
>
> **_PCA_** (both raw and augmentation-averaged) is inherently class-agnostic: both variants obtain **25.00% GridPG**, i.e., chance level for the 2x2 grid evaluation, while achieving only moderate **EnergyPG (61.54 / 65.38%)**, which reflects generic object saliency rather than class-specific attribution. Augmentation improves PCA only marginally (**+3.84 pp EnergyPG**), indicating that averaging alone is insufficient. DAVE clearly outperforms PCA by **+37.94 pp GridPG** and **+10.35 pp EnergyPG** over PCA (with augmentations). Qualitatively, Fig. 17 [RF] shows the same trend: PCA can produce cleaner object-centric structure on DINOv2 with registers, but still does not yield class-specific attributions, whereas DAVE consistently localizes the target class.
>
>
> **_Register tokens_** reduce artifacts, but they also do not close the gap for standard attribution methods: on **DINOv2 with registers DAVE reaches 62.94% GridPG / 75.73% EnergyPG**, compared with the best baselines A-LRP: 37.97% GridPG and LeGrad: 73.72% EnergyPG. This suggests that DAVE’s advantage comes from the attribution mechanism itself, not only from model-level artifact mitigation.
>
> ### W3: Could these attribution maps benefit downstream tasks such as object retrieval?
>
> This is an interesting direction! Our current work focuses on attribution quality rather than downstream transfer, so we do not make a direct claim for object retrieval here. That said, DAVE produces class-discriminative and spatially precise maps, which could plausibly be useful for downstream applications that benefit from reliable object localization or region selection. Potential future directions include applications to prototype-based models ([Choi et al., ICLR 2025](https://arxiv.org/abs/2410.08069)), attribution-guided distillation ([Parchami-Araghi et al., ECCV 2024](https://arxiv.org/abs/2402.03119)), and analysis of intermediate representations ([Boehle et al., CVPR 2022](https://arxiv.org/abs/2205.10268)), where DAVE’s stable and spatially precise attributions may be useful.
>
> We view these as promising future directions rather than claims established in the current paper.
>
>
> Overall, we will clarify more explicitly in the revision that (i) averaging alone is not sufficient, (ii) DINOv2 with registers and PCA baselines still remains clearly below DAVE, and (iii) downstream-task use is an interesting future direction rather than part of our current empirical claim.

---

> > ### Author Rebuttal · Reviewer_fMCC · 2026-04-01
> >
> > The rebuttal directly addresses all three concerns with concrete experiments. The ablation confirms that the effective-transformation decomposition is the key contributor rather than averaging alone, and the DINOv2 with register tokens experiments clearly show DAVE maintains a substantial advantage over PCA and standard baselines. I am maintaining my score

---

> > > ### Author Response · Authors · 2026-04-05
> > >
> > > Dear R-fMCC,
> > >
> > > Thank you for the follow-up and for the positive assessment.
> > >
> > > We appreciate that you found the concerns adequately addressed, and we are glad the additional experiments and analysis were helpful.

---

### Official Review · Reviewer_2a19 · 2026-03-10

**Soundness:** 3
**Presentation:** 3
**Significance:** 3
**Originality:** 3
**Overall Recommendation:** 5
**Confidence:** 4

**Summary:**

The authors introduce a new white box, gradient-based attribution method for ViT models that remedies existing issues with the patch-based structure of the underlying transformer architecture providing poor pixel-level attributions. To do so, they essentially create a new, more complex version of smooth grad that introduces not only noise but Euclidean transformations as well to find the components of the model that do not contribute to the decision and remove them from the explanation. By sampling a large number of these gradients and combining them, they produce greatly improved visual and quantitative results.

**Compliance With Llm Reviewing Policy:**

Affirmed.

**Final Justification:**

This is an interesting paper that solves a prevalent problem for ViTs, which has gone unaddressed in the literature. I was impressed by the work to begin with, and their well-crafted rebuttal allowed me to maintain positivity. This is a paper worthy of acceptance.

**Key Questions For Authors:**

If my description of the method in W1 holds, what would happen if this approach was applied to a CNN? Would attributions see further improvements over what smoothgrad already supplies?

Is there an explanation for the significant loss to smoothgrad shown in the deletion test of Figure 12 B-cos-ViT-B?

**Limitations:**

None discussed. None stand out.

**Strengths And Weaknesses:**

S1. There is great need for the method presented here, as it has been a known and hard-to-solve issue that ViT models do not play well with the model agnostic input-output gradient methods originally built for CNN models.

S2. The construction of the method is interesting and brings novel perspectives to tried-and-true techniques like smoothgrad for extracting high-quality and faithful pixel-level explanations from the ViT.

S3. Experiments for localization and faithfulness are performed with both pointing game and deletion tests across multiple models, and do not only use the default ViT architecture, showing extensibility and breadth. The results are convincing.

S4. Figure 2 is a very interesting figure that not only shows the value of DAVE but also indicates some motivation for how it works. The major shifts (on low importance features) between the Attn LRP attributions under minor input changes support the methodology. It would be good to also present this jointly as it is now but also as a motivational figure for the methodology.

W1. From looking at the code, it appears that my summary of this method as smoothgrad with the addition of Euclidean transforms in the input space is accurate. However, I find the math presented in the paper, while very nice from a rigor sense, hides the simplicity of the approach (unless I have misunderstood something). While I do not think this is intentionally done to make the approach seem more complicated, I think the paper would benefit from a subsection that puts it plainly in natural language. Figure 3 does a nice job at giving this impression, but I would want to see it written out.

W2. There is a lack of qualitative comparisons across models. There are not qualitative examples to fully support the quantitative results shown in quadrants 1, 3, and 4 of figure 12. A larger selection would support confidence of interpretable explanations across models.

W3. How many steps are methods like IG, smoothgrad, and A-LRP run with? Also 50? It would be good to mention this clearly so a reader can have a true runtime estimate.

C1. Figure 7 has the middle and last rows flipped, or the cap[tion is flipped

---

> ### Author Rebuttal · Authors · 2026-03-31
>
> We thank the reviewer for the insightful evaluation and for **recognizing the need for this method, the novelty of the construction, and the strength of the empirical results**. We appreciate the constructive suggestions regarding presentation clarity, qualitative comparisons, and implementation details.
>
> Additional figures/tables are provided in the **anonymous reference file [RF]** https://anonymous.4open.science/api/repo/anrepo-4E3C/file/5431_rebuttal.pdf?v=774bc52f.
>
>
> ### W1.1: Relation to SmoothGrad.
>
> The key distinction is that DAVE averages the **effective transformation**, whereas SmoothGrad averages the **raw gradient**. This difference is fundamental: DAVE estimates the input-conditioned operator underlying the forward computation, rather than local output sensitivity. Empirically, using the raw gradient as the DAVE baseline (i.e., augmented SmoothGrad) yields significantly noisier and less structured maps (Figs. 9-10 [RF]) and fails to separate class-specific objects in multi-object images (Fig. 10 [RF]). Quantitatively, performance degrades substantially (Table 1 [RF]): GridPG drops by **21.63 / 12.15 pp** and EnergyPG by **3.20 / 1.36 points**, confirming that operator variation introduces harmful noise.
>
> ### W1.2: Plain-language explanation.
>
> We agree that the intuition can be presented more directly.
>
> At a high level, DAVE consists of three steps:
> (1) **Layer formulation:** express each layer as an input-dependent linear operator with constant bias (Eq. 1; App. A.1 [Paper]).
> (2) **Effective transformation extraction:** recover this operator from the gradient by discarding the operator-variation term (Eq. 3 [Paper]), implemented via a modified forward pass and backward pass that detaches attention weights, GELU multipliers, and LayerNorm statistics.
> (3) **Neighborhood estimation:** average the resulting operator over transformed and perturbed inputs, yielding a stable, locally equivariant estimate that suppresses artifacts and high-frequency noise.
>
> We will add a short plain-language subsection and pseudocode in the revision. We also thank the reviewer for the suggestion regarding Fig. 2; we will use it more explicitly as a motivation figure.
>
> ### W2: Qualitative comparisons across models.
>
> We agree and have added additional qualitative comparisons across multiple ViT backbones in **Fig. 18 [RF]** (ViT-B/16, DeiT-B/16, DeiT-III-B/16), as well as on B-cos ViTs in **Fig. 19 [RF]**. These results show that DAVE consistently produces sharper, more object-aligned attributions across architectures, complementing the quantitative results. We will incorporate a broader qualitative selection into the main paper.
>
> ### W3: Number of steps / runtime estimate.
>
> We agree this should be stated more clearly. In the paper, we follow the authors’ official implementations / default hyperparameters: SmoothGrad (25 steps) and Integrated Gradients (200 steps). For both methods, we did not observe improvements from using more number of steps (see Fig. 3[RF] and Fig. 11[RF]). AttnLRP, Chefer-LRP, and LeGrad are all single-step methods.
>
> We additionally benchmark all methods in terms of number of steps vs. performance (via GridPG), as well as actual runtime and memory footprint on a fixed hardware setup (NVIDIA L40S, 46 GB VRAM). Results are summarized in Table 2[RF], where we report both runtime and GPU memory usage. This makes the runtime comparison transparent; we will state these settings explicitly in the revision.
>
> ### Q1: Can DAVE be applied to a CNN?
>
> Yes! DAVE is not limited to ViTs and can be applied to CNNs (e.g., ConvNeXt-S) as long as the effective transformation can be extracted from the gradient. However, the current augmentation design is motivated by ViT-specific artifacts and may not yet be optimal for CNNs. Even so, on ConvNeXt-S (Fig. 12 [RF]), DAVE produces sharper and more semantically focused maps than gradient-based methods without any architecture-specific tuning. We agree it would be interesting to explore CNN-specific transformations further.
>
> ### Q2: Why did SmoothGrad appear stronger on B-cos-ViT-B in the original deletion plot?
>
> We identified an inconsistency in attribution post-processing specific to the B-cos pixel-perturbation evaluation: [Captum Library’s Saliency](https://captum.ai/api/saliency.html) applies absolute values by default, whereas the other baselines there were evaluated with signed attributions. This made SmoothGrad appear artificially stronger under ascending deletion order. After harmonizing the sign convention across methods for this evaluation, DAVE achieves the highest deletion AUC on B-cos-ViT-B, consistent with the results on the other architectures (updated in Fig. 8 [RF]). This is also supported by the qualitative results in Fig. 19 [RF].
>
>
> ### C1: Figure 7 caption/row mismatch.
>
> Thank you for catching this; we will correct the figure/caption mismatch in the final version.

---

> > ### Author Rebuttal · Reviewer_2a19 · 2026-03-31
> >
> > Thank you for the high quality response. You have addressed all of my concerns very thoroughly. I am being nitpicky here and I *do not* expect a reply to this, but I think you meant to indicate in RF Fig 8 that pixels are deleted in ascending order if a larger AUC is better. I only mention it so that it is changed correctly if used in a final version.
> >
> > Overall, I stand by my acceptance. Good Luck!

---

> > > ### Author Response · Authors · 2026-03-31
> > >
> > > Dear R-2a19,
> > >
> > > Thank you for the careful follow-up and for catching this. You are right: in RF Fig. 8, pixels should be described as being deleted in **increasing** (ascending) order of attribution, consistent with the protocol used throughout the paper. We will correct this in the final version.
> > >
> > > We also appreciate your positive assessment and support.

---

### Official Review · Reviewer_MqRW · 2026-03-11

**Soundness:** 2
**Presentation:** 3
**Significance:** 3
**Originality:** 2
**Overall Recommendation:** 3
**Confidence:** 3

**Summary:**

This paper proposes DAVE, an attribution method for Vision Transformers that decomposes the input gradient into an input-conditioned “effective transformation” and an operator-variation term, then suppresses the latter to avoid instability. The effective transformation is further filtered via local equivariance averaging over small spatial transformations and low-pass smoothing over Gaussian perturbations to reduce architecture-induced artifacts and noise. On several ViT backbones and B-cos models, DAVE achieves stronger localization and flatter pixel-deletion curves than prior gradient- and attention-based baselines, while yielding sharper pixel-level attributions.

**Compliance With Llm Reviewing Policy:**

Affirmed.

**Key Questions For Authors:**

1.Suppressing operator mutation: Does it remove class-related signals (e.g., attention shifts) and affect the integrity of the results?

2.Could you report the actual runtime/memory footprint and the speedup compared to the baseline?

3.Please provide clearer pseudocode and detailed implementation/memory footprint breakdowns for reproducibility.

4.Please quantify the performance gains from rotation/translation/flip, test for range sensitivity, and discuss the trade-off between clarity and invariance.

**Limitations:**

Yes

**Strengths And Weaknesses:**

Strengths:
1. This paper is clearly structured and logically presented, with a well-organized description of the motivation, methods, and technical solutions. The proposed decomposition strategy provides a reasonable approach to alleviate the instability and artifact problems existing in current ViT attribution methods.

2.The experimental section is detailed, and the results are presented intuitively and understandably. On multiple ViT backbone networks and B-cos models, this proposed method outperforms the considered baseline methods in terms of localization accuracy, pixel deletion curves, and pixel-level attribution quality.

Weakness:
1.Discarding operator mutations may eliminate meaningful sensitivities related to model mechanisms (e.g., attention weight variations), which may possess categorical information; axiomatic effects (e.g., completeness) should be appropriately discussed.

2.The computational cost is high (requiring 50 Monte Carlo samples per input), but aside from Big O notation, no comparisons of runtime and actual runtime to baseline are reported.

3.Implementation details regarding the computation of the effective operators across the entire network using a single forward/backward propagation are brief; more explicit pseudocode, memory considerations, and how to handle separation operations layer by layer will help improve reproducibility.

4.The ablation effect of isovariate groups is limited: performance before and after rotation, translation, and flipping is not quantified in the text; sensitivity to range is not addressed; and the trade-off between clarity and invariance is not considered.

---

> ### Author Rebuttal · Authors · 2026-03-31
>
> We thank the reviewer for the thoughtful evaluation and for recognizing the **utility** and **strong empirical performance** of our method. Below we address the concerns on operator variation/completeness (W1/Q1), computational cost (W2/Q2), implementation details (W3/Q3), and augmentation design (W4/Q4).
>
> **Additional figures/tables are in [RF]** https://anonymous.4open.science/api/repo/anrepo-4E3C/file/5431_rebuttal.pdf?v=774bc52f.
>
>
> ### W1.1 /Q1: Discussion on operator variation
>
> Thank you! We agree and will add this discussion in the final version.
>
> Our goal is to explain the model’s **effective computation**. DAVE therefore uses the effective transformation (Eq. 1 in the paper) and removes the operator-variation term from Eq. 3 as the attribution baseline. It does **not** discard class-related structure entirely: DAVE reintroduces input-dependent effects in a more stable form by averaging the effective transformation over a local neighborhood of augmentations. Thus, unstable high-frequency sensitivity from operator-variation is suppressed, while stable class-relevant structure is retained. Empirically, replacing the effective transformation with the full gradient inside the same DAVE pipeline yields much noisier maps (Figs. 9-10 [RF]) and significantly worse localization: on DeiT-III / ViT-B, GridPG drops by **21.63 / 12.15 pp** and EPG by **3.20 / 1.36 pp** (Table 1 [RF]).
>
>
> ### W1.2/Q1: Completeness discussion
>
> DAVE prioritizes **stable, locally equivariant attribution** over a complete but noisy explanation.
>
> Following FullGrad ([Srinivas et al., 2019](https://arxiv.org/abs/1905.00780)), we discuss completeness under the condition $F(x)=\sum A(x)$, where $A$ denotes the attribution and $F$ the network output.
>
> 1. For linear and piecewise-linear models, the effective transformation is locally constant and equivalent to the input gradient, so Input $\times$ Effective Transformation satisfies completeness.
> 2. For dynamic linear models $F(x)=W(x)x$ (such as the considered B-cos ViTs), it also satisfies completeness, whereas Input $\times$ Gradient does not, due to the operator-variation term.
> 3. For dynamic linear models with bias, $F(x)=W(x)x+b$ (e.g., ViT layers), completeness is generally not satisfied, since constant bias terms are not attributed. Restoring completeness would require additional, non-unique heuristics (as in FullGrad), which we leave as part of future exploration.
>
> DAVE’s neighborhood averaging suppresses high-frequency (noisy) artifacts while enforcing local equivariance. Empirically, this tradeoff yields the **strongest GridPG, EnergyPG, and deletion performance**, showing that stable, semantically meaningful signal improves attribution quality.
>
> ### W2/Q2: Runtime / memory comparisons.
>
> We have added explicit runtime and memory benchmarking on DeiT-III-B/16 (NVIDIA L40S, 46 GB VRAM) for all baselines (Table 2, Fig. 3 [RF]). At **10 steps**, DAVE reaches **65.6 GridPG** in **66.7 ms** with **1058 MB** peak memory, surpassing all single-pass baselines. At **50 steps** (used in the paper), DAVE reaches **65.7 GridPG** in **248.3 ms** with **2044 MB** peak memory. For comparison, A-LRP obtains 54.5 GridPG at 113.8 ms; C-LRP 53.8 at 145.1 ms. While DAVE is more expensive than single-pass methods, the localization gain is substantial and the memory footprint remains modest (**<5% of 46 GB VRAM at 50 steps**).
>
> We will add these runtime / memory comparisons.
>
>
> ### W3/Q3: Implementation details / reproducibility.
>
> Certainly! DAVE computes layer-wise effective transformations by reformulating each layer as in Sec. 3/App. A.1 of the paper and detaching the input-dependent components (e.g., attention weights, gating terms, and normalization statistics) during the modified forward pass. A single backward pass then composes these effective transformations across layers to obtain the effective transformation. Each Monte Carlo sample therefore requires one modified forward/backward pass, and the final attribution is the average across sampled perturbations/transforms.
>
> Code is in the supp material; we will add pseudocode and a clearer per-layer description.
>
>
> ### W4/Q4: Augmentation ablation / range sensitivity.
>
> We quantify each augmentation in Table 3 [RF].
>
> Rotation and translation provide the largest gains (**+10-14 pp** over no augmentation), while horizontal flip alone has negligible effect; progressive noise adds a smaller complementary gain (**~1-2 pp**) on top of geometric transforms, reaching **65.8%** on DeiT-B/16 and **68.4%** on DeiT-III-B/16.
>
> We agree that the choice of transformation ranges should be clarified. The rotation interval and noise levels in DAVE are selected from model sensitivity analyses in the appendix (Figs. 14-15); we also include translation sensitivity (Fig. 1 [RF]). These parameters are chosen from model sensitivity rather than eval metrics, avoiding tuning on evaluation metrics.
> We will clarify the tradeoff between invariance and attribution sharpness.

---

> > ### Author Rebuttal · Reviewer_MqRW · 2026-04-05
> >
> > Thanks for authors' response. I will keep my rating.

---

### Official Review · Reviewer_VEiC · 2026-03-12

**Soundness:** 4
**Presentation:** 4
**Significance:** 4
**Originality:** 4
**Overall Recommendation:** 6
**Confidence:** 4

**Summary:**

This paper proposes DAVE, a new attribution method designed to produce stable and high-resolution explanations for ViTs. The method is based on a structured decomposition of the input gradient, which separates stable, locally equivariant components from architecture-induced artifacts. By leveraging architectural properties of ViTs, DAVE suppresses artifacts caused by patch embeddings and attention routing, enabling precise pixel-level attribution. As a result, DAVE generates robust and class-consistent attribution maps that remain stable under small image perturbations. Experiments on multiple benchmarks show that DAVE produces more spatially precise and stable explanations than existing attribution methods across both supervised and self-supervised ViT models.

**Compliance With Llm Reviewing Policy:**

Affirmed.

**Final Justification:**

Personally, this is the best paper on ViT's interpretability that I have seen in recent years. The proposed method not only provides comprehensive theoretical support but also achieves very strong visual performance, which is particularly valuable. I believe this method will become an important baseline in the future. Therefore, I will increase my score. I hope the authors will consider open-sourcing the code to support applications in other domains.

**Key Questions For Authors:**

1.	Is the proposed method class-agnostic? Specifically, when the model produces incorrect predictions, what do the corresponding attribution heatmaps look like? Providing more visualizations in such cases would help better understand the behavior of the method. In addition, it would be valuable to evaluate whether the approach remains reliable under OOD scenarios.
2.	The evaluation only reports the deletion score. Why were other commonly used interpretability metrics, such as the addition score and Pointing Game, not included? Furthermore, it is unclear whether the reported deletion score represents the mean value across the entire dataset or results from a subset of samples.

**Limitations:**

The authors could further discuss the limitations of the proposed method and outline possible directions for future work.

**Strengths And Weaknesses:**

The paper is well written and technically sound. The analysis of interpretability in the method section is convincing, and the proposed approach produces high-quality heatmaps, demonstrating the effectiveness of the method. Overall, this is a strong paper that provides a new perspective on visual interpretability for ViTs.

---

> ### Author Rebuttal · Authors · 2026-03-31
>
> We thank the reviewer for the positive evaluation and for recognizing **the technical soundness, clarity of presentation, and effectiveness of our method.** We also appreciate the questions regarding class-specific behavior, OOD robustness, evaluation metrics, and limitations, which will help us further improve and clarify the paper
>
> Additional figures/tables are provided in the **anonymous reference file [RF]** https://anonymous.4open.science/api/repo/anrepo-4E3C/file/5431_rebuttal.pdf?v=774bc52f.
>
> ### Q1.1: Attribution for incorrect predictions.
>
> That is a very interesting question.
>
> DAVE is a **class-conditional** attribution method: the heatmap is computed with respect to a chosen target class (e.g., the predicted class or the ground-truth class). To better understand its behavior on model errors, we include additional visualizations on misclassified samples in **Fig. 13 [RF]**. These show that DAVE produces spatially distinct attribution maps for the predicted and ground-truth classes even when the model is wrong, highlighting different semantically meaningful regions for the two classes. This helps explain both why the model made the incorrect prediction and what evidence would support the correct class.
>
> We also include a multi-class comparison in **Fig. 14 [RF]**, where DAVE remains spatially distinct and class-specific across several candidate classes, whereas baselines often produce similar or noisier maps.
>
> We will emphasize this behavior and include these additional examples in the final version.
>
> ### Q1.2: Behavior under OOD samples.
>
> Thank you for the suggestion.
>
> To assess robustness under distribution shift, we include additional experiments on **ImageNet-R** ([Hendrycks et al., ICCV 2021](https://arxiv.org/abs/2006.16241)), which contains out-of-distribution samples with strong stylistic variation (e.g., paintings, cartoons, sketches). As shown in **Figs. 15-16 [RF]**, DAVE produces consistent and class-discriminative attributions across both in-distribution ImageNet samples and OOD ImageNet-R samples. In particular, it localizes similar semantic object parts despite substantial changes in texture and style, suggesting that it captures class-relevant structure rather than domain-specific cues. In contrast, baseline methods degrade under distribution shift, often producing noisier or less focused attributions.
>
> We will emphasize these results and discussion in the final version.
>
> ### Q2: Additional evaluation metrics.
> Beyond deletion, we report **GridPG** and **EnergyPG** (Tables 1-2 in main paper) on ImageNet-1k localization, and additionally include the **addition/insertion** metric (Fig. 2 [RF]). DAVE achieves the best AUC (**66.7**), outperforming all baselines (AttnLRP 63.3, IG 63.8, LeGrad 60.7, SmoothGrad 55.5).
>
> We further conducted **two user studies** with **120 participants**. In a **subjective preference task**, DAVE is consistently preferred over all baselines (**70-77\%**, $p<0.001$; Fig. 4-5 [RF]). In a **class-identification task** (similar to the HIVE study, [Kim et al., ECCV 2022](https://princetonvisualai.github.io/HIVE/)), DAVE achieves the highest accuracy **67.5\%**), outperforming LeGrad (47.3\%), IG (52.4\%), AttnLRP (52.8\%), Chefer-LRP (46.3\%), and SmoothGrad (52.0\%) ($p<0.01$; Fig.6-7 [RF]). These results indicate that DAVE provides more intuitive and informative explanations.
>
> **Regarding protocol details**, GridPG is evaluated on 500 constructed $2\times2$ grids (4000 ImageNet-1k validation images), EnergyPG uses ILSVRC bounding boxes (50k images), and deletion uses a 5k subset, following standard protocols  ([Chefer et al., CVPR 2021](https://arxiv.org/abs/2012.09838), [Boehle et al., CVPR 2021](https://arxiv.org/pdf/2104.00032)).
>
> We will clarify these details and highlight the user study results in the final version.
>
> ### Q3: Discussion on limitations and future work.
>
> While DAVE performs strongly in our experiments, there are some natural directions for future work. (1) The transformation group used for neighborhood averaging is currently chosen based on empirical analysis and could potentially be adapted more automatically across architectures. (2) The current augmentation design is motivated mainly by ViT-specific artifacts, and further tuning may improve transfer to other model families (as shown by initial results on CNNS, specifically ConvNeXt-S (Fig. 12 [RF])). (3) In addition, reducing the computational cost of the current Monte Carlo estimation would be useful to explore.
>
> We will integrate this discussion more prominently in the final version.

---

> > ### Author Rebuttal · Reviewer_VEiC · 2026-04-07
> >
> > Personally, this is the best paper on ViT's interpretability that I have seen in recent years. The proposed method not only provides comprehensive theoretical support but also achieves very strong visual performance, which is particularly valuable. I believe this method will become an important baseline in the future. Therefore, I will increase my score. I hope the authors will consider open-sourcing the code to support applications in other domains.

---

> > > ### Author Response · Authors · 2026-04-07
> > >
> > > Dear R-VEiC,
> > >
> > > Thank you very much for your follow-up and the positive assessment. We appreciate your support and are glad that the additional analyses and experiments helped clarify the strengths of the paper.
> > >
> > > Regarding open-sourcing the code, we are already working on making our code public and will release it with the final version of the paper.

---

### Decision · Program_Chairs · 2026-04-30

**Decision:**

Accept (spotlight)

**Comment:**

Producing stable, high-resolution attribution maps for Vision Transformers is difficult due to architectural artifacts from patch embeddings and attention routing. This paper proposes a method that decomposes input gradients to isolate locally-equivariant components from architecture-induced artifacts. This leads to clean class-consistent attribution maps with improved spatial precision and stability.
This paper tackles and interesting problem and is very well executed. The proposed method is well motivated, sound and works well.
All reviewers saluted the strong empirical results, the quality of the writing and clarity of figures. Finally - the proposed approach is widely applicable across ViT architectures.
Most of reviewer's concerns were addressed.

For all the above reasons, I firmly recommend this paper for acceptance.